# Laboratory study of the heterogeneous ice nucleation on black carbon containing aerosol

Leonid Nichman[1, 4, *], Martin Wolf[2], Paul Davidovits[1], Timothy B. Onasch[1, 4], Yue Zhang[1, 4, 5], Doug R. Worsnop[4], Janarjan Bhandari[6], Claudio Mazzoleni[6], Daniel J. Cziczo[2, 3]

[1]Department of Chemistry, Boston College, Chestnut Hill, MA, 02467, USA
[2]Department of Earth, Atmospheric and Planetary Sciences, Massachusetts Institute of Technology, Cambridge, MA, USA
[3]Department of Civil and Environmental Engineering, Massachusetts Institute of Technology, Cambridge, MA, USA
[4]Aerodyne Research, Inc, Billerica, MA, 01821, USA
[5]Department of Environmental Science and Engineering, Gillings School of Global Public Health, University of North Carolina at Chapel Hill, Chapel Hill, NC, 27599, USA
[6]Department of Physics and Atmospheric Sciences Program, Michigan Technological University, Houghton, MI, 49931, USA.
(*) Flight Research Laboratory, National Research Council of Canada, Ottawa, ON, K1V 9B4, Canada

*Correspondence to*: Leonid Nichman (leonid.nichman@nrc-cnrc.gc.ca)

**Abstract.** Soot and black carbon (BC) particles are generated in the incomplete combustion of fossil fuels, biomass, and biofuels. These airborne particles affect air quality, human health, aerosol-cloud interactions, precipitation formation, and climate. At present, the climate effects of BC particles are not well understood. Their role in cloud formation is obscured by their chemical and physical variability, and by the internal mixing states of these particles with other compounds. Ice nucleation in field studies is often difficult to interpret. Nonetheless, most field studies seem to suggest that BC particles are not efficient ice nucleating particles (INP). On the other hand, laboratory measurements show that in some cases, BC particles can be highly active INP under certain conditions. By working with well-characterized BC particles, our aim is to systematically establish the factors that govern the ice nucleation activity of BC. The current study focuses on laboratory measurements of the effectiveness of BC-containing aerosol in the formation of ice crystals in temperature and ice supersaturation conditions relevant to cirrus clouds.

We examine ice nucleation on BC particles under water-subsaturated cirrus cloud conditions, commonly understood to be deposition mode ice nucleation. We study a series of well-characterized commercial carbon black particles with varying morphologies and surface chemistries, as well as ethylene flame-generated combustion soot. The carbon black particles used in this study are proxies for atmospherically relevant BC aerosols. These samples were characterized by electron microscopy, mass spectrometry, and optical scattering measurements. Ice nucleation activity was systematically examined in temperature and saturation conditions ranging between $217 \leq T \leq 235$ K; $1.0 \leq S_{ice} \leq 1.5$; and $0.59 \leq S_{water} \leq 0.98$, respectively, using a SPectrometer for Ice Nuclei (SPIN) instrument, which is a continuous flow diffusion chamber coupled with instrumentation to measure light scattering and polarization. To study the effect of coatings on INP, the BC-containing particles were coated with organic acids found in the atmosphere, namely, stearic acid, cis-pinonic acid, and oxalic acid.

The results show significant variations in ice nucleation activity as a function of size, morphology and surface chemistry of the BC particles. The measured ice nucleation activity dependencies on temperature, supersaturation conditions, and the physicochemical properties of the BC particles are consistent with an ice nucleation mechanism of pore condensation followed by freezing. Coatings and surface oxidation modify the initial formation efficiency of pristine ice crystals on BC-containing aerosol. Depending on the BC material and the coating, both inhibition and enhancement in INP activity were observed. Our measurements at low temperatures complement published data, and highlight the capability of some BC particles to nucleate ice under low ice supersaturation conditions. These results are expected to help refine theories relating to soot INP activation in the atmosphere.

## 1 Introduction

Ice nucleating particle (INP) types in the atmosphere vary widely across the globe. Although their number concentrations are typically low, their atmospheric impact, governing ice cloud formation and properties, is significant. The role of soot in atmospheric ice nucleation (IN) processes remains poorly understood. The indirect effect of soot particles, particularly on upper tropospheric cirrus clouds and aviation contrails, may result in either positive or negative forcing depending on the type of soot and the ambient conditions (Zhou & Penner, 2014). Bond et al. (2013) modelled the contribution of soot to clouds and climate and distinguished between the homogeneous and heterogeneous freezing mechanisms. They showed that in the case of ice clouds when homogeneous nucleation dominates, coverage of high clouds is reduced and cooling prevails while when heterogeneous nucleation of BC prevails, more high clouds are formed that in turn contributes indirectly to the warming effect. The increasing number concentration of emitted soot particles since the preindustrial times (Bond et al., 2013; Lavanchy et al., 1999), the emissions of soot particles in the upper troposphere from aviation (Lee et al., 2009), and estimates that the concentration of soot will remain high in the near future (Gasser et al., 2017) underscore the importance in understanding the efficiency of soot particles acting as INP.

Current field results are inconclusive about the efficiency of soot particles in initiating ice nucleation. Recently, Chen et al. (2018) collected aerosol samples in a highly polluted environment and subsequently measured their IN activity. The soot samples showed no correlation between IN activity and black carbon (BC) fraction of the aerosol. Low BC activity was also found by Pratt et al. (2009) in their analysis of airborne particle residuals in orographic wave clouds, and by Eriksen-Hammer et al. (2018) in mixed phase clouds at the high-altitude research station Jungfraujoch. Other field studies showed higher IN activity of soot (Petzold et al., 1998). Phillips et al. (2013) suggested that black carbon is a major type of INP in clouds influenced by biomass-burning particles, however they couldn't rule out that another INP species, internally mixed with soot, might have nucleated the observed ice. Likewise, Levin et al. (2014) concluded that fires could be a significant source of INP in mixed-phase clouds.

Laboratory studies of soot INP in the cirrus cloud regime reveal a widespread IN activity (e.g. Kulkarni et al. (2016); Friedman et al. (2011); Ullrich et al. (2017); Demott et al. (2009); Häusler et al. (2018)). Kulkarni et al. (2016) demonstrated the activity of 120 nm uncoated diesel soot particles at 3 temperatures below the homogeneous freezing line with evidence of a temperature dependence below - 40 ºC. The reported effective density of these diesel soot particles was 0.58 g cm$^{-3}$ and their compaction did not affect their IN activity. Kulkarni et al. (2016) concluded that

the variability between their and previously reported values of soot IN activity might be associated with size or radius of the curvature of soot nanostructures and the different physio-chemical surface properties. Möhler et al. (2005a) and Ullrich et al. (2017) showed a clear IN activity for soot particles, well below the homogeneous freezing line in a 'U' shape curve. Ullrich et al. (2017) further parameterized 5 types of BC of various diameters (190 - 730 nm) and associated the shape of the curve, at the higher temperatures, to the transition from classical nucleation theory (CNT) to pore condensation and freezing (PCF) mechanism (Marcolli, 2014). On the other hand, Friedman et al. (2011) observed purely homogeneous freezing for soot particles 100 – 400 nm in diameter. Their measured droplet activated fractions and breakthrough of bare soot particles did show however, a size effect at - 20 and - 30 °C, with the largest sized particles showing droplet nucleation at relatively lower RH values. At - 40 °C, Friedman et al. (2011) reported evidence for 400 nm soot particles freezing below the threshold for homogeneous freezing, but they could not validate the result due to the uncertainty in their RH measurement. Friedman et al. (2011) also reported that oxidation of the surface of soot particles by ozonolysis, surprisingly, did not significantly change the activity of the uncoated soot INP. Another study by Demott et al. (2009) demonstrated homogeneous freezing for biomass combustion particles at - 46, -51, and -60 °C. However, they noted that the degree of lowering of $RH_w$ for heterogeneous versus homogeneous freezing in the haze particle regime at temperatures below - 40 °C is not well known and may depend on particle size. Therefore, their result could fall within the uncertainty of their measurement and miss the heterogeneous freezing. A laboratory study by DeMott et al. (1999) have shown that the ice can form on soot, similarly to the formation of visible contrails (condensation trails) behind aircraft. The so-called soot-induced cirrus cloud formation is also described by Jensen and Toon (1997) and Kärcher et al. (2007), where the exhaust soot disperses to form or modify cirrus cloud. Persistent linear contrails and induced-cirrus cloudiness, also known as aviation-induced cloudiness, are predicted to increase in the coming years as the importance of aviation, and its consequent climate impact, will continue to increase (Lee et al., 2009). Other laboratory experiments indicate that the heterogeneous IN activity of soot in the deposition mode may be minimal (e.g. DeMott 1990, Kanji and Abbatt, 2006) and therefore would not contribute to cirrus coverage. A more exhaustive list of studies that demonstrate the currently known widespread in IN activity of soot can be found in reviews by Hoose and Möhler, (2012), Cziczo et al. (2016), and Kanji et al. (2017).

Both field and laboratory studies indicate that coating of soot particles in the atmosphere, in most cases, decreases their IN activity compared to their bare counterparts (Kärcher et al., 2007, Möhler et al., 2005a). Further, Crawford et al. (2011) showed that propane burner soot can be a highly active INP in the deposition mode if it has a sufficiently low organic carbon content and is uncoated.

Modeling INPs requires quantitative parameterization of the IN activity. Two common approaches to parameterize IN of atmospherically relevant particles include a stochastic description based on classical nucleation theory and a deterministic or singular description (Vali, 2014; Knopf et al, 2018 and references therein). For the latter, the pragmatic description of active site density ($n_s$), is often used (Connolly et al., 2009; Niemand et al., 2012; Vali, 2014). As demonstrated in Niemand et al. (2012), if the activated ice fraction is small (< 0.1), the active site density can be expressed as the fraction of the ice particles out of the total aerosol concentration divided by the averaged particle surface area. This approach describes $n_s$ as a function of temperature, allowing for intercomparison between independent observations and subsequent parametrization for modelling purposes, but does not take into account the

kinetics (i.e. time dependence) of nucleation (Welti et al., 2012). In this study, we report results suitable for both types of parameterizations, time dependent (Vali, 1994; Vali, 2014; Murray et al., 2012; Sect. 3.1) and time independent (Sect. 3.2), for INP representation in models.

A kinetic-based PCF mechanism provides, at least in part, a possible explanation for the widespread IN activity observed for soot particles. This mechanism was formulated to explain ice nucleation by porous materials (Everett, 1961; Blachere and Young, 1972). The PCF mechanism proposes that empty spaces between aggregated primary particles fill with water due to capillary condensation at relative humidities ($RH_w$) below water saturation, which freezes homogeneously (Higuchi and Fukuta (1966); Christenson (2013); Marcolli (2014); David et al. (2019)). The PCF mechanism, which models ice formation inside small pores using homogeneous freezing theory, describes the heterogeneous formation of ice on solid particles, such as soot particles, due to the presence of the porous structure. Therefore, we refer to PCF as a heterogeneous ice freezing mechanism (Vali et al., 2015). The diameter of mesopores (2-50 nm) and the surface properties of the pore substrate affect the condensation process in the particles, especially for pores less than 10 nm in diameter where the Kelvin effect is greatest. In large pores (>> 10 nm), the water vapor pressure is not sufficient to cause condensation below water saturation (< 100 % $RH_w$). On the other hand, in pores with diameters too small (<4 nm), the growth of an ice embryo may be inhibited (Marcolli, 2014; Vali et al., 2015). Pore diameters in soot materials range from micro (<2 nm pore diameter), through meso (2-50 nm diameter), to macro (>50 nm diameter) and are dependent on the specific soot material. Manufactured carbon black material (e.g., Kruk et al., 1996) can be produced with similar or higher surface areas and mesoporosity than combustion-related soot particles (e.g., Rockne et al., 2000). All else being equal, IN activity of a material is expected to increase with increasing number of pores in the suitable diameter range (Vali, 2014). The PCF mechanism also predicts the observed decrease in ice nucleation activity with increasing temperature over the temperature range about 210 K to 240K (typical of cirrus clouds, Hoose and Möhler, 2012). Porous materials such as mesoporous silica, zeolites, porous silicon, porous glass, and carbon nanotubes have morphologies similar to those of soot particles (Marcolli (2014, 2017); Wagner et al. (2016); Ullrich et al. (2017); Mahrt et al. (2018), Umo et al. (2019)). Therefore, the PCF mechanism may be applicable to soot particles as well.

Soot particles are emitted directly into the atmosphere from combustion processes such as agricultural burning, forest fires, domestic heating and cooking, and transportation (McCluskey et al., 2014; Arora & Jain, 2015; Vu et al., 2015; Sakamoto et al., 2016) and are ubiquitous in the Earth's troposphere (Heintzenberg, 1989; Seinfeld and Pandis, 1998; Pósfai et al., 1999; Finlayson-Pitts and Pitts, 2000; Murphy et al., 2006). The chemistry and structure of soot depends on the type of fuel, combustion temperature, combustion kinetics and chemistry (Marcolli, 2014 and references therein; Murr and Soto, 2005). In general, soot particles consist primarily of elemental carbon with a chain-like structure of aggregated primary spherules, which are typically tens of nanometers in diameter (Buseck et al., 2014). Soot and black carbon nomenclature is often used interchangeably for particles with negligible organic matter content. The size or mass of a soot particle depends upon the number of primary spherules and their arrangement as chain-like aggregates and agglomerates of aggregates, as illustrated schematically in Fig. 1a. Electron-microscope images of an aggregate and a compact agglomerate of ethylene combustion soot are shown in Figs. 1b and 1c, respectively.

The compact agglomerate structures contain pores. The edges of the agglomerate might have some external branched aggregates (Fig. 1c) that do not contribute to the porous structure. In this study we will refer to any confined empty spaces between aggregates as pores. Note that in this view, pores can occur both within and on the surface of the particle.

The critical factors that make some BC aerosols effective IN agents have not fully been established experimentally. To elucidate these issues, we measured the ice nucleation properties for a series of five well-characterized commercial carbon black samples (proxies of BC) with varying morphologies and surface chemistries, as well as for ethylene flame-generated combustion soot. The studies were performed under systematically varied temperature in the range 217 – 235 K, simulating cirrus cloud-forming conditions (Krämer et al., 2016). In this connection, we studied the following factors on soot particle ice nucleation activity:

(1) **Effect of particle morphology.** In the PCF mechanism, the dimensions and shape of the pores play an important role (Marcolli, 2017). In BC particles, it is not clear whether the PCF mechanism occurs in the empty spaces between primary spherules or in the pores formed between the aggregates (see Fig. 1). Therefore, spherule size, the degree of branching in a single aggregate, the stereo arrangement of the aggregates, and the location of the pores may affect the IN activity of particles with similar mobility diameter. We studied the change in IN activity in agglomerates of the same selected mobility diameter but different internal spatial configuration of aggregates of spherules.

(2) **Effect of particle generation.** A question has been raised whether water processing of soot in the atmosphere and in the laboratory reduces the IN activity of soot. A possible explanation to the IN activity discrepancies observed in past laboratory studies of BC particles were ascribed to the technique of aerosol generation. The technique of aerosol generation can often affect the morphology of the particle through a compaction mechanism (China et al., 2015a), which can then change the density and therefore affect the IN activity of the particle. Some laboratory studies (e.g. Ma et al., 2013; Friedbacher et al., 1999) showed that a water droplet that encloses an aggregate followed by subsequent water evaporation in the diffusion dryer will lead to the collapse of the soot structure driven by the water surface tension exerted on the aggregate core. However, this compaction from aqueous suspensions was observed only in the laboratory and only for aggregates of approximately 200 nm in diameter (e.g. Ma et al., 2013; Khalizov et al., 2013). Some suggest that it can occur also in the atmosphere (e.g. China et al., 2015b). We examined both dry and wet particle generation techniques.

(3) **Role of particle size.** To test the role of particle size in the IN process, we compared IN activity of large BC agglomerates with inner pores between aggregates (Fig. 1c) to IN activity of smaller size selected aggregates (Fig. 1b) with similar chemistry and morphology but potentially different pores sizes and numbers.

(4) **The influence of surface oxidation.** The influence of surface oxidation on IN activity of BC is still unclear. Some studies (e.g. Koehler et al., 2009; Gorbunov et al., 2001; Marcolli 2014) suggested that heterogeneous ice nucleation is favored on oxidized hydrophilic soot. However, others (e.g. Whale et al., 2015; Lupi et al., 2014a, 2014b; Biggs et al., 2017) suggested that a lower degree of oxidation leads to enhanced ice nucleation efficiency. Our experiments explore the effect of oxidized surfaces.

(5) **The effect of organic coating.** Previous studies have shown that organic coatings are often present onto the inorganic and soot particles, leading to potentially significant changes of their physicochemical properties (Möhler et al., 2008; Shrivastava et al., 2017; Zhang et al., 2018a). A higher organic carbon content has been observed in laboratory experiments to suppress ice nucleation on soot particles (e.g. Möhler et al., 2005b; Crawford et al., 2011; Kärcher et al., 2007). However, for some types of soot such a suppression of IN activity by coating is insignificant while in others it is notable. On the other hand, some organic acids could enhance IN activity (Zobrist et al., 2006; Wang and Knopf, 2011). The coating material may first fill the pores, thus depending on the type, coating may bring about inhibition or enhancement of IN activity. A series of experiments performed with a range of coatings on BC particles, provides some clarification on the effect of organics on the IN activity of BC particles.

## 2 Experimental Method

### 2.1 Materials Studied

In the present experiments, the ice nucleation properties of the six types of BC particles, listed in Table 1, were studied. The first five of these materials are commercial carbon black, a form of elemental carbon obtained from the incomplete combustion of organics (typically liquid hydrocarbons) under controlled conditions. The first four were supplied by Cabot Corporation, the fifth material, Raven 2500 Ultra (R2500U), manufactured by Birla Carbon was chosen for its relatively large specific surface area. These uniform commercial powders with known physical properties allow a systematic screening of selected particle properties important for ice nucleation in the atmospherically relevant 217-235 K temperature regime.

Submicron soot particles were produced by an inverted flame soot generator (Argonaut Scientific Corp.) through combustion of ethylene ($C_2H_4$). In the present study, we chose to maintain the flame at a low net fuel equivalence ratio of 0.017 to avoid the polydisperse size distribution mode shifting to larger sizes due to agglomeration and also to avoid clogging of tubing by the soot. These particles were collected on a filter and reaerosolized for ice nucleation measurements (see Sect. 2.2.1).

The Braunauer-Emmett-Teller (BET) and Oil Absorption Number (OAN) values shown in Table 1 were provided by the manufacturers. The BET method (Brunauer et al., 1938) measures the specific surface area of materials by determining gas adsorption (usually $N_2$). The BET value is affected by primary particle size. Higher BET is associated with smaller primary particle size, in this case spherules. The OAN is an international standard measurement for characterizing carbon black, obtained by a well-defined test ASTM D2414 (ASTM, 2017) consisting of adding oil to the carbon black sample. This parameter is associated with the degree of branching of the black carbon particle. Higher OAN corresponds to more highly branched particle structures.

The surface chemistry and hydrophilicity can affect the IN activity of a particle (e.g. Koehler et al., 2009). To test this effect, we included the surface oxidized Regal 400R carbon black pigment in this study, which was oxidized by the manufacturer in a post combustion process. In order to test the acidity and solubility of surface groups on BC, we measured the pH of aqueous BC suspensions. We used an ultrasonic homogenizer and a sympHony B10P pH meter

(VWR Scientific). The pH measurements confirmed the surface chemistries indicated by the manufacturers, with the Regal 400R carbon black samples exhibiting an acidic suspension, whereas the other samples all generated near neutral suspensions (Table 1). In situ characterization measurements are described in the following sections.

## 2.2 Experimental Setup

### 2.2.1 Aerosol generation and characterization

A schematic diagram for the experimental apparatus is shown in Fig. 2. The table inserted in the figure provides the code to instrument abbreviations. The BC particles which are in the form of a powder under dry conditions are dispersed into a free-flowing dry nitrogen gas using a novel printed fluidized bed generator with an acronym PRIZE (Roesch et al., 2017). In addition to dry dispersion via the fluidized bed, samples were also atomized from an aqueous suspension with a 3-Jet Collison Nebulizer (BGI) to assess the impact of aerosol generation techniques on IN measurements. The atomized BC particles were dried inline in a diffusion dryer (Topas DDU 570/H) filled with silica gel. Next, to study the effect of size and structure as outlined in Sect. 1, we selected particles of two mobility diameters ($D_m$), 100 and 800 nm.  The size was selected by a Differential Mobility Analyzer (DMA, Brechtel) and counted by a mixing condensation particle counter (MCPC 1710, Brechtel).

In order to study the effect of coating by organic carbon content on IN, BC particles were coated either with stearic acid (SA) (>99 % purity, Aldrich), cis-pinonic acid (98 % purity, Aldrich), or oxalic acid (>99 % purity, Aldrich), similar to the vapor condensation method described by Zhang et al. (2018b).

Stearic acid ($C_{18}H_{36}O_2$) is one of the abundant saturated fatty acids and it is a common constituent of atmospheric particles in urban areas that cook large amounts of meat (Katrib et al., 2005). Stearic-acid coated particles are a gross simplification of atmospheric particles since urban aerosol particles are composed of hundreds if not thousands of organic molecules (e.g. Goldstein and Galbally, 2007). However, these particles could serve as a proxy of the broader class of soot coated with fatty acids.

Humic-like substances are very efficient surfactants. One of the commonly used model surfactants is cis-pinonic acid ($C_{10}H_{16}O_3$). This compound originates from boreal forests, and blooming algae in the oceans yielded from the photochemical oxidation of the evaporated α-pinene in the lower troposphere (Luo and Yu, 2010).

Dicarboxylic acids are another important group of organic compounds identified in the atmospheric aerosols. Their contribution to the total particulate carbon ranges from about 1-3% in the urban and semi-urban areas to values close to or even above 10% in the remote marine environment (Kerminen et al., 2000). Dicarboxylic acids have several different sources, including primary emissions from fossil fuel combustion and biomass burning (Chebbi and Carlier, 1996). Here, we examine coatings with oxalic acid ($C_2H_2O_4$), an abundant dicarboxylic acid in the lower troposphere, comprising a significant fraction of the total diacid mass concentration (Kerminen et al., 2000).

Size-selected BC particles were passed at a flow rate of 1.3 L min$^{-1}$ through a heated, temperature-controlled reservoir that contained the coating substance (Fig. 2). Temperatures during the coating process were adjusted for each organic material, and kept below the homogeneous nucleation point of the coating substance. At temperatures below 200 K, the morphology of the BC aerosol would not change significantly (Bhandari et al., 2017). The phase and the thickness of the coating material was not directly determined in these experiments. At the temperature and relative humidity

conditions of the ice nucleation experiments, coatings consisting of super-cooled aqueous solutions, as well as crystalline or glassy solids, can form (Hearn and Smith, 2005; Knopf, 2018; Murray, 2008; Zobrist et al., 2008).

**2.2.2 BC particle characterization**

The DMA-selected (800 nm) BC particles were characterized for chemical composition by the Particle Analysis by Laser Mass Spectrometry (PALMS) instrument, described in detail by Cziczo et al., (2006) and Zawadowicz et al., (2015). Briefly, particles are collimated in an aerodynamic inlet and pass through two 532 nm Nd:YAG laser beams. The time difference between scattering signals corresponds to particle velocity, which can be converted to vacuum aerodynamic diameter. Particles are then ionized using a 193 nm excimer laser. Either positive or negative ions can be detected. Particle vacuum aerodynamic diameter and chemical composition are measured in situ and in real time at the single-particle level. Due to highly variable ionization efficiencies and matrix effects of common atmospheric materials, single particle mass spectrometry instruments such as PALMS are not considered quantitative. Therefore, hundreds of single particle spectra are acquired to compare relative compositions between similar samples.

In addition, BC particles of 800 nm mobility diameter were collected on 300-mesh copper lacy formvar grids (Ted Pella, Inc.) and analyzed offline in a scanning electron microscope (Hitachi S-4700 FE-SEM). While BC aggregates are often considered highly branched (Fig. 1a), the microscopic images of the collected 800 nm size-selected agglomerates generated by dry dispersion show compact clusters. Based on analysis of 40-50 particles per BC particle type, approximately 50% of the agglomerates were classified as highly compact, near spherical (Figs. 1c, A2); the remaining were classified as compact, prolate agglomerates. For analysis purposes, we assume these agglomerates to be highly compact spherical particles. This extrapolation to the whole agglomerate population will induce potential statistical error and will define the upper limit of the effective surface area calculations (Supplement Sect. S1). The surface area is then used together with the activated fraction to calculate the density of the active sites (Vali et al., 2015). The active site density, $n_s(T)$, can be defined as

$$n_s = \frac{A_f(T) \cdot L_f}{S_{eff}} \; [m^{-2}], \tag{1}$$

where $A_f(T)$ is the activated fraction at a given temperature ($A_f(T) < 0.1$), $L_f$ is a correction factor obtained in flow calibrations, and $S_{eff}$ is the effective surface area, which is calculated from the sum of the areas of spherules that form the outer shells of a BC particle of a given geometric volume. The mean diameter of the circular 2-D shape was used to calculate the volume of a geometric sphere having the same diameter in all axes. We assume that the geometric volume of the spheroidal agglomerates is comparable with the volume derived from the selected mobility diameter. This assumption may introduce an uncertainty factor of 3 due to the uncertainty in the number of outer layers that contribute to the total surface area (Supplement Sect. S1). The active site density analysis of the 800 nm particles enables us to study the effect of complex agglomerate structures on the IN activity (see Sect. 3).

**2.3 Ice nucleation measurements**

The ice nucleation measurement technique utilized here (blue dashed frame, Fig. 2), including the operation of the SPectrometer for Ice Nuclei (SPIN, Droplet measurement Technologies), calibration of the laminar flow fraction, and the minimization of uncertainty of IN measurements is described in detail elsewhere (Garimella et al., 2016, 2017).

Here, we describe only briefly the experimental steps. The SPIN instrument is a continuous flow diffusion chamber consisting of two flat, parallel aluminum plates, cooled independently to create a temperature gradient. The plates are covered with a layer of ice and due to the difference in their temperature, a linear gradient in water vapor partial pressure and temperature is set up between the two plates. Because of the nonlinear relationship between saturation vapor pressure and temperature, supersaturation with respect to ice, in water-subsaturated conditions up to and including water-saturation, is achieved along the center of the chamber, allowing for ice nucleation..

The laminar flow in the chamber can reach temperatures as low as 213 K, with a fixed particle residence time of approximately 10 s inside the chamber, at a sample flow rate of approximately 1 L min$^{-1}$ such that nucleated ice crystals are able to grow to sizes of several micrometers in diameter. In each experiment, the average temperature of the laminar flow was held constant while the relative humidity was slowly increased from subsaturated to supersaturated conditions with respect to ice and constantly subsaturated conditions with respect to water. We investigated BC heterogeneous ice nucleation activity at temperatures of 217 to 235 K and relative humidity with respect to ice (RH$_i$) between 100 and 150 %, typical to cirrus clouds (Krämer et al., 2016). The SPIN was operated in a supersaturation scanning mode, from low RH$_i$ to high RH$_i$ at each fixed temperature.

Aerosol particles fed into the SPIN nucleation chamber in a lamina sample flow are nominally constrained to the lamina with a sheath flow of about 9.0 L min$^{-1}$. However, turbulence at the inlet causes a fraction of particles to spread outside the centerline, decreasing the RH$_i$ they are exposed to. As these particles are less likely to activate as IN, a correction factor, $L_f$ in Eq. (1), 5.8 is applied to fractional activation data obtained from the machine learning algorithm. Depending on experimental conditions, correction factors between 1.86 and 7.96 have been previously reported (Garimella et al., 2017; Wolf et al., 2019).

Nucleated ice crystals are detected in the SPIN by an optical particle counter (OPC, Droplet Measurement Technologies) at the chamber outlet. The OPC measures particles from 500 nm to 15 µm diameter and has the ability to discriminate the phase of the particle by the 3 detected polarization components in 2 scattering angles using machine learning algorithms developed by Garimella et al. (2016). Ice formation onset is reported as the RH$_i$ at which ice particles are identified by the machine learning algorithm.

Ice onset point is defined as the combination of supersaturation with respect to ice and temperature at which the activated fraction threshold is reached. The activated fraction is defined as the number concentration of aerosol that are activated to form ice crystals detected by the OPC divided by the total number concentration, counted by MCPC (Fig. 2). This activation threshold does not have a universally agreed value (Kanji et al., 2017). In our study, we have set it at 1 %.

## 3 Results and discussion

### 3.1 Ice onset

The ice onset points for 800 nm particles, together with experimental error bars, which represent the variability of the laminar conditions based on CFD simulations by Kulkarni and Kok (2012), are presented in Fig. 3. This figure shows a plot of water vapor supersaturation ratio over ice versus temperature at which (1 %) ice onset occurs. This time

dependent approach is based on kinetics in the ~10 s period of particle passing through the chamber and the set supersaturation conditions. The most obvious aspect of these results is the bifurcation in the ice onset measurements with decreasing temperatures. The Regal 400R results remain indistinguishable from the homogeneous ice nucleation line, whereas the other BC particle types all exhibit freezing abilities below Koop line (i.e. the homogeneous freezing

and water saturation line in Fig. 3). The relationship between the supersaturation with respect to ice at the ice nucleation onset, $SS_i$, and temperature is known as "isoline" (see Fig. 19 of Hoose and Möhler, 2012). More efficient INP have lower isolines.

Measurements of IN activity for non-oxidized commercial carbon-black particles show the same trend in IN activity of soot collected from ethylene combustion. Both types demonstrated a temperature dependence of ice onset,

increasing ice onset point with increasing temperature in the range 217 - 235 K, similar to some of the earlier observations of soot (e.g., Ullrich et al., 2017; Bond et al., 2013; Hoose and Möhler, 2012). These previous reports suggested that PCF mechanisms based on water-condensation prior to ice nucleation, and not classical deposition nucleation mechanisms (i.e., vapor to solid), may account for these observations.

Heterogeneous ice nucleation was observed for the non-oxidized BC proxies and soot with identical mobility diameter

of 800 nm. As is evident, the data differ for each sample. To understand these differences and to simplify the discussion, we will address each issue outlined in Sect. 1, separately.

We extracted and replotted salient data from Fig. 3 related to the specific issues and display them in Figs. 4 to 7. An additional set of measurements was obtained for the 100 nm particles. Those results are presented in Fig. 6.

**(1) Effect of particle morphology:** The most notable feature in Figs. 3, 4 is the gradient between the data sets for

BC particles R2500U (red squares) and Regal 330R (yellow squares), including the BC types in between. The displayed onset supersaturation difference shows that the IN activity of R2500U is higher than that of Regal 330R for the majority of runs in the temperature range 228 – 233 K. This is most likely due to the difference in the BET parameters for the two species; for R2500U BET = 270 $m^2\,g^{-1}$ and for Regal 330R BET = 90 $m^2\,g^{-1}$. As stated in Sect. 2.1, higher BET values correlate with higher surface areas, typically implying smaller primary particle size. In turn

smaller spherules tend to lead to higher branching within the BC material and hence a greater number of pores. Chughtai et al. (1999) showed that larger surface area of carbonaceous material determines the adsorption capacity of water molecules at $RH_w$ = 83 %. Other studies demonstrated ice formation in hydrophobic confinements (e.g. Bampoulis et al., 2016; Bi et al., 2017; Zhu et al., 2018). Therefore, a higher number of smaller pores in BC particles may facilitate condensation of water in sub-saturated conditions and subsequent freezing in ice super-saturated

conditions. In accord with the PCF mechanism, higher number of pores is associated with greater IN activity as is consistent in our results.

A hint for the subtle influence of aggregate branching on IN activity can be found in the comparison of Monarch 880 (grey squares), Monarch 900 (blue squares), and R2500U (red squares) (Fig. 4), all have nearly the same specific surface area (Table 1), while the OAN number, which is a measure of branching, is higher for Monarch 880 by 60 %.

Higher branching may contribute to less compact structure and larger diameter pores that display a weaker inverse Kelvin effect and therefore can only fill with water at higher $RH_w$. Therefore, in accord with the PCF mechanism, one

would expect Monarch 880 to display lower IN activity in comparison to Monarch 900 and R2500U, as is observed (Figs. 3, 4).

The dimensions of the pores in the BC agglomerate, that is, the density of the BC agglomerate, in this case, is primarily governed by the entangled contribution of the single spherule size and the degree of branching of the enclosed aggregate. The ratio of the vacuum aerodynamic diameter, measured by the PALMS instrument for each air dispersed BC sample, and the mobility diameter gives the effective density of the agglomerate (DeCarlo et al., 2004). For a constant selected mobility diameter of 800 nm, we observed variability in the effective density (Table 1). The aerodynamic diameter of the agglomerates is related to their mass, and the shape factor (DeCarlo et al., 2004; Jayne et al., 2000). A similar round shape of agglomerates was observed for numerous BC samples in the electron microscope (Fig. A2). The variability in IN activity of particles that have the same mobility diameter and chemical composition (Fig. 4) can likely be explained by the variability in particle effective density (i.e. pore number concentration and dimensions). The variability in IN activity for temperatures greater than ~230 K (Fig. 4), where the measured critical supersaturations at IN onset increases with increasing temperature, appears to be inversely correlated with their measured effective densities (Table 1). This same relationship is observed in the temperature at which the measured critical supersaturations for the different BC particle types intersect with the homogeneous freezing line. These observations suggest that for temperatures at or greater than 230 K, more effective BC INPs have higher effective densities (i.e., are more compact particles), implying that IN active pore sizes may be related to particle effective densities.

Another method for physical characterization is the detection of the shift in polarization in the light scattered from 800 nm BC particles in the OPC. A shift in polarization of the linearly polarized incident beam will occur if the particle is optically anisotropic (having aspherical shape, branches, roughness, or variations in internal structure). This polarization shift is used in classification of particles by their 'optical shape' (e.g. Garimella et al. 2016; Nichman et al., 2016; Kobayashi et al., 2014; ; Glen and Brooks, 2013). Even spheroidal shaped particles produce a unique shape-specific phase function distinctly different from those produced by other spheroidal particles (Mischenko et al., 1997). Thus parallel and perpendicular polarization measurements can be used to differentiate between particles of the same diameter. Francis et al. (2011) showed that both, agglomerate diameter and spherule diameter of soot, affect the polarization within specific boundaries. In our study, the optical sphericity could shed light on the BC particle shape (i.e. round and compact versus branched and lacy) and its influence on the IN mechanism in BC agglomerates. High optical sphericity of a particle is determined by a low polarization shift in the light scattered from the particle. OPC data median values of single-particle optical measurements for each BC sample are listed in Table 1. In our experiments, the most active INP (e.g. R2500U, Monarch 900) showed lower polarization shift signatures, which suggests a more optically spherical shape.

**(2) Effect of particle generation:** In order to test the effect of aerosol generation technique on IN measurements, we measured the IN activity of BC atomized from a liquid suspension and dried in a diffusion dryer in comparison with IN activity of the same BC dry-aerosolized. The results showed no visible sign of compaction effects on IN activity of 800 nm BC particles (Fig. 5). The dry-aerosolized round compact shapes observed in Fig. A2 may explain the lack of further compaction during the atomization process of 800 nm BC aerosol. Further qualitative support for the

hypothesis of initial compactness of the particles was provided by the low values measured in the OPC (Table 1), which are associated with the sphericity of the particles. The tolerance of these compact BC particles towards further compaction is a very unique feature of BC as compared to other insoluble compositions, e.g., dust surrogates (see Sullivan et al., 2010).

**(3) Role of particle size:** To test the extremes of the aerosol diameters used in previous studies (e.g. Friedman et al. (2011); Crawford et al. (2011)), we selected 100 nm and 800 nm BC particles. The results of experiments shown in Fig. 6 suggest that there is an influence of the BC particle size on its IN activity. The smaller mobility diameters of 100 nm, which are likely to be single aggregates (Fig. 1b), do not nucleate as readily as the larger agglomerates of 800 nm of the same composition, spherule size, and branching. The BC particles are hydrophobic and non-crystalline with particle size-independent packing density as shown in Zangmeister et al. (2014), therefore, according to the stochastic approach, an increase of the surface area should not affect their IN activity, however according to singular approach, the probability of active sites will be higher for larger particles, increasing their probability to nucleate ice. Hence, the enhancement of the IN activity in each 800 nm BC sample shown in Fig. 6 suggests a unique structural change as the mobility diameter is increased. It is plausible that these particles nucleate via the PCF mechanism by capillary condensation in empty spaces between soot aggregates due to the inverse Kelvin effect. The number of aggregates that form the agglomerate define the number and the dimensions of pores that act as nucleation sites, which in the extreme case of single aggregate (i.e. without suitable pores) nucleate at or above the homogeneous freezing threshold (e.g. grey circles in Fig. 6). The error bars of Monarch 880 data points partially overlap in some of the 100 and 800 nm runs. Nonetheless, several runs don't have an overlap in uncertainty at the same temperature and the trend of lower IN activity, in 100 nm particles, repeats itself. Similarly, the IN activity of 100 nm ethylene flame soot is reduced in comparison to the 800 nm soot. Despite the reduction in activity, the 100 nm soot has nucleated ice below homogeneous freezing conditions. It is possible that a bias introduced by multiply charged particles of flame soot's broad size distribution, passed at the same DMA voltage, maintained the high IN activity of 100 nm mobility diameter soot.

**(4) The influence of surface oxidation:** In Fig. 7, the oxidized sample of Regal 400R (green) is compared to a similar non-oxidized sample of Regal 330R (yellow), and ethylene soot. We measured the pH values (Table 1) obtained for aqueous suspensions of the BC samples. Ethylene soot has a slightly lower pH, as expected for a combustion generated soot. However, Regal 400R particle type is the only BC sample in this study that had both an oxidized surface and generated an acidic aqueous suspension. The surface acidity is likely due to surface-bound oxygen-containing functional groups; both the surface functional groups and surface porosity have been shown to influence the amount and the energetics of the water adsorbed to the carbon surfaces (Salame and Bandosz, 1999; Marsh and Rodriguez-Reinoso, 2006). Currently, we have no way of discriminating between the effects of surface oxidation and surface acidity on the observed IN activity of Regal 400R. In addition to pH values, we used the PALMS instrument to confirm the oxidation state of dry particles. Negative and positive ion spectra of about 1000 particles were collected for each BC sample. The oxygen negative ion peaks were then plotted against carbon negative ion peaks and color-coded for each BC sample (Fig. A1). Regal 400R particles cluster demonstrates noticeably higher peaks of oxygen in comparison

to other BC samples. This oxidized sample froze homogeneously even at temperatures as low as 219 K while the non-oxidized sample showed ice nucleation activity, well below homogeneous freezing conditions. However, small amounts of surface oxidation on soot does not appear to significantly affect IN activity. These observations are initially counterintuitive due to the presumed hydrophilicity of the oxidized sample, however the ubiquity of oxygenated surface groups on BC surfaces does not mean that soot particles will appear hydrophilic on a macroscopic scale or nucleate ice (Friedman et al. 2011). For example, fresh, oxidized soot particles do not generally activate as cloud-condensation nuclei (CCN) under atmospherically relevant conditions (Corbin et al., 2015). Moreover, molecular dynamics calculations show that hydrophilicity is not a sufficient condition for IN (Lupi et al., 2014a, 2014b). In fact, Biggs et al. (2017) reported an increase in the ice nucleation activity due to a decrease in hydrophilicity. The freezing behavior of water confined in pores of hydrophilic silica or hydrophobic carbon was similar, suggesting that pore hydrophobicity may play a limited role in PCF-type freezing (Morishige, 2018). Häusler et al. (2018) suggested that agglomeration may lead to a favorable positioning of the functional sites and therefore to an increase in the IN activity, even though a decrease in the surface area occurs. However, it was found that the increased proportion of oxygen increases the hydrophilicity of graphene, reduces agglomeration and hence increases the surface area and reduces the number of pores (Häusler et al., 2018). Thus, it is possible that by oxidizing the surface of Regal 400R particles, the micro structure of the particles changed (Cabrera-Sunfelix & Darling 2007), reducing the number of PCF active pores. Fletcher (1959) noted that a highly polar surfaces could raise the free energy of formation of ice embryos, reducing the efficiency of heterogeneous ice formation, providing another potential explanation. Finally, if the surface is highly oxidized, the Regal 400R particles may condense monolayers of water more readily than non-oxidized Regal 330R, affecting the ice formation. All of these potential explanations are also consistent with the surprisingly high effective density of Regal 400R sample.

The combined contribution of single spherule size, particle size, surface oxidation, and morphology to IN activity affects the spatial arrangement and thus the adjacent angles in the pores that dictate the formation, and perhaps the type, of the ice lattice (Bi et al., 2017; Zhu et al., 2018). However, further screening of BC samples accompanied by thorough characterization (e.g. BET, Atomic Force Microscopy, contact angle, cold-stage experiments) is needed to confirm these findings.    .

 **(5) The effect of organic coating:** Surface oxidation is not the only process altering the IN activity. Crawford et al. (2011) showed that alteration of the organic carbon content from minimum (5 %) to medium (30 %) results in a clear transition between heterogeneous and homogeneous freezing mechanism, respectively. On the other hand, some organic acids enhance IN activity (Zobrist et al., 2006; Wang and Knopf, 2011). Coating material may fill the pores, and the extent of inhibition or enhancement of IN activity may imply the prevalent ice formation mechanism for BC agglomerates. The cis-pinonic and stearic acid when atomized, nucleate ice homogeneously (Fig. S3). When used as coatings they decreased the IN activity of R2500U (Fig. 8b, c). The pure atomized oxalic acid on the other hand, nucleates ice heterogeneously at ice supersaturation as low as 10 %. When used as a coating material, it has increased the IN activity of a homogeneously nucleating BC sample, Regal 400R, to ~30 % superstation at ice onset (Fig. 8a).

### 3.2 Active site density

As we mentioned in Sect. 1, two approaches to explain the IN data are commonly used: a stochastic description based on classical nucleation theory and a deterministic or singular description (Knopf et al, 2018). The time-independent, singular approach is often used in models where a surface density of sites active on a particle surface (Eq. 1) can initiate ice nucleation at a given temperature, assuming that one site gives rise to a single ice crystal.

The least square fitted isolines of the calculated active site densities (Eq. 1) for 800 nm particles of Monarch 900, Ethylene combustion soot, and R2500U, the most active BC proxies, are presented in Fig. 9 and lie in the range 0.6 - $1.2 \cdot 10^{10}$ m$^{-2}$. Shown in the figure as black dashed lines, are active site density isolines ($10^{10}$ m$^{-2}$, $10^{11}$ m$^{-2}$ and $10^{12}$ m$^{-2}$) derived from empirical parametrization of ice nucleation on soot from five sources by Ullrich et al. (2017). Isolines of non-oxidized particles, as measured in this study, lie mostly within the boundaries confined by the $10^{10}$ and $10^{11}$ m$^{-2}$ active site density isolines from Ullrich et al. (2017). The uncertainties in calculations of the effective surface area together with activated fraction uncertainties are propagated to a total uncertainty of less than an order of magnitude, which may partially explain the gap between the isolines in both studies. Other plausible explanations may include differences in composition, organic carbon content, and the spread of data points in Ullrich et al. (2017). For particles of 100 nm in diameter and the same IN onset threshold (1 %), the effective surface area will be approximately 2 orders of magnitude lower. Therefore, for these particles the active site density isolines will have to be two orders of magnitude higher, closer to the homogeneous freezing line.

In Fig. 10, the ice onset points of 800 nm BC proxies tested in our study complement the data reproduced from Kanji et al. (2017) for the lower temperature regime in water subsaturated conditions. This narrow spread of data points indicate a clear increase in IN activity as a function of decreasing temperature. As we explained earlier, the bigger diameter in our study may be the main explanation for higher IN activity in comparison to data from earlier studies.

### 4 Atmospheric implications

The carbon black, an industrial powder product, is potentially atmospherically relevant both directly, as a pollution component, especially when generated in a "channel process", mostly in developing countries, utilizing natural gas impingement on iron channels to produce carbon black (Dannenberg et al., 2000; Hardman, 2017), and indirectly as a proxy for atmospheric particle as was recently studied by Dalirian et al., 2018 and others. Canagaratna et al. (2015) have shown that Regal black and flame soot appear very similar, at least from the perspective of the mass spectrometry. However, in the ambient setting, BC particles can vary significantly in terms of their physical and chemical properties, and are usually mixed with other pollutants present in the atmosphere. The widespread in IN activity of BC obscures understanding of the radiative properties of clouds and Earth's climate. Our findings show that large agglomerates, such as observed in wild fires (Chakrabarty et al., 2014), could be a potential source of efficient heterogeneous INP via the PCF mechanism in cirrus cloud conditions in the troposphere. While the concentration of such large particles is usually low, limiting their detection, these efficient INP may contribute to the warming effect, as was shown by Bond et al. (2013) for heterogeneous nucleation of BC. Our study explained some of the main factors that affect the

high variability in IN activity of different BC, which can shift between heterogeneous and homogeneous ice nucleation. For comparable oxidized and non-oxidized particles, lower activity was observed in the oxidized particles. Hence, the singular, time independent, parameterization approach in models should take into account the oxidation state and coating of the BC particles. Organic coatings not only cover the outer layers of BC aggregates but may also fill the internal pores among primary spherules and have the ability to either inhibit or enhance IN activity.

We showed that our results of BC ice formation below homogeneous freezing conditions are consistent with the PCF mechanism. However, these findings seem to contradict some field observations, where no clear evidence of heterogeneous IN activity of BC was found (see Introduction Section). Such contradictory observations between laboratory and field measurements can partly be explained by ice multiplication processes (e.g. Ladino et al., 2017), as well as, decrease of IN activity by surface coatings (Friedman et al., 2011), instrumental limitations (Cziczo et al., 2014), and scarcity of data from field measurements. Another possible reason is the surface oxidation of the particle as part of the aging process along the timeline of its trajectory in the atmosphere. Several studies, including this one, have shown that oxidation on carbonaceous surfaces can, in some cases, reduce the efficiency of ice nucleation. However, if PCF is to occur in atmospheric BC particles, less studied mechanisms of ice multiplication, resembling supercooled droplet splintering upon freezing (Wildeman et al., 2017), may occur during pore condensation and freezing, inside the confined geometry (Vlahou and Worster, 2010; Kyakuno et al., 2010; Kyakuno et al., 2016). This in turn could be another plausible explanation to the scarcity of BC in ice residuals collected in field measurements and the reason for the underestimation of BC IN activity and its global IN importance.

## 5 Conclusions

In this study, we systematically examined the ice nucleating activity of well-characterized commercial carbon black samples and soot generated in an inverted diffusion flame. A commercial continuous flow diffusion chamber (SPIN) was used to simulate the temperature and humidity conditions of the in situ type cirrus clouds (Krämer et al., 2016). Our results complement the ongoing research of IN activity of BC aerosol in atmospherically relevant conditions. The majority of BC samples tested here showed IN activity at low supersaturation over ice with a strong dependence on the temperature, in agreement with previous reported results. Our data suggests that the main IN mechanism in the majority of the particles tested is consistent with pore condensation and freezing, which occurs in empty spaces between the aggregates. Our main observations are listed below:

- Differences in the morphology of the agglomerate corresponded with differences in IN activity. Particles of the same diameter, with higher specific surface area and lower branching showed the highest IN activity.
- Aerosol generation techniques (i.e. dry versus wet-dried), and compaction, previously reported for ~200 nm BC, did not seem to have a significant effect on the IN activity of atomized 800 nm agglomerates.
- While comparing particle size (i.e. agglomerate versus aggregate) of the same BC sample, which has the same IN efficiency, we observed a lower IN activity at 100 nm mobility diameter versus 800 nm.
- Oxidized particles nucleated homogeneously for all tested temperatures.

- Organic surface coatings demonstrated the capability of both enhancing and inhibiting the IN activity on BC proxies.

Our study indicates that ice nucleation activity of commercial carbon black can take place well below homogeneous freezing conditions. One can select a well-defined morphology of compact, non-oxidized agglomerates with high BET and low OAN values, which correspond to smaller spherules and low branching, respectively. Such a material will allow controlled conditions for more efficient ice nucleation. In future studies, IN activity enhancement with size should be tested in more detail, in the range 100 – 800 nm (e.g. Mahrt et al. (2018)). This enhancement in non-oxidized, compact agglomerates of commercial carbon black should be similarly tested with other compounds of comparable morphology to understand the impact of the PCF mechanism on atmospheric processes and for characterization of IN at low temperatures of the cold cirrus temperature regime.

## 6 Appendix A.

## 7 Author contribution

LN designed the experiments and LN, MW, YZ carried them out. PD, TO, DW, DC, CM supervised and administrated the project, acquired funding, and provided resources and facilities to conduct the experiments. MW processed the ice nucleation data. JB processed the SEM data. LN, PD, TO prepared the manuscript with contributions from all co-authors.

## 8 Acknowledgments

This work was supported by Department of Energy (DOE) award #DE-SC0011935, National Science Foundation (NSF) Division of Chemistry under Grants No. 1506768, No. 1507673, and No. 1507642, NSF Division of Atmospheric and Geosciences Grant No. 1524731, and the Boston College undergraduate research fund. Y. Zhang was supported by a NSF Postdoctoral Fellowship (AGS-1524731) and the National Institute of Health (NIH) training grant. The authors would like to thank Dr. Michael Roech for providing the PRIZE aerosol generator, Dr. Maria Zawadowicz for PALMS guidance, Prof. Kenneth Metz for providing the pH meter, Cabot Corporation and Birla Chemicals for providing the carbon black samples.

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

**Table 1.** Aerosol materials with selected properties. BET, OAN, and surface type information is provided by the manufacturer and pertain to bulk properties. The data in the last four columns were collected in our laboratory; pH was measured for the bulk suspensions; effective density ($\rho_{eff}$), O:C ratio, and OPC data were collected for dry dispersed BC particles.

| Sample | [a]BET [m² g⁻¹] (manufacturer) | [b]OAN [ml/100 g] (manufacturer) | Surface type (manufacturer) | [c]pH | [d]$\rho_{eff}$ [g cm⁻³] | PALMS [e]median O:C ratio (Fig. A1) | [f]OPC Log(S1), Log(P1) |
|---|---|---|---|---|---|---|---|
| Regal 330R | 90 | 65 | Non-oxidized | 6.9 | 0.79 | 0.018 | 2.4, 3.3 |
| Regal 400R | 90 | 70 | Oxidized | 4.9 | 1.24 | 0.057 | 2.3, 3.2 |
| Monarch 880 | 240 | 110 | Non-oxidized | 6.7 | 0.76 | 0.015 | 1.4, 2.9 |
| Monarch 900 | 240 | 70 | Non-oxidized | 6.8 | 0.84 | 0.024 | 1.4, 2.9 |
| Raven 2500 Ultra | 270 | 67 | ---------- | 7.0 | 1.12 | 0.036 | 1.4, 2.8 |
| Ethylene combustion soot | ---------- | ----------- | ---------- | 6.4 | 0.59 | 0.037 | 2.0, 3.1 |

[a]BET (Brunauer–Emmett–Teller) method measures the surface adsorption area to mass ratio of the bulk powder.

[b]OAN (Oil Absorption Number) is proportional to the absorbed oil volume to mass ratio of the bulk powder.

[c]pH measured for bulk suspension of BC materials with measurement precision of 0.1.

[d]$\rho_{eff}$ is the effective density calculated from the ratio of the vacuum aerodynamic diameter ($D_a$) measured by the PALMS (Particle-Analysis-by-Laser-Mass-Spectrometry) instrument and the constant mobility diameter ($D_m$) of 800 nm, multiplied by the standard density of 1 g cm⁻³.

[e]Median O:C ratio measured by PALMS instrument for selected mobility diameter ($D_m$) of 800 nm. On average, up to 50 % deviations from the mean values were observed.

[f]OPC Optical Particle Counter. Logarithmic values of parallel (S1) and perpendicular (P1) polarization, measured by the OPC for size selected 800 nm mobility samples, are shown in the last column.

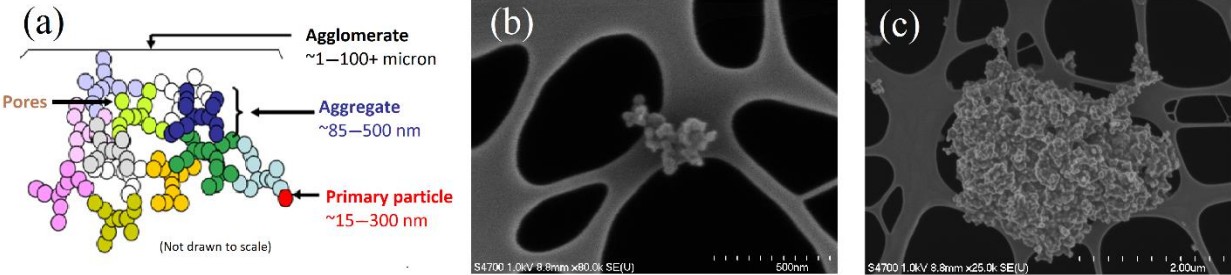

**Figure 1. (a) Illustration of agglomerate, aggregate, and spherule definitions, reproduced with permission from Long et al. (2013). Electron microscope images of (b) Soot aggregate on a substrate, and (c) soot agglomerate. In the images, underlying the soot, is the substrate, on which the soot is collected.**

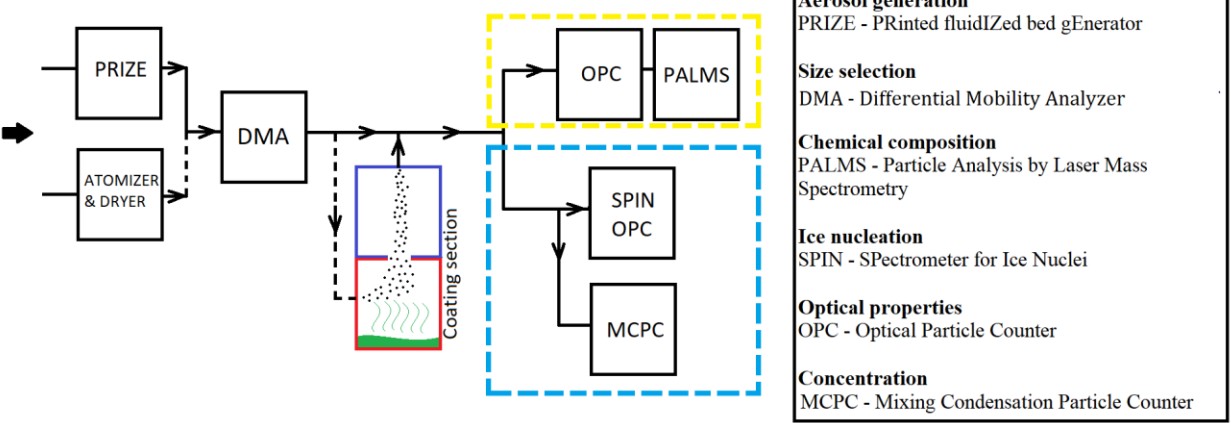

**Figure 2. Simplified diagram of the apparatus showing aerosol generation, coating, characterization (yellow dashed frame), and ice nucleation measurements (blue dashed frame). The red frame is the flask containing the organic acid, which was heated to a temperature slightly lower than the onset of homogeneous particle nucleation of organic compounds. The blue frame is a chilled condenser (-20 °C) which promotes condensation of organic acids on BC.**

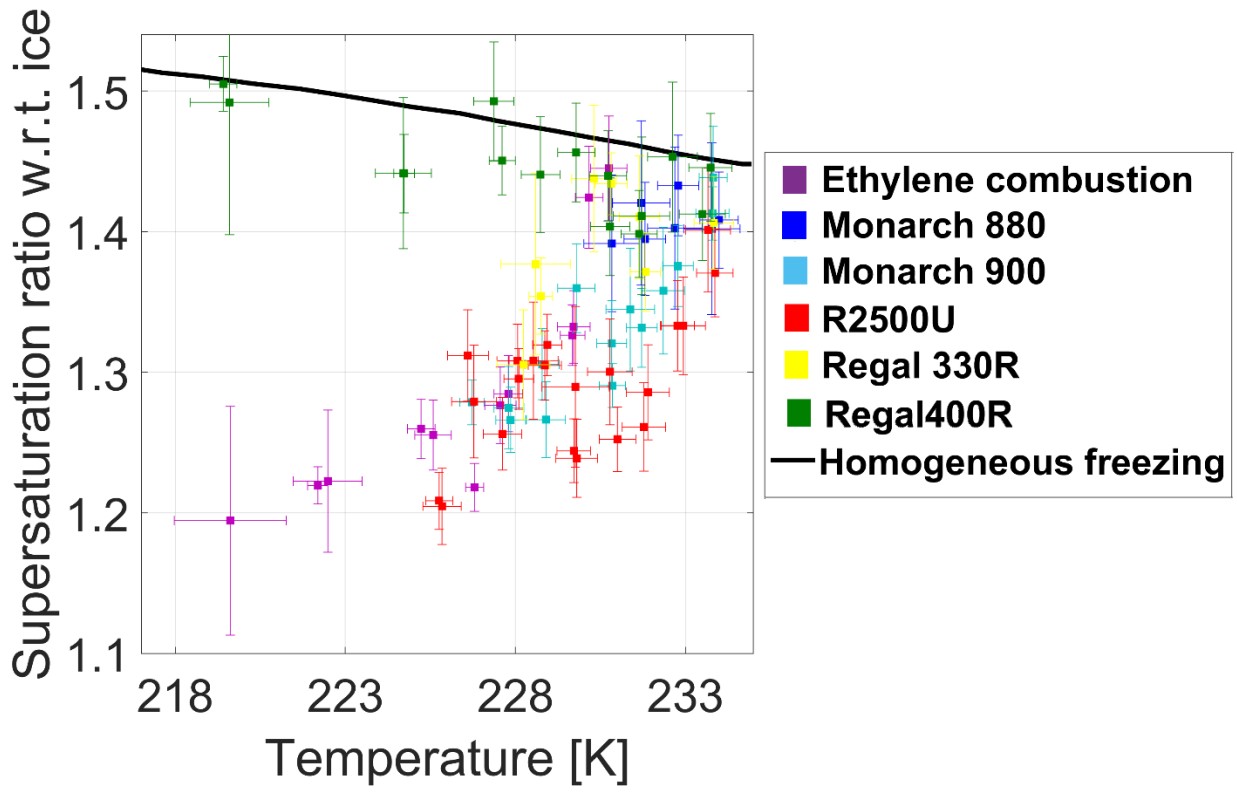

Figure 3. Ice nucleation onset conditions defined as 1 % of the total aerosol to nucleate ice for 800 nm BC particles. Solid black line is the homogeneous freezing threshold (Koop et al., 2000).

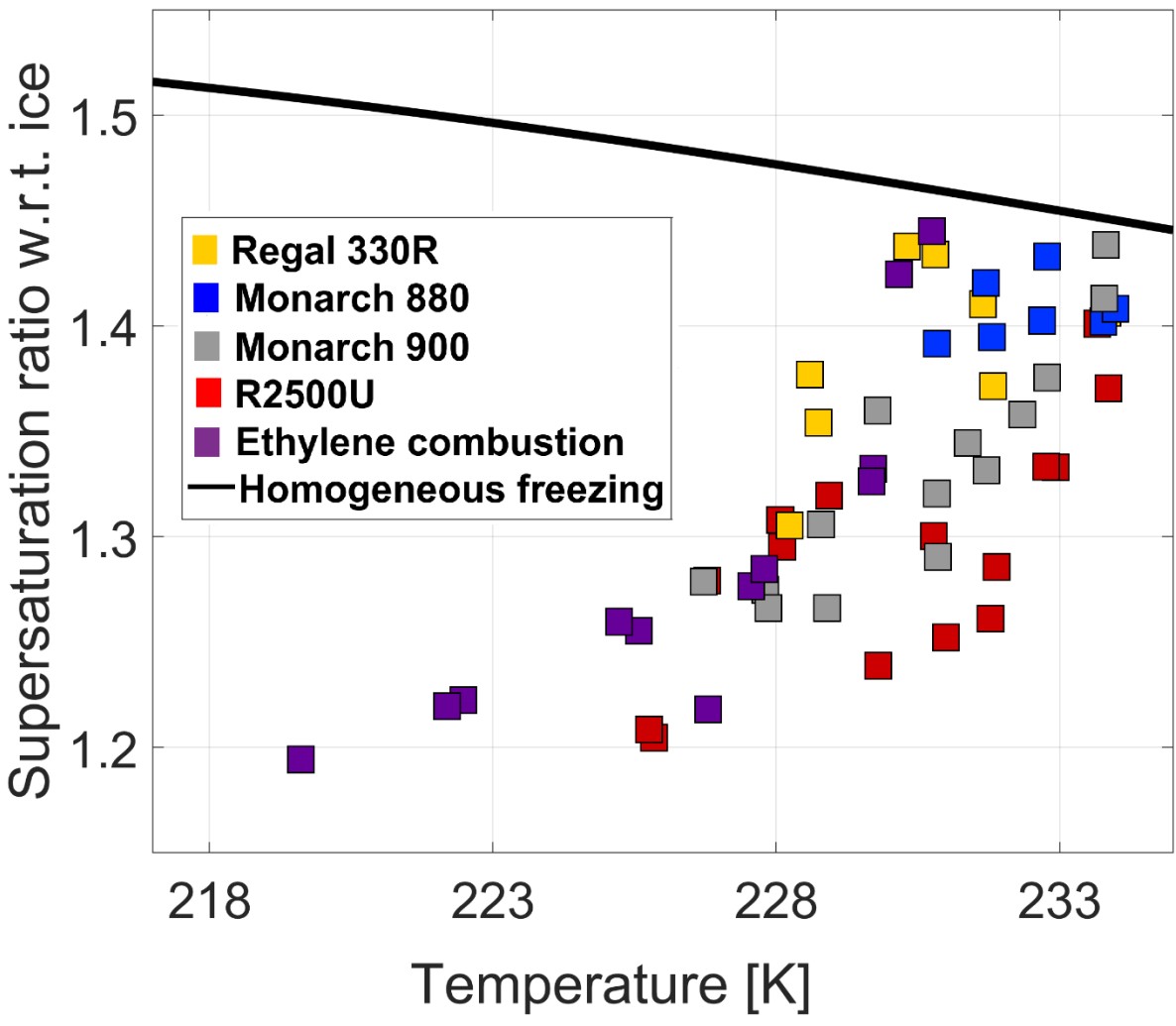

**Figure 4. Morphological effect on ice activity of 800 nm BC particles. The black solid line is the homogeneous freezing threshold.**

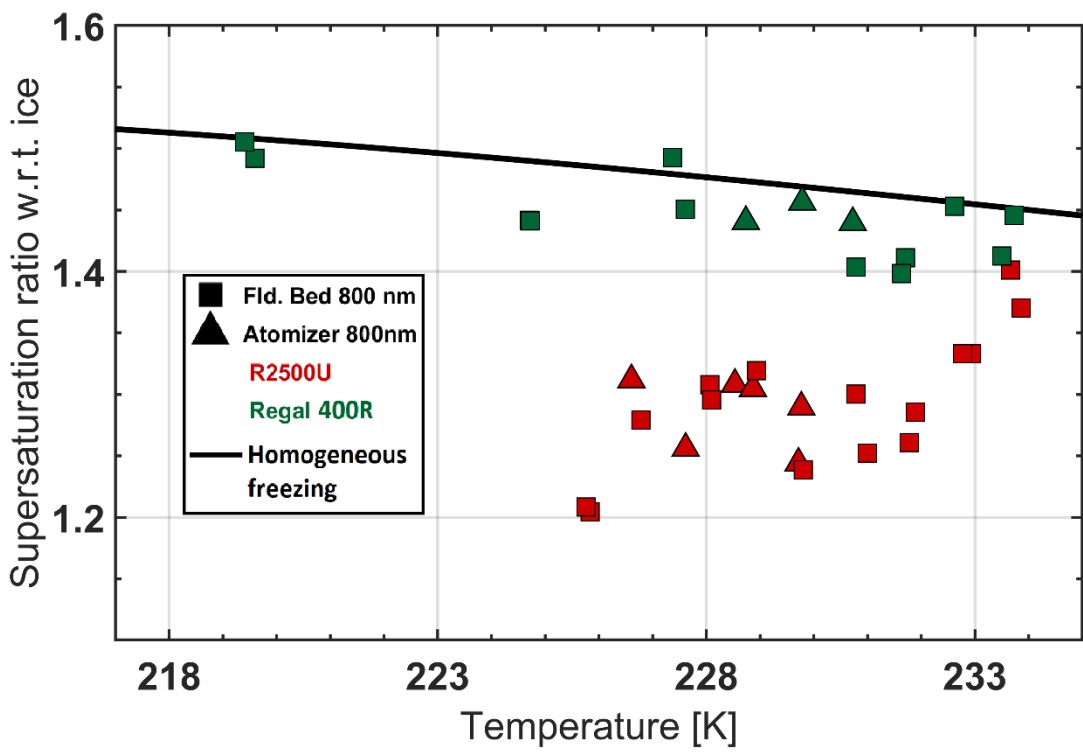

**Figure 5. Aerosol generation technique effect on IN activity. The black solid line is the homogeneous freezing threshold.**

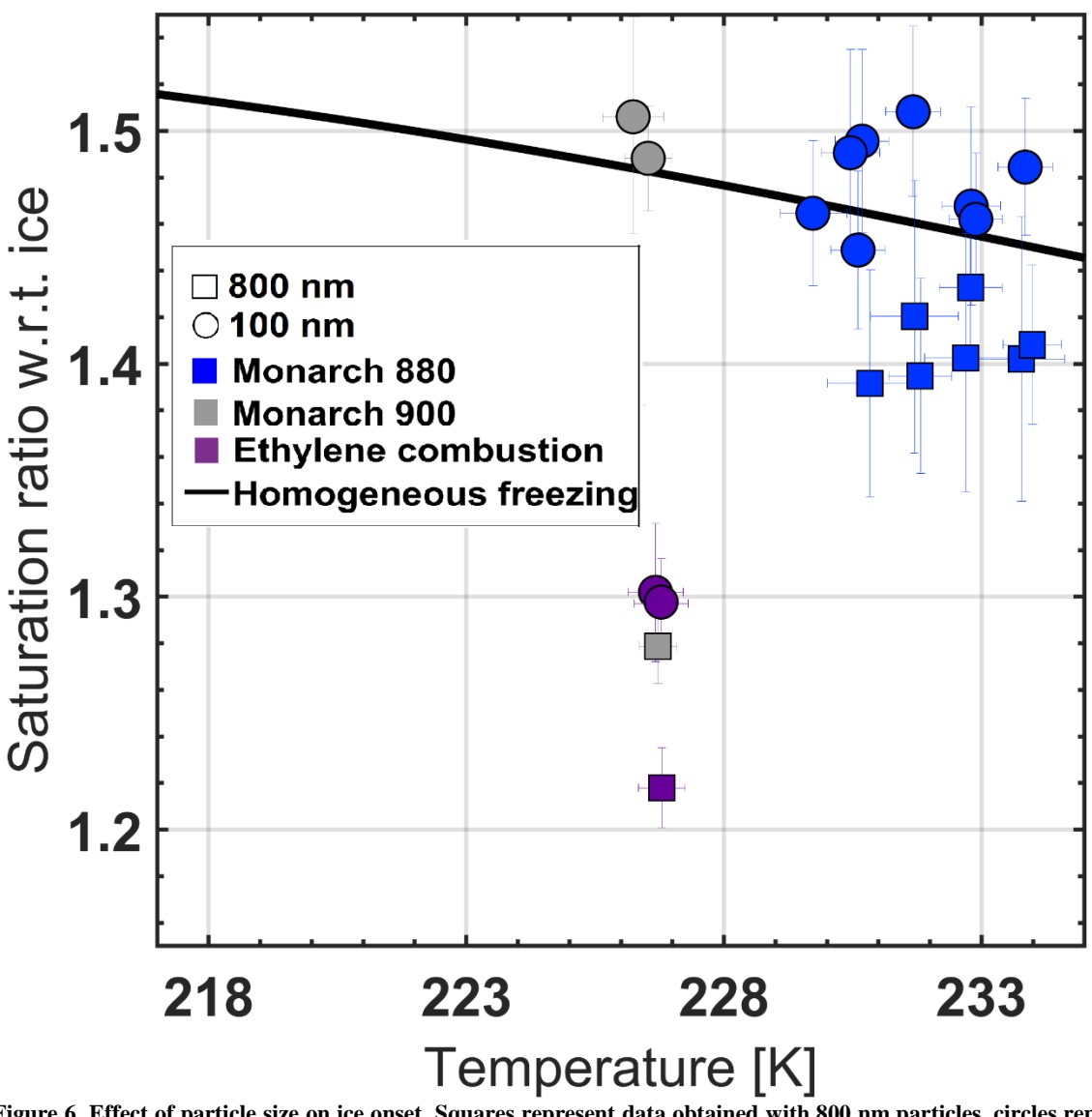

**Figure 6. Effect of particle size on ice onset. Squares represent data obtained with 800 nm particles, circles represent 100 nm particle data. The black solid line is the homogeneous freezing threshold.**

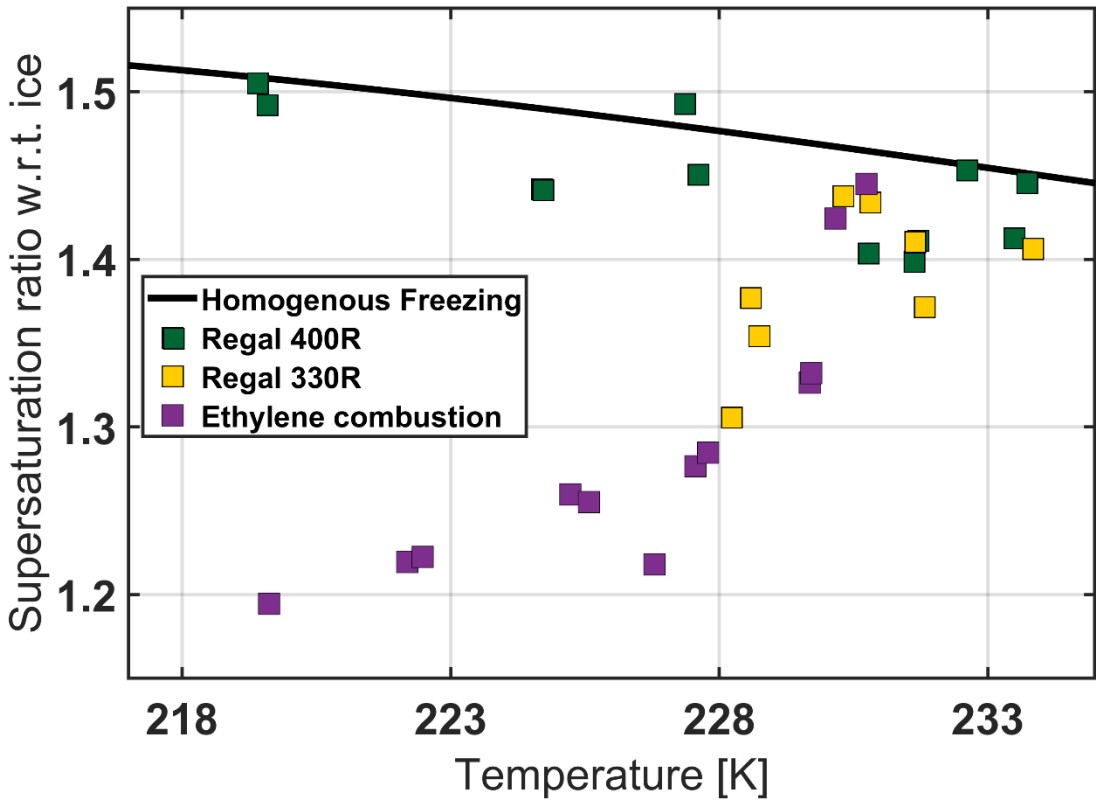

**Figure 7. Influence of surface oxidation on IN activity. Comparison of 2 BC compounds with identical BET and comparable OAN values (Table 1), mainly differing in the degree of surface oxidation (Fig. A1) and a sample of ethylene combustion soot with intermediate degree of oxidation. The black solid line is the homogeneous freezing threshold.**

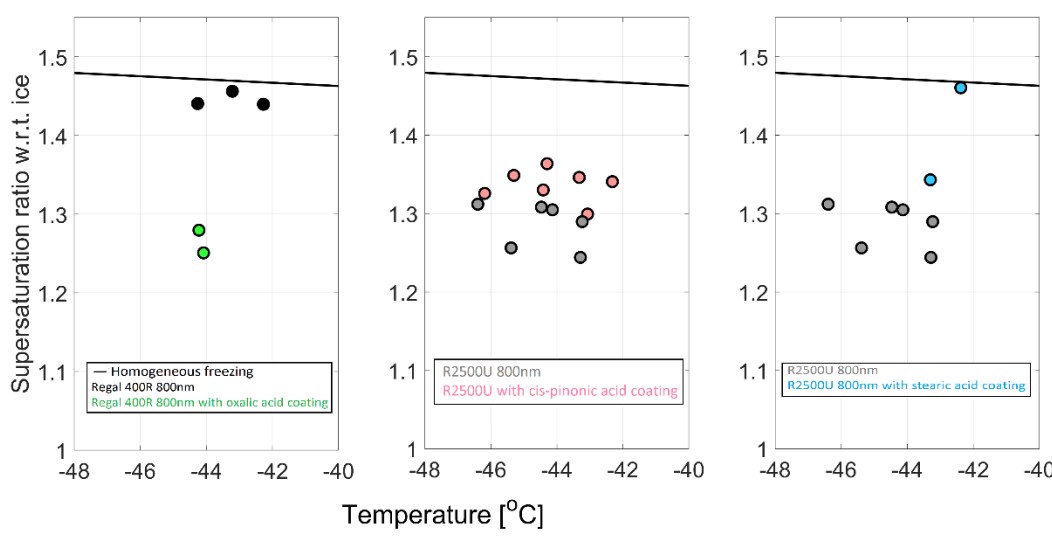

**Figure 8. Modification of ice nucleation onset on BC particles by organic coating. (a) oxalic acid on Regal 400R. (b) cis-pinonic acid on R2500U. (c) stearic acid on R2500U.**

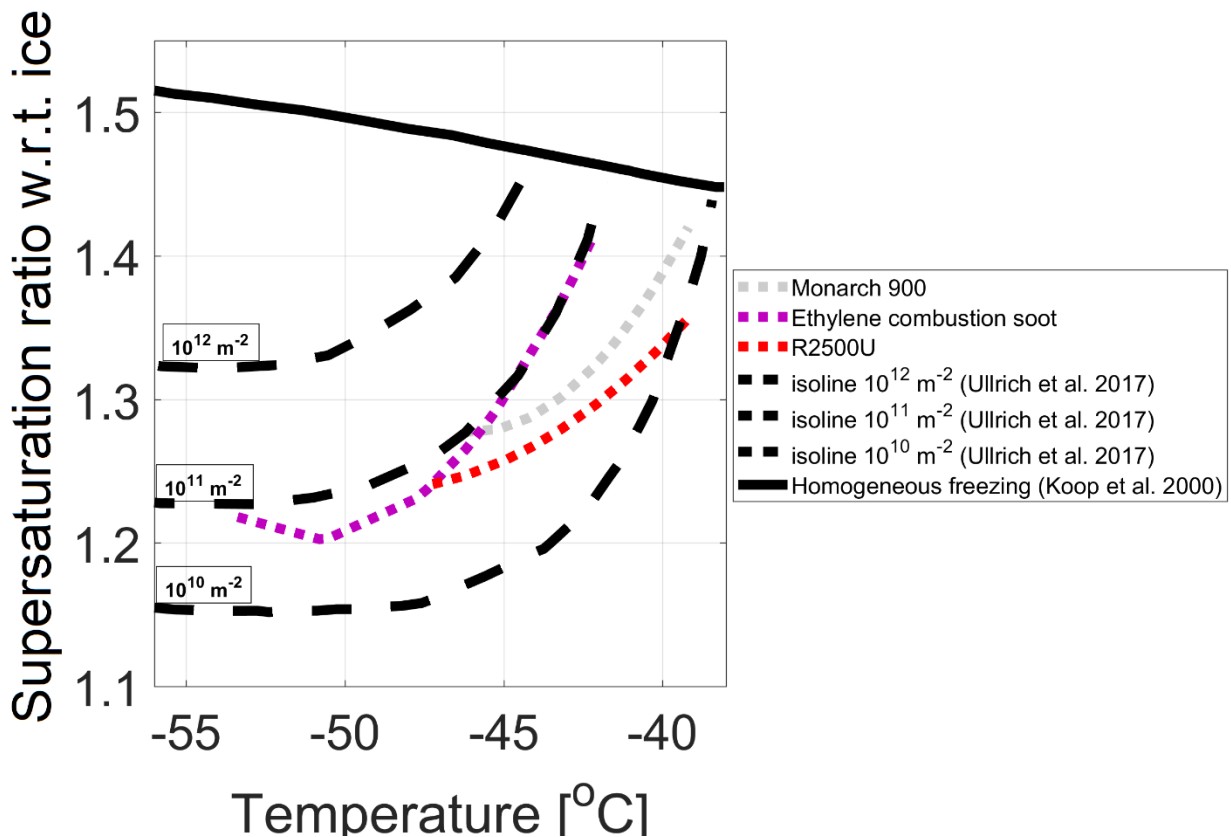

**Figure 9. Least-squares fitted active site density isolines of three most active INP in this study: Monarch 900, ethylene combustion soot, and R2500U (0.6 - 1.2 $\cdot 10^{10}$ m$^{-2}$), plotted together with isolines ($10^{10}$ - $10^{12}$ m$^{-2}$) reproduced from empirical parametrization of soot IN activity by Ullrich et al. (2017). The black solid line is the homogeneous freezing threshold.**

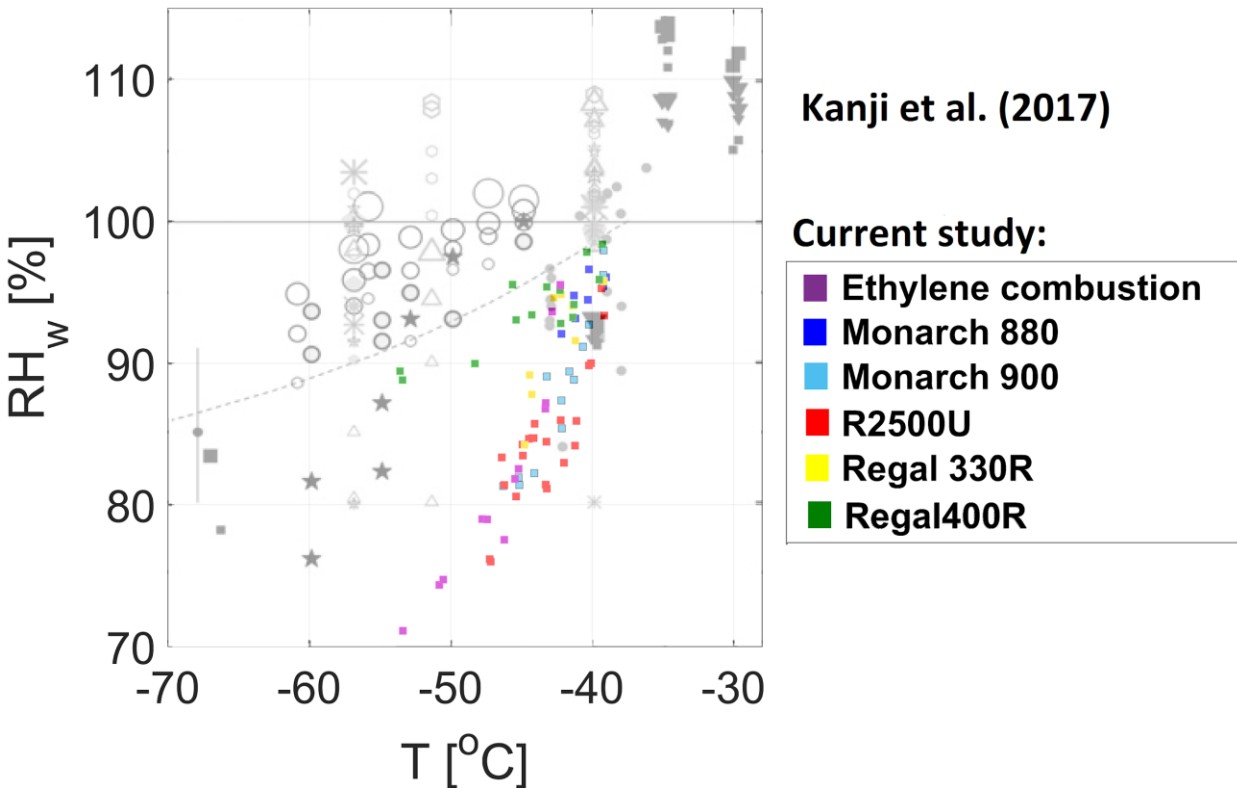

**Figure 10. Ice nucleation on 800 nm BC particles at subsaturated conditions w.r.t. water in the temperature range 217 to 235 K by 1 % of the particles. Current study results (colorcoded) complement the data on a figure reproduced from Kanji et al. (2017) (grayscale) and references therein. © American Meteorological Society. Used with permission.**

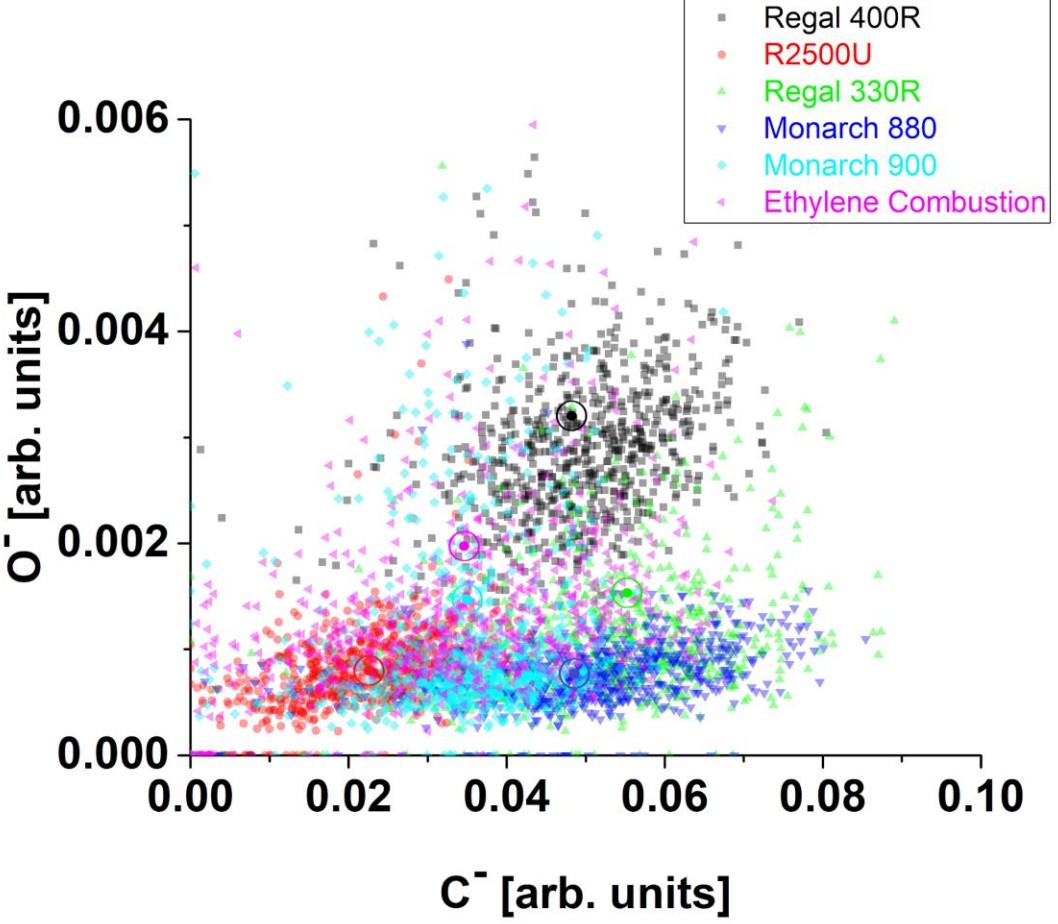

**Figure A1. Negative oxygen ion peak area plotted against carbon negative ion peak area derived from ~1000 size-selected (800 nm) single particle spectra of each BC sample (color-coded). Cluster centroids are color-coded and marked as ⊙. Regal 400R appears to have the highest O⁻ content.**

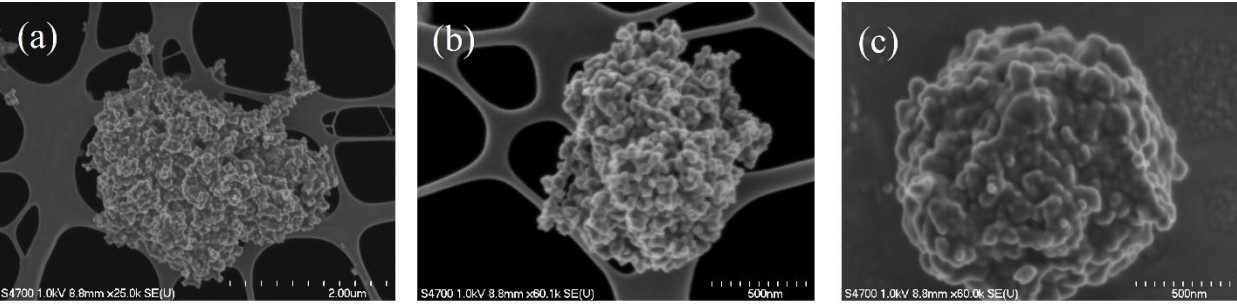

**Figure A2. Selected electron microscope images of dry dispersed agglomerates of (a) Ethylene combustion product, (b) Regal 400R, (c) R2500U. Most often occurring shape is compacted spheroidal.**