# Peer review of "Laboratory study of the heterogeneous ice nucleation on black carbon containing aerosol"

_Atmospheric Chemistry and Physics, 2018_

## Referee Comment (RC1) · Anonymous Referee #1 · 23 Oct 2018

*General comments: The authors conducted careful and dedicated lab experiments. The manuscript is well structured and carefully written to derive a delicate conclusion (i.e., P13L14-22). The research topic is an important addition to ACP. Though some parts seem speculative, the authors are knowledgeable in the subject, and their findings warrant future follow up studies. I support publication of this manuscript in ACP after the following comments (some are major/critical; e.g., P12 comments) are properly addressed.

*Summary: The authors studied the ice nucleation behavior of particles generated using diverse, commercially available BC materials and an ethylene combustion product in the temperature and SSi range of 217 - 235 K and 1.0 - ~1.5, respectively. In particular, six BC materials (possessing different physicochemical properties as experimental

variables) were used to look into the relationship of their IN abilities vs. morphologies, aerosolization methods, sizes (100 and 800 nm), degrees of surface oxidation and organic surface coating types under controlled settings. Out of these variables, the morphology and a subset of surface coatings seem dominant factors altering IN propensities of BC particles according to the results presented. The reviewer finds the notable suppression of BC-IN via oxidation and the tolerance (no substantial deformation) of BC-IN to the employed particle generation methods (i.e., both wet & dry dispersion) very interesting and informative, and their findings should be shared in the IN research community and beyond.

*Specific and technical comments: P1L16: It should read "...human health, aerosol-cloud interactions, precipitation formation, and climate.". Without aerosol-cloud interactions and precipitations, there is no climatic impact.

P1L19: It should read "...formation of ice crystals in temperature and ice supersaturation conditions relevant to cirrus clouds.".

P1L21 and elsewhere applicable: ice nuclei (IN) should be replaced with ice-nucleating particles (INPs) to be consistent with the common terminology typically used in the IN research community (Vali et al., 2015, ACP; https://doi.org/10.5194/acp-15-10263-2015). The authors can use IN for the abbreviation of 'ice nucleation' (e.g., P2L7).

P1L29: The reviewer suggests the authors to report/add SSi and SSw ranges used in SPIN alongside the T range here.

P1L34: "...dependence on temperature, supersaturation condition, and...".

P1L36: "initial ice nucleation ability" → "initial formation efficiency of pristine ice crystals" The authors provided an observational hint, but not any nanoscopic evidences of oxidation altering IN ability.

P2L7-9: The reviewer suggest the authors to expand the discussion of the indirect effect of soot/BC particles here. Perhaps, the discussion of Bond et al. (2013) in Sect.

4 better fits in here?

P3L7-9: "Modeling INPs requires . . . description (Knopf et al., 2018)." → "Two common approaches to parameterize IN of atmospherically relevant particles include a stochastic . . . description (e.g., Knopf et al., 2018)."

The reviewer presumes the authors mean parameterization by modeling. There exist many more papers should be cited here.

P3L9-10: Add Connolly et al. (2009; Atmos. Chem. Phys., 9, 2805–2824, 2009) and Niemand et al. (2012; DOI: 10.1175/JAS-D-11-0249.1) in addition to Vali (2014).

P3L11: The n_s abbreviation has been already given in L9.

P3L14-15: Cite at least Murray et al. (2012, Chem. Soc. Rev, 41), Vali (1994, J. Atmos. Sci., 51) and Vali (2014, ACP, 14) for references of time-dependence approach.

P3L19: (Marcolli, 2014 and 2017; Wagner et al., 2016; Ullrich et al., 2017) should be placed at the end of L19 instead of L20.

P3L24-28 & P9L16: So what is the pore size, which influences condensation process? The reviewer wonders if the authors can be a bit more quantitative than 'on the order of nanometers (L27)' on this discussion.

P4L13: The reviewer suggests the authors to add SSw and SSi ranges in addition to the T range.

P6L4 & P10L24-: What could be the potential influence of acid suspension generation process on IN activity? It can be another variable, correct?

Fig. 2 & P6L16: DMA should replace SEMS in Fig. 2, unless the authors measured size distribution of size-selected BC particles and are willing to present those data.

P6L18-20: Please include a discussion of why these three particular organic acids were used in this study. Atmospheric relevance? Please justify with proper citations.

P7L12: Please clarify how you estimated a geometric volume from a 2-D image.

P7L33: The reviewer presumes that SPIN was operated in the way of scanning SS from low SS to high SS at fixed T until the ice active fraction (AF) of 1% was observed. If so, please state so in the text. If other operational procedures were employed for this study, please describe and include them in the text.

L8L1: Maybe "..., a correction factor, Lf in Eqn. 1, (5.8) is ..." – the authors may want to remind the reader that this correction factor is relevant to what's discussed in the previous section.

L8L16: Can the authors please explain experimental error bars in the text or figure caption more quantitatively? How did the authors estimate these uncertainties. They seem larger at colder T and lower SS. What is causing it?

P8L21: Maybe "... other BC particle types all exhibit heterogeneous freezing abilities below Koop line (= homogeneous freezing and water saturation line in Fig. 3)." Is this what the authors mean? The relationship between the ice nucleation onset SSi and T is known as "isoline" (see Fig. 19 of Hoose and Möhler, 2012, ACP). The authors may explain it here for the reader to understand your point properly.

P9L5-12: Very nice results.

P9L27-28: Was the particle effective density measured using atomized- or air dispersed particles? Or the authors tried both individually and confirmed no difference? The reviewer is not asking any additional measurements. Please just address what has been done.

P9L31-37: Briefly describe the shift in polarization (i.e., s1, p1) and tell the readers what it physically means in the context of your study. Elaborate a bit further with including proper references.

P10L1-7: This part is interesting – the tolerance of BC particles towards compaction is a very unique feature of BC as compared to other compositions (e.g., dust surrogate,

Sullivan et al., 2010, AS&T, 44). The authors might want to address this point. The reviewer thinks that adding this only strengthens the paper.

P10L15-16 & L20-21: How does the $n\_s$ value of 800 nm ethylene combustion BC particles compare to that of 100 nm ones? The IN "efficiency" is perhaps similar? Please discuss it within this section.

P11L6-8: Speculative, but it is good as is.

P11L15: What would you suggest on how such a "through characterization" can be done? Are there any specific techniques/methods currently available for the nanoscopic surface characterization while cooling?

P11L16-27 & P12L26-27 & P13L21-22 & P1L32: What is the overall atmospheric implication of the observed results/differences depending on the type of organic coating? For instance, the enhancement of IN due to oxalic acid coting matters in what occasion/situation?

P12L1-2: Comparing AF onset of 1% to $n\_s$ is the apples-to-oranges comparison. The authors can covert 1% AF to $n\_s$ in Fig. 9 using Eqn. 1, and delineate their own "$n\_s$ isolines". For more information regarding the isoline, the authors may refer to Hiranuma et al., 2014 (Atmos. Chem. Phys., 14, 13145–13158, 2014) and/or Hoose and Möhler, 2012 (ACP). The authors might want to revise Fig. 9.

P12L7-10: Fig. 1-7 (b) of Kanji et al. (2017) represents a compilation of immersion and/or contact mode freezing data. Fig. 1-7 (a) of K17 is for the deposition data. Why do the authors compare their deposition ice nucleation data to Fig. 1-7 (b) instead of (a)?

P12L7: Why do the authors choose to estimate $n\_s$ "at the lowest measured temperatures"? If the authors wish to compare their data to K17 Fig. 1-7 (b), they should estimate the $n\_s$ values at the nearest point of water saturation (i.e., Koop line). Note that immersion freezing can be considered part of isolines (Hiranuma et al., 2014; Atmos. Chem. Phys., 14, 13145–13158, 2014). The reviewer suggests revising Fig. 10.

P12L34: The reviewer suggests adding the discussion of the atmospheric implication of what the authors found for the effect of oxidation here.

P13L18-19: See my comment for P10L15-16 & L20-21. Can the authors add the statement regarding IN "efficiency" to the IN activity statement?

P13L26-29: And the IN characterization at low Ts of the cold cirrus T regime.
* * *

---

## Referee Comment (RC2) · Anonymous Referee #2 · 29 Nov 2018

The comment was uploaded in the form of a supplement:
https://www.atmos-chem-phys-discuss.net/acp-2018-915/acp-2018-915-RC2-supplement.pdf

---

## Author Comment (AC1) · 8 Jul 2019

We thank the Reviewers for their detailed comments and helpful suggestions in an effort to improve this manuscript. We have addressed each comment below, with the Reviewers comment text written in black and our response text written in blue.

We deeply apologize for the extraordinary long delay in response to this review due to transition of the main author to a new position in another country and immediate deployment to two subsequent field campaigns.

**Reviewer #1**

*General comments: The authors conducted careful and dedicated lab experiments. The manuscript is well structured and carefully written to derive a delicate conclusion (i.e., P13L14-22). The research topic is an important addition to ACP. Though some parts seem speculative, the authors are knowledgeable in the subject, and their findings warrant future follow up studies. I support publication of this manuscript in ACP after the following comments (some are major/critical; e.g., P12 comments) are properly addressed.

*Summary: The authors studied the ice nucleation behavior of particles generated using diverse, commercially available BC materials and an ethylene combustion product in the temperature and SSi range of 217 - 235 K and 1.0 - ~1.5, respectively. In particular, six BC materials (possessing different physicochemical properties as experimental variables) were used to look into the relationship of their IN abilities vs. morphologies, aerosolization methods, sizes (100 and 800 nm), degrees of surface oxidation and organic surface coating types under controlled settings. Out of these variables, the morphology and a subset of surface coatings seem dominant factors altering IN propensities of BC particles according to the results presented. The Reviewer finds the notable suppression of BC-IN via oxidation and the tolerance (no substantial deformation) of BC-IN to the employed particle generation methods (i.e., both wet & dry dispersion) very interesting and informative, and their findings should be shared in the IN research community and beyond.

The authors thank Reviewer #1 for his/her helpful and valuable comments and suggestions. We have implemented our responses in the current revised manuscript as indicated in the "Response" sections following the comments.

**Specific and technical comments of Referee #1:**

**Comment 1:** P1L16: It should read "…human health, aerosol-cloud interactions, precipitation formation, and climate.". Without aerosol-cloud interactions and precipitations, there is no climatic impact.

**Response**: This suggested text-change has been implemented in the manuscript.

**Comment 2:** P1L19: It should read "…formation of ice crystals in temperature and ice supersaturation conditions relevant to cirrus clouds.".

**Response**: This suggested text-change has been implemented in the manuscript.

**Comment 3:** P1L21 and elsewhere applicable: ice nuclei (IN) should be replaced with ice-nucleating particles (INPs) to be consistent with the common terminology typically used in the IN research community (Vali et al., 2015, ACP; https://doi.org/10.5194/acp-15-10263-2015). The authors can use IN for the abbreviation of 'ice nucleation' (e.g., P2L7).

**Response**: These suggested changes in notation have been implemented in the manuscript.

**Comment 4:** P1L29: The Reviewer suggests the authors to report/add SSi and SSw ranges used in SPIN alongside the T range here.

**Response**: We have added the following sentence into the text:

"Ice nucleation activity was systematically examined in lamina temperature and saturation conditions ranging between $217 \leq T \leq 235$ K; $1.0 \leq S_{ice} \leq 1.5$; and $0.59 \leq S_{water} \leq 0.98$, respectively, using a SPectrometer for Ice Nuclei (SPIN) instrument, which is a continuous flow diffusion chamber coupled with instrumentation to measure light scattering and polarization."

**Comment 5:** P1L34: "…dependence on temperature, supersaturation condition, and…".

**Response**: This suggested text-change has been implemented in the manuscript.

**Comment 6:** P1L36: "initial ice nucleation ability"→"initial formation efficiency of pristine ice crystals" The authors provided an observational hint, but not any nanoscopic evidences of oxidation altering IN ability.

**Response**: We agree with the Reviewer's comment. This suggested text-change has been implemented in the manuscript.

**Comment 7:** P2L7-9: The reviewer suggest the authors to expand the discussion of the indirect effect of soot/BC particles here. Perhaps, the discussion of Bond et al. (2013) in Sect. 4 better fits in here?

**Response**: The discussion was moved here as suggested and the text now reads:

"Bond et al. (2013) modelled the contribution of soot to clouds and climate and distinguished between the homogeneous and heterogeneous nucleation types. They showed that in the case of ice clouds when homogeneous nucleation dominates, coverage of high clouds is reduced and cooling prevails while when heterogeneous nucleation of BC prevails, more high clouds are formed that in turn contributes indirectly to the warming effect."

**Comment 8:** P3L7–9: "Modeling INPs requires… description (Knopf et al., 2018)."→"Two common approaches to parameterize IN of atmospherically relevant particles include a stochastic… description (e.g., Knopf et al., 2018)." The reviewer presumes the authors mean parameterization by modeling. There exist many more papers should be cited here.

**Response**: The Reviewer is correct. This suggested text-change has been implemented in the manuscript.

**Comment 9:** P3L9-10: Add Connolly et al. (2009; Atmos. Chem. Phys., 9, 2805–2824, 2009) and Niemand et al. (2012; DOI: 10.1175/JAS-D-11-0249.1) in addition to Vali (2014).

**Response:** These citations have been added.

**Comment 10:** P3L11: The n_s abbreviation has been already given in L9.

**Response**: The text has been removed here.

**Comment 11:** P3L14-15: Cite at least Murray et al. (2012, Chem. Soc. Rev, 41), Vali (1994, J. Atmos. Sci., 51) and Vali (2014, ACP, 14) for references of time-dependence approach.

**Response**: These citations have been added.

**Comment 12:** P3L19: (Marcolli, 2014 and 2017; Wagner et al., 2016; Ullrich et al., 2017) should be placed at the end of L19 instead of L20.

**Response**: This change has been implemented, in addition this sentence has been moved to the next paragraph as requested by Reviewer2 (P3L18-21).

**Comment 13:** P3L24-28 & P9L16: So what is the pore size, which influences condensation process? The reviewer wonders if the authors can be a bit more quantitative than 'on the order of nanometers (L27)' on this discussion.

**Response**: The relevant pore sizes are 2-50 nm mesopores, especially pores less than 10 nm in diameter where the Kelvin effect is greatest. Both combustion-related soot particles and manufactured carbon particles can exhibit pores in this size range. We have changed the relevant text to read:

"… in accord with the inverse Kelvin equation (Marcolli, 2014). The diameter of mesopores (2-50 nm) in the particles affects the condensation process, especially for pores less than 10 nm in diameter where the Kelvin effect is greatest. In large diameter pores (>> 10 nm), the water vapor pressure is not sufficient to cause condensation below water saturation (< 100 % RHw). On the

other hand, in pores with diameters too small (<4 nm), the growth of an ice embryo may be inhibited (Marcolli, 2014). Pore diameters in soot materials range from micro (<2 nm pore diameter), through meso (2-50 nm diameter), to macro (>50 nm diameter) and are dependent on the specific soot material. Manufactured carbon black material (e.g., Kruk et al., 1996) can be produced with similar or higher surface areas and mesoporosity than combustion-related soot particles (e.g., Rockne et al., 2000). All else being equal, IN activity of a material is expected to increase with increasing number of pores in the suitable diameter range. The PCF mechanism also predicts the observed decrease in ice nucleation activity with increasing temperature over the temperature range about 210 K to 240K (typical of cirrus clouds) (Hoose and Möhler, 2012)."

**Comment 14:** P4L13: The reviewer suggests the authors to add SSw and SSi ranges in addition to the T range.

**Response**: This has been implemented in connection with Comment 4 and is now incorporated in the text of the manuscript.

**Comment 15:** P6L4 & P10L24-: What could be the potential influence of acid suspension generation process on IN activity? It can be another variable, correct?

**Response**: The Reviewer is correct. We initially chose Regal 400R due to its similarity to Regal 330R, with the exception of its oxidized surface. As the Reviewer points out, Regal 400R is both the only surface oxidized BC particle type we studied, but also the only one that generates an acidic aqueous suspension. We have included the following text on P10:

"We also measured the pH values (Table 1) obtained for aqueous suspensions of the BC samples. The Regal 400R particle type is the only BC sample in this study that has both an oxidized surface and generates an acidic aqueous suspension. The surface acidity is likely due to surface-bound oxygen-containing functional groups; both the surface functional groups and surface porosity have been shown to influence the amount and the energetics of the water adsorbed to the carbon surfaces (Salame and Bandosz, 1999; Marsh and Rodriguez-Reinoso, 2006). Currently, we have no way of discriminating between the effects of surface oxidation and surface acidity on the observed IN activity of Regal 400R."

**Comment 16:** Fig. 2 & P6L16: DMA should replace SEMS in Fig. 2, unless the authors measured size distribution of size-selected BC particles and are willing to present those data.

**Response**: We changed the abbreviation in Fig.2..

**Comment 17:** P6L18-20: Please include a discussion of why these three particular organic acids were used in this study. Atmospheric relevance? Please justify with proper citations.

**Response**: To address the Reviewer's question, we added the following text:

"Stearic acid ($C_{18}H_{36}O_2$) is one of the abundant saturated fatty acids and it is a common constituent of atmospheric particles in urban areas that cook large amounts of meat (Katrib et al., 2005). Stearic-acid coated particles are a gross simplification of atmospheric particles since urban aerosol particles are composed of hundreds if not thousands of organic molecules. However, these particles could serve as a proxy of the broader class of soot coated with fatty acids.

Humic-like substances are very efficient surfactants. One of the commonly used model surfactants is cis-pinonic acid ($C_{10}H_{16}O_3$). This compound originates from boreal forests, and blooming algae in the oceans yielded from the photochemical oxidation of the evaporated α-pinene in the lower troposphere (Luo and Yu, 2010).

Dicarboxylic acids are another important group of organic compounds identified in the atmospheric aerosols. Their contribution to the total particulate carbon ranges from about 1-3% in the urban and semi-urban areas to values close to or even above 10% in the remote marine environment (Kerminen et al., 2000). Dicarboxylic acids have several different sources, including primary emissions from fossil fuel combustion and biomass burning (Chebbi and Carlier, 1996). Here, we examine coatings with oxalic acid ($C_2H_2O_4$), an abundant dicarboxylic acid in the lower troposphere, comprising a significant fraction of the total diacid mass concentration (Kerminen et al., 2000)."

**Comment 18:** P7L12: Please clarify how you estimated a geometric volume from a 2-D image.

**Response**: This is a well-known challenge in SEM and other 2-D image analysis, for example an analysis of ice crystal 2-D shadows by airborne cloud probes. One suggested approach that we tested with several particles is scanning the tilted sample, however it is not practical to scan all possible rotated facets for each particle. Therefore, we had to consider previous observations of BC agglomerates of this diameter and make some assumptions (as stated in the text) based on the inherent characteristics of the process and complementary observations. For example, since there is no spatial orientation preference in the flow during the process of agglomeration, we assume that a non-ideal symmetrical shape is formed. In addition, if the 3-D shape of the particle was significantly aspherical i.e. asymmetric in one or more of the axes, we would expect to see a bimodal or higher mode size distributions measured by the OPC or SEMS scans; however, this was not the case. Thirdly, unlike visibly branched shapes, the compact shapes, which have round 2-D projection would have to go through a highly unfavorable aerodynamic agglomeration processes to have a non-symmetrical shape at the end, e.g. cylinder without being lost to the walls. Lastly, if our assumption was completely wrong, we would expect to find approximately equal amount of aspheroidal shapes in SEM images, this was not the case.

Similar simplistic approaches of spherical symmetry is often used for the representation of BC particles in models (Cappa et al., 2012; Bond et al., 2013).

*Cappa, C. D. et al. Radiative Absorption Enhancements Due to the Mixing State of Atmospheric Black Carbon. Science 337, 1078–1081 (2012).

*Bond, T. et al. Bounding the role of black carbon in the climate system: a scientific assessment. J. Geophys. Res. 118, 5380–5552 (2013).

We added the following sentence:

"The mean diameter of the circular 2-D shape was used to calculate the volume of a geometric sphere having the same diameter in all axes."

**Comment 19:** P7L33: The reviewer presumes that SPIN was operated in the way of scanning SS from low SS to high SS at fixed T until the ice active fraction (AF) of 1% was observed. If so, please state so in the text. If other operational procedures were employed for this study, please describe and include them in the text.

**Response**: The Reviewer is correct, the SPIN was operated in a scanning mode, from low SSi to highest SSi. However, the run was stopped at the maximum RHi =150 % as explained in P7L33 and not until 1% AF. This procedural information was added to the text:

"The SPIN was operated in a supersaturation scanning mode, from low RHi to high RHi at each fixed temperature."

**Comment 20:** P8L1: Maybe "…, a correction factor, Lf in Eqn. 1, (5.8) is …" – the authors may want to remind the reader that this correction factor is relevant to what's discussed in the previous section.

**Response**: A short reminder was added as suggested by the Reviewer.

**Comment 21:** P8L16: Can the authors please explain experimental error bars in the text or figure caption more quantitatively? How did the authors estimate these uncertainties. They seem larger at colder T and lower SS. What is causing it?

**Response**: Due to slight heterogeneities in wall temperatures, the temperature and supersaturation in the SPIN aerosol lamina vary. The effect of this variability on lamina conditions is quantified using results from CFD simulations performed by Kulkarni and Kok (2012). The uncertainty in the average lamina temperature and supersaturation is calculated using 16 thermocouples positioned on each wall. At higher supersaturations and colder lamina temperatures, the wall temperatures exhibit greater variability as the PID temperature controllers rely less on the thermoelectric heaters to maintain cooler wall temperatures. This increases the heterogeneity in wall temperature, leading to higher uncertainties.

We added the following: ".... error bars, which represent the variability of the laminar conditions based on CFD simulations by Kulkarni and Kok (2012), are presented….".

**Comment 22:** P8L21: Maybe "… other BC particle types all exhibit heterogeneous freezing abilities below Koop line (= homogeneous freezing and water saturation line in Fig. 3)." Is this

what the authors mean? The relationship between the ice nucleation onset SSi and T is known as "isoline" (see Fig. 19 of Hoose and Möhler, 2012, ACP). The authors may explain it here for the reader to understand your point properly.

**Response:** Reviewer2 was concerned with the word heterogeneous, therefore the authors changed the sentence to:

"The Regal 400R results remain indistinguishable from the homogeneous ice nucleation line, whereas the other BC particle types all exhibit freezing abilities below Koop line (i.e. the homogeneous freezing and water saturation line in Fig. 3). The relationship between the ice nucleation onset SSi and temperature is known as "isoline" (see Fig. 19 of Hoose and Möhler, 2012). More efficient INP have lower isolines."

**Comment 23:** P9L5-12: Very nice results.

**Response**: Thank you!

**Comment 24:** P9L27-28: Was the particle effective density measured using atomized- or air dispersed particles? Or the authors tried both individually and confirmed no difference? The reviewer is not asking any additional measurements. Please just address what has been done.

**Response**: The densities were measured only for the dry-dispersed particles. We added this information in the sentence below:

"The ratio of the vacuum aerodynamic diameter, measured by the PALMS instrument for each air dispersed BC sample, and the mobility diameter gives the effective density (DeCarlo et al., 2004). For a constant selected mobility diameter of 800 nm, we observed variability in the effective density (Table 1)."

**Comment 25:** P9L31-37: Briefly describe the shift in polarization (i.e., s1, p1) and tell the readers what it physically means in the context of your study. Elaborate a bit further with including proper references.

**Response**: Text changed to:

"Another method for physical characterization is the detection of the shift in polarization in the light scattered from 800 nm BC particles in the OPC. A shift in polarization of the linearly polarized incident beam will occur if the particle is optically anisotropic (having aspherical shape, branches, roughness, or variations in internal structure). This polarization shift is used in classification of particles by their 'optical shape' (e.g. Garimella et al. 2016; Nichman et al., 2016; Kobayashi et al., 2014; ; Glen and Brooks, 2013). Even spheroidal shaped particles produce a unique shape-specific phase function distinctly different from those produced by other spheroidal particles (Mischenko et al., 1997). Thus parallel and perpendicular polarization measurements can be used to differentiate between particles of the same diameter. Francis et al. (2011) showed that both, agglomerate diameter and spherule diameter of soot, affect the polarization within specific boundaries. In our study, the optical sphericity could shed light on the BC particle shape (i.e. round

and compact versus branched and lacy) and its influence on the IN mechanism in BC agglomerates. High optical sphericity of a particle is determined by a low polarization shift in the light scattered from the particle. OPC data cluster centers of single-particle optical measurements for each BC sample are listed in Table 1."

**Comment 26: P**10L1-7: This part is interesting – the tolerance of BC particles towards compaction is a very unique feature of BC as compared to other compositions (e.g., dust surrogate, Sullivan et al., 2010, AS&T, 44). The authors might want to address this point. The reviewer thinks that adding this only strengthens the paper.

**Response**: We added a sentence at the end of the paragraph:

"The tolerance of these compact BC particles towards further compaction is a very unique feature of BC as compared to other insoluble compositions, e.g., dust surrogates (see Sullivan et al., 2010)."

**Comment 27:** P10L15-16 & L20-21: How does the $n_s$ value of 800 nm ethylene combustion BC particles compare to that of 100 nm ones? The IN "efficiency" is perhaps similar? Please discuss it within this section.

**Response**: The IN efficiency should be similar if we take the 1% value as the IN onset point for both diameters and for simplicity disregard the interference of the doubly charged particles. In that case, the main factor will be the effective surface area which is approximately two orders of magnitude smaller for 100 nm particles in comparison to 800 nm particles. Therefore, the active site density isoline for 100 nm particles will be two orders of magnitude higher (Eq. 1), closer to the homogeneous freezing line. A higher active site density requires higher saturation to nucleate 1% of the particles as the water molecules are spread in a denser active sites environment.

We added this calculation in the SI. The authors think that the addition of discussion in the main text would not fit here since the active site density is discussed much later in Section 3.2. We added the following text in Sect. 3.2:

"For the same IN onset threshold (1%) for particles of 100 nm in diameter, the effective surface area will be approximately 2 orders of magnitude lower. Therefore, for particles of 100 nm in diameter the active site density isolines will have to be two orders of magnitude higher, closer to the homogeneous freezing line".

**Comment 28:** P11L6-8: Speculative, but it is good as is.

**Response**: Thank you.

**Comment 29:** P11L15: What would you suggest on how such a "thorough characterization" can be done? Are there any specific techniques/methods currently available for the nanoscopic surface characterization while cooling?

**Response**:  The authors meant that screening of additional BC samples with known properties, by changing one parameter at a time, could be useful. A thorough characterization of each sample before cooling, in particular complementary measurements of BET, contact angle, chemical analysis by thermal desorption of organics, conductivity/resistance measurements and other advanced techniques (e.g. Fung et al., 2008) could shed more light on the intrinsic properties of the particles and the pores (e.g. Fahlén and Salmén, 2005).

On the other hand, the commonly used techniques for in situ 'surface characterization while cooling' are 'cold stage' experiments, where instruments such as SEM-EDX, AFM and others are used.

*Fung, R., Shneerson, V., Saldin, D. K. and Ourmazd, A.: Structure from fleeting illumination of faint spinning objects in flight, Nat. Phys., 5, 64 [online] Available from: https://doi.org/10.1038/nphys1129, 2008.

*Fahlén, J. and Salmén, L.: Pore and Matrix Distribution in the Fiber Wall Revealed by Atomic Force Microscopy and Image Analysis, Biomacromolecules, 6(1), 433–438, doi:10.1021/bm040068x, 2005.

We added to the text:

"However, further screening of BC samples accompanied by thorough characterization (e.g. BET, Atomic Force Microscopy, contact angle, cold-stage experiments) is needed to confirm these findings."

**Comment 30:** P11L16-27 & P12L26-27 & P13L21-22 & P1L32: What is the overall atmospheric implication of the observed results/differences depending on the type of organic coating? For instance, the enhancement of IN due to oxalic acid coating matters in what occasion/situation?

**Response**:  It is challenging to draw generalized atmospherically relevant conclusions from such a simplistic and narrow representation. In addition, there is also dependence on the thickness and pore filling by the coating. Overall our understanding is that the activity of an efficient porous BC INP can be inhibited by high degree of oxidation and coating of most of the tested organic acids, while the activity of inefficient BC INP can be enhanced by the active pores and by oxalic acid coating.

For example, Kawamura and Kaplan (1987) studied acids in the Los Angeles ambient air, they found that oxalic acid was the dominant specie among dicarboxylic acids in the aerosol phase. These aerosol particles can reach regions with elevated temperatures such as forest fires and transit into the vapor phase together with oxalic acid vapors emitted directly from forest fires in that area. The vapors can then condense at higher altitude on the bigger BC particles blown from the fires or BC from fossil fuel combustion blown from Los Angeles area. These coated particles could be potentially more efficient ice nuclei at lower relative humidity.

We think that it would be too speculative to add any specific scenario to the text, and we will keep the generalized conclusion that "Organic surface coatings demonstrated the capability of both enhancing and inhibiting the IN activity on BC proxies".

*Kawamura K. and Kaplan I. R. (1987) Motor exhaust emissions as primary source for dicarboxylic acids in Los Angeles ambient air. Envir. Sci. Technol. 21, 105-110.

**Comment 31:** P12L1-2: Comparing AF onset of 1% to n_s is the apples-to-oranges comparison. The authors can covert 1% AF to n_s in Fig. 9 using Eqn. 1, and delineate their own "n_s isolines". For more information regarding the isoline, the authors may refer to Hiranuma et al., 2014 (Atmos. Chem. Phys., 14, 13145–13158, 2014) and/or Hoose and Möhler,2012 (ACP). The authors might want to revise Fig. 9.

**Response**: The authors appreciate this comment. Figure 9 was changed. The new Fig. 9 has 3 fitted active site density isolines for the most active samples. All fall in the narrow range 0.6 - 1.2 x$10^{10}$ m$^{-2}$.

**Comment 32:** P12L7-10: Fig. 1-7 (b) of Kanji et al. (2017) represents a compilation of immersion and/or contact mode freezing data. Fig. 1-7 (a) of K17 is for the deposition data. Why do the authors compare their deposition ice nucleation data to Fig. 1-7 (b) instead of (a)?

**Response**: As the Reviewer suggested to compare to deposition mode data from Kanji et al (2017), Fig 1-7(a),  we replaced figure 10 to accommodate his request.

**Comment 33:** P12L7: Why do the authors choose to estimate n_s "at the lowest measured temperatures"? If the authors wish to compare their data to K17 Fig. 1-7 (b), they should estimate the n_s values at the nearest point of water saturation (i.e., Koop line). Note that immersion freezing can be considered part of isolines (Hiranuma et al., 2014; Atmos. Chem. Phys., 14, 13145–13158, 2014). The reviewer suggests revising Fig. 10.

**Response**: Fig. 10 was replaced to the lower temperature regime, as suggested in the previous comment by the Reviewer.

**Comment 34:** P12L34: The reviewer suggests adding the discussion of the atmospheric implication of what the authors found for the effect of oxidation here.

**Response**:  We have added the following text:

"Another possible reason is the surface oxidation of the particle as part of the aging process down the timeline of its trajectory in the atmosphere. Several studies, including this one, have shown that oxidation on carbonaceous surfaces can, in some cases, reduce the efficiency of ice nucleation".

**Comment 35:** P13L18-19: See my comment for P10L15-16 & L20-21. Can the authors add the statement regarding IN "efficiency" to the IN activity statement?

**Response**: We changed the text in the bullet point to:

"While comparing particle size (i.e. agglomerate versus aggregate) of the same BC sample, which has the same IN efficiency, we observed a significantly lower IN activity at 100 nm mobility diameter versus 800 nm."

**Comment 36:** P13L26-29: And the IN characterization at low Ts of the cold cirrus T regime.

**Response**: We have added the suggested text.

"....and for characterization of IN at low temperatures of the cold cirrus temperature regime".

**Reviewer #2**

The manuscript by Nichman et al. presents a laboratory investigation of the ice nucleation ability of black carbon (BC) particles of different type, an important topic, which fits within the scope of ACP.

The authors used two different sizes of BC particles (100 nm, 800 nm) in order to investigate the effects of particle physicochemical properties on their ice nucleation ability (morphology, surface oxidation, coating). The main conclusion from the presented manuscript is the attribution of a pore condensation and freezing (PCF) mechanism for the ice nucleation ability of BC particles.

The manuscript in its current state is incomplete (investigation of the effect of coating) and seems speculative for some of the effects investigated (surface oxidation). Besides, the manuscript lacks a proper in-depth analysis and discussion of some of the results presented (e.g. comment P8L25-27) and stays superficial at many places (e.g. effect of particle generation).

In general the description of observed "heterogeneous ice nucleation of BC particles" (P8L28) and the suggested PCF mechanism, where the pore water freezes homogeneously, is misleading and/or not stated with sufficient care: While a PCF mechanism as such could not take place without the presence of a pore, i.e. the particle, forming the site for the water vapor to condense ("heterogeneous freezing process"), the actual ice formation takes place through homogeneous nucleation of this pore water.

Many flaws could have been prevented by a more critical read by some of the more senior authors prior to submission. I suggest major revisions of the manuscript in its current form and reevaluating it after the points listed below have been carefully addressed and incorporated.

We thank the Reviewer for the careful read of our manuscript and the exhaustive efforts to help ensure the integrity of our work. We have carefully addressed and incorporated many of the issues raised by Reviewer #2 and trust that our efforts have strengthened our manuscript.

**General Comments:**

How do the industrial BC particles investigated in this study compare to atmospheric soot particles, for instance those emitted by aircrafts or fly ash particles (Grawe et al. 2018; Umo et al. 2015)? The particles tested might not be atmospherically relevant due to different physiochemical properties. This limitation should be explained and discussed in more detail in Sect. 4. Also, why do you refer to your particles as "BC-containing particles" in the abstract and Sect. 1? What else is in these particles?

The carbon blacks investigated in this study were selected by their morphology and surface oxidation to resemble and serve as proxies of BC, similar to the BC we collected from ethylene combustion in our laboratory. The carbon blacks, an industrial powder product, is potentially atmospherically relevant both directly, as a pollution component, especially when generated in processes such as "channel process", mostly in developing countries (Dannenberg et al., 2000; Hardman, 2017), and indirectly as a proxy for atmospheric particle as was done in several previous studies (e.g. recently by Dalirian et al., 2018).

We didn't do a comparison study with aircraft emissions or fly ash particles originating from various sources with high variability. Therefore, it would be hard to speculate how a well-defined commercial batches of carbon black compare to samples collected in the atmosphere at various locations / conditions / techniques. Canagaratna et al. (2015) have shown that Regal black and flame soot appear very similar, at least from the perspective of the mass spectrometry. However, it should of course be borne in mind that in the ambient setting, BC particles can vary significantly in terms of their physical and chemical properties, and are usually mixed with other pollutants present in the atmosphere. Laboratory studies allow us to examine the individual components of the ice nucleation process of BC in a controlled environment.

We will briefly discuss the relevance of these BC samples to the real atmosphere in Sect. 4 as suggested by the Reviewer. We have changed the text to read: " The carbon black, an industrial powder product, is potentially atmospherically relevant both directly, as a pollution component, especially when generated in a "channel process", mostly in developing countries (Dannenberg et al., 2000; Hardman, 2017), and indirectly as a proxy for atmospheric particle as was recently studied by Dalirian et al., 2018 and others. Canagaratna et al. (2015) have shown that Regal black and flame soot appear very similar, at least from the perspective of the mass spectrometry. However, in the ambient setting, BC particles can vary significantly in terms of their physical and chemical properties, and are usually mixed with other pollutants present in the atmosphere"

In terms of BC terminology, we have chosen to follow the recommendations described by Petzold et al. (2013) as closely as possible. In general, soot particles are incomplete combustion products, a subset of which contain an elemental carbon-based, refractory, light-absorbing component termed black carbon (BC). Therefore, some soot particles, including cases described in the introduction, may consist of many chemical components that have little BC content (e.g. chars, coke-oven emissions, Brown Carbon). In this study, we used ethylene combustion soot and manufactured carbon black particles, which consist of black carbon particles without organic matter. These are BC particles. We also coat some of these particles with specific compounds, generating BC-containing particles. We have checked the manuscript to ensure that our terminology is consistent, making changes as necessary.

I suggest to reduce the discussion on field study results (P2L16-27) and put the focus on results of laboratory studies, comparable to the presented manuscript.

We have reduced the discussion on field study results (P2L16-27) as follows:

[revised manuscript text omitted]

**Investigation of surface oxidation:** P10L24-P11L15: Even though, I agree that surface oxidation state might play an important role for the ice nucleation ability of (soot) aerosol particles, your data are insufficient to make a clear statement here, as there is no systematic experiments provided, showing a clear effect of surface oxidation on the ice nucleation ability of

soot. In the framework of PCF, a surface oxidation should lower the soot-water contact angle and thus enhance the ice nucleation ability of a soot by PCF.

We thank the Reviewer for underscoring the importance of investigating BC particle surface oxidation effects on IN activity. In this work, we explicitly tested two different carbon black samples (Cabot Regal 400R and 330R) that exhibited similar physical characteristics, but differed in their surface oxidation state. We investigated the IN activity (i.e., ice onset) of these two samples as a function of temperature. We report on observations showing different IN activities for these two samples, where the surface oxidized sample exhibited lower IN activity at a given temperature than did the non-oxidized sample.

We respectfully disagree with the Reviewer concerning the clarity of our observed differences in IN activity. Please refer to Figures 4 and 7. Our studies were as systematic as our resources allowed for this project. It is unfortunate that the temperature range investigated for the Regal 330R was narrower than that for the Regal 400R, but our observed differences in IN activity are clear. Further, our observations are not dependent upon the theoretical framework of the PCF mechanism, even as we attempt to make sense of our observations within the PCF framework. While we agree with the Reviewer's inferred prediction based on the PCF mechanism, the fact that our observations counter the PCF-predicted result provides all the more reason to publish our results to the wider ice nucleation community.

Next, the observation, that the surface oxidized soot (Regal 400R) freezes homogeneously is lacking a proper discussion. The pH-values of this soot is measured at 4.9, indicating that soluble material was washed off the particle surface in solution. This indicates the presence of soluble material, which agrees well with the observed freezing along the homogeneous freezing line. I therefore suggest to completely revising the discussion on surface oxidation and its influence on ice nucleation presented in the manuscript.

We respectfully disagree with the Reviewer on the suggestion that our experiments or discussion on the effects of surface oxidation is incorrect, however lacking it may be. We point out that oxidizing the surface of carbon black particles does not necessarily imply that there are soluble materials on the surfaces of the particles; the oxidation may result in "dangling" -OH or -ROH or R-COOH or R-SO4 (for example) functional groups covalently bonded to the BC particle surfaces, as noted in activated carbon (Marsh and Rodriguez-Reinoso, 2006). These functional groups would not wash off, though they could become further chemically modified.

In this study, we systematically measured the ice onset of the surface oxidized Regal 400R in the temperature range 219 - 235K. All results showed ice onset near the homogeneous freezing limit, despite our expectations (as the Reviewer stressed, oxidized surfaces should have enhanced the wettability and the IN activity of soot). We compared the Regal 400R at higher temperatures (e.g. 228K) with its counterpart - a non-oxidized Regal 330R with the same physical parameters of AON and BET and observed a significant difference in the IN ability.

Finally, if soluble material was in fact the explanation to this counterintuitive behavior of Regal 400R, we would expect to see significant differences for different generation techniques (Fig. 5), such as with water-based atomization, where the "soluble material" may get removed. However, the ice onsets were the same for both generation techniques.

Thus, through a systematic (if limited) study of the temperature and aerosol generation techniques of two different carbon black samples, that differed by their surface oxidation, we show an explicit and unexpected observation whereby the surface oxidized carbon black exhibited a lower IN activity (i.e., higher ice onset) than the non-oxidized carbon black for the same size (mobility selected 800 nm particles) and temperature.

As the Reviewer's comments underscore, surface oxidation is an important aspect of BC particle IN activity to understand. In this regard, we fully agree. Based on our discussions here and comment 15 (P6L4 & P10L24) from Reviewer #1, we have revised our discussion on surface oxidation and its influence on ice nucleation as follows.

**"(4) The influence of surface oxidation:** In Fig. 7, the oxidized sample of Regal 400R (green) is compared to a similar non-oxidized sample of Regal 330R (yellow). These samples were both produced by the same company (Cabot) with very similar physical parameters of AON and BET, but had different surface chemical properties. The surface oxidized Regal 400R samples exhibited ice onset near the homogeneous freezing limit across the temperature range 219 - 235 K. In contrast, non-oxidized Regal 330R samples exhibited a divergent ice onset trend, exhibiting a decreasing supersaturation over ice at ice onset (relative to the homogeneous freezing limit) with decreasing temperature across the temperature range 228 – 235 K.

We corroborated the manufacturer's stated difference in surface oxidation by measuring with the PALMS instrument to confirm the oxidation state of dry particles. Negative and positive ion spectra of about 1000 particles were collected for each BC sample. The oxygen negative ion peaks were then plotted against carbon negative ion peaks and color-coded for each BC sample (Fig. A1). Regal 400R particles cluster demonstrates noticeably higher peaks of oxygen in comparison to other BC samples. Finally, to address the potential question of whether soluble material on the BC samples could explain this counterintuitive behavior, we generated 800 nm diameter Regal 400R particles using two different aerosol generation techniques, water-based atomization, where soluble material would get diluted or removed, and dry particle dispersion. As shown in Fig. 5, the ice onsets were the same for both generation techniques.

We also measured the pH values (Table 1) obtained for aqueous suspensions of the BC samples. The Regal 400R particle type is the only BC sample in this study that has both an oxidized surface and generates an acidic aqueous suspension. The surface acidity is likely due to surface-bound oxygen-containing functional groups; both the surface functional groups and surface porosity have been shown to influence the amount and the energetics of the water adsorbed to the carbon surfaces (Salame and Bandosz, 1999; Marsh and Rodriguez-Reinoso, 2006). Currently, we have no way of discriminating between the effects of surface oxidation and surface acidity on the observed IN activity of Regal 400R.

The result that oxidized carbon surfaces nucleate ice less well than non-oxidized carbon surfaces ones is somewhat counterintuitive due to the presumed hydrophilicity of the oxidized sample. However, the ubiquity of oxygenated surface groups on BC surfaces does not mean that soot particles will appear hydrophilic on a macroscopic scale. For example, fresh, oxidized soot particles do not generally activate as cloud-condensation nuclei (CCN) under atmospherically relevant conditions (Corbin et al., 2015). Moreover, molecular dynamics calculations show that hydrophilicity is not a sufficient condition for IN (Lupi et al., 2014a, 2014b). In fact, Biggs et al. (2017) reported an increase in the ice nucleation activity due to a decrease in hydrophilicity. The

freezing behavior of water confined in pores of an oxidized BC sample is not affected by pore wall hydrophobicity or hydrophilicity (Morishige, 2018). Häusler et al. (2018) suggested that agglomeration may lead to a favorable positioning of the functional sites and therefore to an increase in the IN activity, even though a decrease in the surface area occurs. However, it was found that the increased proportion of oxygen increases the hydrophilicity of graphene, reduces agglomeration and hence increases the surface area and reduces the number of pores (Häusler et al., 2018).

Our observations of homogeneous ice nucleation by the oxidized BC sample suggest that the surface oxidation of BC inhibits heterogeneous ice nucleation. If the ice nucleation activity of the BC samples we studied are controlled by nanometer-sized pores, as indicated by the PCF mechanism, then the surface oxidation is inhibiting the formation of ice within these pores. The oxidized pore surfaces may impose dipole orientations in the pore-enclosed water that raise the free energy of formation of ice embryos (Fletcher, 1959). The oxidized surfaces may affect the surface tension of the condensed water near the pore entrances, reducing the Kelvin effect required for the PCF mechanism (Marcolli, 2014). These oxidized surfaces likely attract a water layer under subsaturated conditions (Salame and Bandosz, 1999), which may explain the surprisingly high effective density of the Regal 400R sample.

The combined contribution of single spherule size, particle size, surface oxidation, and morphology to IN activity affects the spatial arrangement and thus the adjacent angles in the pores that dictate the formation, and perhaps the type, of the ice lattice (Bi et al., 2017; Zhu et al., 2018). However, further screening of BC samples accompanied by thorough characterization is needed to confirm these findings."

**Investigation of coating:** P11L16-27: The coating is insufficiently characterized and SEM or PALMS measurements should be provided to do so.

If characterization here refers to coating thickness, we did not collect samples of coated particles for SEM analysis or PALMS, which are by definition not the ideal tools to conduct a thorough investigation of thin organic coatings. However, we did measure IN activity of pure compounds (Fig. S3) and our comparison of all the samples suggests that the only possible effect of coating on IN has been demonstrated.

On P6L18-26 the authors make it sound like an effect of coating on the ice nucleation ability of soots is systematically investigated using different soots and different coatings. The results presented in Fig. 8 and discussed on P11L16-27 read like a contingent selection of data, which is insufficiently discussed and interpreted, with the majority of P11L16-27 presenting a repetition of information that had already been given in the introduction. Why was R2500U only coated by cis-pionic acid and stearic acid, but not with oxalic acid?

The authors did not claim that coatings were systematically investigated. The BC samples and the organic coatings were selected to show that coatings on BC particles can overshadow the underlying IN activities. BC materials with highest and lowest IN activity were chosen to test the effect of coating on IN activity. Similarly, acids with lowest and highest IN activity were chosen as coatings. The atmospheric importance of the selected organic compounds was added to the manuscript. There is a low probability to detect inhibition on less active BC INP and similarly it

is very difficult to detect an enhancement in IN activity on very active BC INPs. For this reason, we selected the materials with the highest probability to observe shifts as a result of coating. These measurements are complementary to the main results of IN activity of BC and are presented here to highlight the importance of coatings even in the case of very efficient PCF mechanism.

Please provide all coating measurements performed for each soot type, so that an effect can be systematically investigated and discussed. Otherwise the promises made on P5L12 about providing "some clarification on the effect of organics on the IN activity of BC particles" is obsolete. In case this data is not available, I suggest removing all data and discussion on organic coating of BC particles from the manuscript.

Despite the limited number of compounds studied here, there are neither previous reports of coatings on these BC compounds nor reports of their ice nucleation measurements. Therefore, the authors think these new results cannot be 'obsolete'. Moreover, the removal of data, which is reported for the first time in the literature demonstrating the effect of coating on PCF mechanism, will likely contribute less to the progress of the field.

**Stochastic vs. deterministic description of ice nucleation data:** Sect. 3.2: The ns-based interpretation of the data is inconsistent with the interpretation of the authors, that PCF is the mechanism with which soot nucleates ice. While for the ns concept, the ice nucleation is triggered by so-called active sites, describing a _heterogeneous_ ice nucleation process, the PCF mechanism can purely be described with CNT: Capillary condensation taking place in pores and subsequent _homogeneous_ freezing of this pore water. In other words, the ns-based approach is invalid for homogeneous freezing. The whole discussion about the two concepts of stochastic vs. deterministic description of ice nucleation as presented in the manuscript (P3L7-15, P11L28-P12L11) is superficial and does not fit the framework of the paper. Since the authors claim a PCF mechanism, I suggest to remove any discussion of ns from the manuscript for consistency. At the same time, this avoids discussion of any uncertainties introduced through the estimation of BC effective surface area (P7L6), which lacks a proper discussion of assumptions made.

We agree with the Reviewer that discussing the results of ice nucleation on insoluble black carbon particles is highly complicated, given that there is not an explicit theory that both explains the results and can be currently experimentally verified. It is highly possible that our discussions are inadequate and we will rely on the Reviewer's comments to help strengthen our discussions.

That said, we respectfully disagree with the Reviewer on several points here: (1) the Reviewer claims PCF mechanism "can be purely described with CNT"; (2) "the $n_s$-based approach is invalid for homogeneous freezing", whereby the Reviewer is equating the PCF mechanism with homogenous freezing; and, (3) "The whole discussion about the two concepts of stochastic vs. deterministic description of ice nucleation as presented in the manuscript (…) is superficial and does not fit the framework of the paper." While these discussions are important in general, we stress that our work is primarily experimental in nature and that this work is not the appropriate platform for detailed discussions of theoretical frameworks.

Explicitly, we offer the following to indicate why we disagree with the Reviewer on the several points mentioned above. (1) CNT does not account for the required pore dimensions nor surface chemistries that are inherent in the PCF mechanism, therefore PCF cannot be a purely CNT-based mechanism. (2) The PCF mechanism explicitly requires pore sizes on the order of ~3-10 nm in diameter; thus, PCF efficiency could be discussed in terms of the number of correctly sized pores for a given insoluble particle, which is very similar (though to be sure, not exactly the same) as discussing the number of active surface sites for a given insoluble particle. In either case, both require an insoluble surface of some kind that initiates ice formation at explicit active sites. (3) We feel that since our results are time-dependent measurements, we should note this in our discussions. Further, since much of the other laboratory results for heterogeneous ice nucleation on black carbon particles have been discussed in terms of active site densities, we feel that it is appropriate to also place our results within this context. Of course, how well we have accomplished this for our current active site density discussion is another matter that we address below.

Our manuscript is first and foremost an experimental report that describes five (5) related experiments exploring the IN activity of black carbon particles using the SPIN instrument. As the SPIN instrument is an on-line ice nucleation measurement, the results from this technique are inherently time-dependent. The fundamental measurement from SPIN is the active fraction (AF) or the ratio of ice nucleating particles (INP) to the total number of particles ($N_{total}$) sampled for a given residence time (~10 s) at a given center-line temperature and center-line water vapor concentration (and hence supersaturation with respect to ice). We present the critical wave vapor concentration and temperature from these measurements when the AF ratio crosses the 1% (i.e., 0.01) threshold. Thus, it is relevant to discuss a stochastic description of our results (i.e., time-dependence). We could use our results to provide a measured freezing rate (i.e., number of ice particles formed per time), and we provide the information for others to do so. However, without more explicit information about the specific microphysical and chemical properties of our BC particle surfaces (i.e., where ice nucleates), it is not clear how to directly compare these time-dependent results with measurements done using significantly different techniques, such as cold-stage or chamber experiments.

On the other hand, if we follow the approach taken by many other laboratory studies and present our results in the context of an active site density, we can directly compare our results with previous published results. We have chosen to do this and we feel that it is both appropriate and useful. By doing this, we do not, in any way, diminish the potential importance or usefulness of the PCF mechanism.

We tend to agree with the Reviewer that our discussions border on the qualitative (which is not the same as superficial). Given the state of the science concerning ice nucleation and our limited resources, we are doing the best we can. (Please refer the Knopf et al. 2018 review of the role of organic aerosols in atmospheric ice nucleation, in general and the section on "Analysis of ice nucleation data" specifically, and references therein, as a rational summary of some of these uncertainties).

Finally, we agree that our current discussion on surface area calculations was insufficient. We have added a paragraph in the Supplement, which describes the calculations of the effective surface area and the assumptions we have used.

**Specific comments:**

P1L1: The title should be revised as it is misleading and inconsistent with the major finding of the authors on P13L12-14. The freezing of pore water is homogeneous and not heterogeneous as implied by the title.

We respectfully disagree. The title "Laboratory study of the heterogeneous ice nucleation on black carbon containing aerosol" describes, in a general manner, all the aspects of this study, including coatings. Additionally, in the proposal for ice nucleation terminology (Vali et al., 2015), heterogeneous nucleation is defined as "Ice nucleation aided by the presence of a foreign substance so that nucleation takes place at lesser supersaturation or supercooling than is required for homogeneous ice nucleation". The foreign BC particles provide the porous structure, which allows nucleation at a lesser supersaturation, which would not occur without the foreign structure. Lastly, the use of 'heterogeneous' in the title confines and clarifies the temperature and supersaturation ranges of this study. If 'homogeneous' is used instead, the added value to the clarity of the title becomes debatable.

P1L18: I suggest to move the sentence "The current study focusses on laboratory measurements…" at the end of p.1 l.23 to improve the flow of the reading, first discussing field study observations, then laboratory studies and finally say that your current study focusses on laboratory measurements.

Moved.

P1L22: add "… can be highly active IN under certain conditions."

Added.

P1L24: the addition "… commonly understood to be deposition mode ice nucleation." is not true. Other ice formation mechanisms, e.g. homogeneous freezing, can also take place within the cirrus temperature regime. You should delete that part of the sentence, or specify that you refer to RHw conditions below homogeneous freezing conditions.

Changed to: "We examine ice nucleation on BC particles under water-subsaturated cirrus cloud conditions, commonly understood to be deposition mode ice nucleation."

P1L25: Why are you using "carbon black" and not "black carbon (BC)"? I suggest being consistent throughout the manuscript.

Carbon blacks are manufactured materials engineered for specific properties and are typically produced as industrial black pigments (Section 2.1), whereas soot and BC particles are generally defined as the by-product of incomplete combustion of fossil fuels, biomass, and biofuels (P1L15). The terminology of carbon black, black carbon, soot, and elemental carbon is often confused or misused (Long et al., 2013; Petzold et al., 2013). Unfortunately, in this kind of research it is unavoidable. However, the right terminology should be used when addressing different types of particles. We have chosen to follow the recommendations described by Petzold et al. (2013) as closely as possible.  Please refer to our response to P2L7 for our definitions of BC and soot.

P1L34: change dependence to depends

Changed to dependencies.

"The measured IN activity dependencies on temperature and the physicochemical properties of the BC particles are consistent with an ice nucleation mechanism of pore condensation followed by freezing".

P2L2: Please specify: "low ice supersaturations".

Changed.

P2L5: Your usage of "INP" differs from your definition on P1L21, where you describe ice nucleation particles as "IN", which in turn is inconsistent with your usage of "IN" on P2L7. I suggest to stick to the terminology presented in Vali et al. (2015) and consistently use INP throughout the text.

Text amended. We now use INP for ice nucleating particles and IN for ice nucleation, consistent with Vali et al. (2015).

P2L7: Do you consider "soot" and "BC" to be the same? You should clarify this and give a definition on how you use the term soot in your manuscript and then stick to this for consistency.

As stated earlier, we have chosen to follow the recommendations described by Petzold et al. (2013) as closely as possible.  In general, soot particles are incomplete combustion products, a subset of which contain an elemental carbon-based, refractory, light-absorbing component termed black carbon. Therefore, some soot particles, including cases described in the introduction, may consist of many chemical components that have little black carbon content (e.g. chars, coke-oven emissions, Brown Carbon). In this study, we used ethylene combustion soot and manufactured carbon black particles, which consist of black carbon particles without organic matter.  These are BC particles.  We also coat some of these particles with specific compounds, generating BC-containing particles.  Since 'BC-containing' particles is a more inclusive terminology, we find it more appropriate to use it in the title of the manuscript and in several places in the manuscript.

We define soot on P3L32. For clarity, we've added a sentence to explain this point in the text:

"Soot and black carbon nomenclature is often used interchangeably for particles with negligible organic matter content".

P2L15 Change "ice nucleation particles" to INP.

Changed.

P2L19: Please specify what samples were studied in Pratt et al. (2009) and Eriksen-Hammer et al. (2018). Did they probe samples similar to Chen et al. (2018)?

Changed to:

"Low BC IN activity was also found by Pratt et al. (2009) in their analysis of airborne particle residuals in orographic wave clouds, and by Eriksen-Hammer et al. (2018) in mixed phase clouds at the high-altitude research station Jungfraujoch."

P2L20-22: Please specify for which cloud types soot is considered an important INP for the different studies. All clouds or just cirrus clouds?

Phillips et al. (2013) discussed the empirical parameterization of heterogeneous ice nucleation and its applicability in various cloud scenarios. They mostly discussed shallow wave clouds but mention other temperature regimes as well. They concluded their discussion with a general statement that black carbon is a major type of ice nucleus in clouds influenced by biomass-burning particles.

Levin et al. (2014) operated their CFDC at −30°C and 105% RHw. However, in their conclusion, they generalized their results stating that "these results indicate that fires could be a significant source of INP to clouds, at least locally".

We changed the text to be reflect these statements:

"Phillips et al. (2013) suggested that black carbon is a major type of INP in clouds influenced by biomass-burning particles, however they couldn't rule out that another INP species, internally mixed with soot, might have nucleated the observed ice. Likewise, Levin et al. (2014) concluded that fires could be a significant source of INP in mixed-phase clouds."

P2L22-27: "While in these…" I suggest removing this paragraph from the manuscript to improve readability and streamline the introduction. It does not contribute to the general understanding of ice nucleation ability of soot particles and thus to the scope of the presented manuscript, which is not on ice-multiplication processes.

Removed.

P2L29: I suggest to remove the reference to China et al. (2015)a. This study does not investigate ice nucleation activity of soot particles, but quantifies a morphological change of these particles upon ice formation (and cloud droplet formation) on the soot particles and the consequences for the radiative impacts of the soot particles.

Removed.

P2L30: Kärcher and Lohmann (2013) is not a laboratory study. This reference is miss placed here and thus should be deleted.

Removed.

P2L32: Change to aircrafts.

Aircraft is also the plural form.

P2L36: Your usage of the Kärcher et al. (2007) study to motivate the absence of a contribution of soot to ice nucleation is delusive, as you use the same study on P2L33 to say the opposite. Please clarify.

We removed Kärcher et al. (2007) from P2L36 to avoid confusion.

P3L3: Your citation does not support your statement that "numerous soot types have shown high IN activity" when coated. Crawford et al. (2011) investigated soot derived from a propane burner (miniCAST and CAST). In this case the burning conditions (air to fuel ratio) can be changed to mimic soots with different organic carbon content, also all soots are from the same source. Is that what you refer to as "numerous soot types"? Please clarify.

Changed:

"Further, Crawford et al. (2011) showed that propane burner soot can be a highly active INP in the deposition mode if it has a sufficiently low organic carbon content (i.e., uncoated)."

P3L4: Add this reference: Möhler et al. (2005)

Added.

P3L7: What do you mean with "quantitative relationships"? Relating physicochemical properties of soot particles to their ice nucleation activity? Please clarify.

In the sentence, "Modeling INPs requires quantitative relationships that governs the IN activity," we mean that in order to model the relationships between physicochemical properties and IN activity a mathematical expression must be established and used in the model.

We will replace the sentence with: "Modeling INPs requires quantitative parameterization of the IN activity. Two approaches to explain and parametrize the IN observations are commonly used:

a stochastic description based on classical nucleation theory and a deterministic or singular description (Knopf et al, 2018)".

P3L10: This statement is only true for activated fractions less than 0.1 as shown in e.g. Niemand et al. (2012),Hiranuma et al. (2014). Please change your statement accordingly.

Changed to: "As demonstrated in Niemand et al. (2012), if the activated ice fraction is small (< 0.1), the active site density is expressed as the fraction of the ice particles out of the total aerosol concentration divided by the averaged particle surface area."

P3L12: Do you want to compare measurements to modelling results or do you want to use ns parametrizations derived from measurements for modelling purposes.

We want to compare the observations from independent experiments and note that these can be parametrized and used for modelling purposes.

We changed the sentence to: "This approach describes the density of active sites (ns) as a function of temperature, allowing for intercomparison between independent observations and subsequent parametrization for modeling purposes, …"

P3L13: None of the given references preliminary discusses the limitations of the deterministic description of ice nucleation. I suggest deleting these here. In case you like to give an example for time dependent ice nucleation, e.g. use: Welti et al. (2012).

We changed the order of the references in the sentence and added Welti et al. (2012):

"This approach describes the density of active sites (ns) as a function of temperature (e.g. Marcolli, 2014; Wagner et al 2016; Ullrich et al 2017; Kanji et al 2017; Kiselev et al. 2017; Campbell 2017), allowing for intercomparison between independent observations and subsequent parametrization for modeling purposes, but does not take into account the kinetics (i.e. time dependence) of nucleation (Welti et al., 2012)."

P3L18-21: "Porous materials…" I suggest moving these sentences after the explanation of the PCF mechanism given in the next paragraph.

Moved to the end of the PCF paragraph.

P3L20: Please add the following references: Alstadt et al. (2017), Mahrt et al. (2018) that are very relevant to the work discussed in the current manuscript.

We agree that Mahrt et al. (2018) fits in the context of this study; however, Alstadt et al. (2017) is an immersion-freezing study for carbon nanotubes (CNTs) and therefore does not intuitively fit into this list of studies of heterogeneous soot nucleation at temperatures below 235 K.

We have changed the sentence to read: "Therefore, the PCF mechanism may be applicable to soot particles as well (Marcolli (2014, 2017); Wagner et al. (2016); Ullrich et al. (2017); Mahrt et al. (2018))."

P3L22: Change to: "… fill with water due to capillary condensation at relative humidities (RHw) below water saturation, which freezes homogeneously (Marcolli 2014).".

Changed.

P3L22: You might want to give a more well rounded list of references describing the PCF mechanism, including e.g. Christenson (2013); Higuchi and Fukuta (1966).

Added.

P3L23: PCF is not only determined by the pore diameter, but also by the contact angle, see for instance Marcolli (2016).

We have changed the sentence to read: "The diameter of the pore and the surface properties of the pore substrate affect the water condensation process."

P3L23 delete "(< 100% RHw)" it is sufficient to say below water saturation.

Deleted.

P3L22-29: The discussion about "suitable" pore sizes for PCF stays superficial. What does suitable mean? Is it pores with diameters in the range of μm or nm? I suggest taking a contact angle typical for (atmospheric) soot and calculating some critical pore diameters, i.e. diameters at which a pore would fill with water (at a typical cirrus temperature), in order to support your argument and then relate these pore sizes to those typically observed in soot aggregates.

The relevant or "suitable" pore sizes are 2-50 nm mesopores, especially pores less than 10 nm in diameter where the Kelvin effect is greatest. Both combustion-related soot particles and manufactured carbon particles can exhibit pores in this size range. We have changed the relevant text to read:

"… in accord with the inverse Kelvin equation (Marcolli, 2014). The diameter of mesopores (2-50 nm) and the surface properties of the pore substrate affect the condensation process in the particles, especially for pores less than 10 nm in diameter where the Kelvin effect is greatest. In large diameter pores (>> 10 nm), the water vapor pressure is not sufficient to cause condensation below water saturation (< 100 % RHw). On the other hand, in pores with diameters too small (<4 nm), the growth of an ice embryo may be inhibited (Marcolli, 2014). Pore diameters in soot materials range from micro (<2 nm pore diameter), through meso (2-50 nm diameter), to macro (>50 nm diameter) and are dependent on the specific soot material. Manufactured carbon black material (e.g., Kruk et al., 1996) can be produced with similar or higher surface areas and mesoporosity than combustion-related soot particles (e.g., Rockne et al., 2000). All else being

equal, IN activity of a material is expected to increase with increasing number of mesopores in the suitable diameter range (~10 nm). The PCF mechanism also predicts the observed decrease in ice nucleation activity with increasing temperature over the temperature range about 210 K to 240K (typical of cirrus clouds) (Hoose and Möhler, 2012)."

P3L29-31: Please specify how the PCF mechanism predicts the observed decrease in ice nucleation activity with increasing temperature between 210-240 K.

We have changed the text to read: "The PCF mechanism has been discussed with respect to the observed decrease in ice nucleation activity with increasing temperature over the temperature range about 210 K to 240K (typical of cirrus clouds) (Ullrich et al., 2017)."

P3L30: Change to "… (typical of cirrus clouds, Hoose and Möhler 2012)."

Changed.

P3L36: Marcolli et al. (2014): I cannot find this citation in your reference list.

Changed to Marcolli (2014).

P4L2: Change "spheres" to "spherules" to be consistent with terminology on P4L1.

Changed.

P4L4: "In the image…" Delete this sentence here and put is to the caption of the figure.

Moved.

P4L6: I disagree with this statement. There can be slit-like pores formed in between sintered primary spherules. Since your "external branch" in Fig. 1c) consist of multiple primary spherules, it can have pores and thus contribute to PCF.

We have changed the text to read:  "In this study, we will refer to any confined empty spaces between aggregates or spherules as pores."

P4L17-18: This statement is not consistent with your statement on P4L7, where you define a pore as an "empty space between aggregates". Please clarify.

Clarified by P4L6 change to text.

P4L27: The way you use the reference of China et al. (2015a) is misleading, as it is not about the impact of aerosol generation method on the soot morphology. Please rephrase.

We have removed the China et al. (2015a) reference from this sentence.

P3L34: Is only the number density of pores changed between the large BC agglomerates and the small BC agglomerates or also the pore size distribution? At a similar pore size distribution, the ice nucleation ability, when caused by PCF, should be identical between the large and small agglomerates, as the pore filling is just a function of pore diameter (and contact angle). Please clarify.

The Reviewer is correct that we do not know whether the pore size distributions or number concentrations varies as a function of carbon black particle size. We have modified this sentence to read: "To test the role of particle size in the IN process, we compared IN activity of large BC agglomerates with inner pores between aggregates (Fig. 1c) to IN activity of smaller size selected aggregates (Fig. 1b) with similar surface chemistry and particle morphology but potentially different pores sizes and numbers."

P5L16-19: The numbering of the soot types in the main text is confusing. I suggest indicating the supplier directly within the table and referring to the sample names throughout the text to be consistent.

We use the carbon black sample names throughout the text to refer to each specific sample, rather than a numbering system. To make this clear, we have changed the text to read: "The first five of these materials listed sequentially in Table 1 are commercial carbon blacks, a form of elemental carbon obtained from the incomplete combustion of organics (typically liquid hydrocarbons) under controlled conditions. The "Regal" and "Monarch" samples were supplied by Cabot Corporation and Raven 2500 Ultra was manufactured by Birla Carbon. Raven 2500 Ultra was chosen for its relatively large specific surface area (high BET) value."

The rational for not including the manufacturers name in the table is that we would be adding a column to the table with two entries, making the formatting of the table more complicated to little benefit. The manufacturers' names are well described in the text.

P5L17: The statement about the Raven 2500 Ultra seems weird, as the Monarch 880 and Monarch 900 have very comparable BET surface areas.

Not clear what changes the Reviewer is suggesting here. No changes made.

P5L20: Delete "… in the deposition mode regime." This is confusing with the point the authors try to convey, that soot nucleates ice via PCF.

Changed to read: "These uniform commercial powders with known physical properties allow a systematic screening of selected particle properties important for ice nucleation in the atmospherically relevant 210-230 K temperature regime."

P5L27: Delete comma after Table 1.

Deleted.

P5L27: Move explanation for OAN from L30 to here.

Changed to read: "The Braunauer-Emmett-Teller (BET) and Oil Absorption Number (OAN) values shown in Table 1, were provided by the manufacturers."

P5L35: Please specify what this oxidation process encompassed.

We do not know. Unfortunately, the full production processes of commercial carbon blacks are protected intellectual property and are not readily available to the public. We can only speculate about the steps taken to oxidize the BC surfaces (e.g. post processing or in the process of formation). The information we were able to obtain with the available instruments is presented in this paper.

P5L13: I suggest removing the references to "issue xy in the Introduction" here and at the other location sued. This seems unnecessary and repetitive. If you want to keep these please use proper Section numbering when referring to different parts of your manuscript, e.g. "as outlined in Sect. 1", consistent with the style of ACP ("Manuscript composition"): https://www.atmospheric-chemistry-and-physics.net/for_authors/manuscript_preparation.html

Removed where unnecessary.

P6L18-20: Why did you choose these acids for coating? Please briefly discuss their atmospheric abundance and relevance.

To address the Reviewer's question, we added the following text:

"Stearic acid ($C_{18}H_{36}O_2$) is one of the abundant saturated fatty acids and it is a common constituent of atmospheric particles in urban areas that cook large amounts of meat (Katrib et al., 2005). Stearic-acid coated particles are a gross simplification of atmospheric particles since urban aerosol particles are composed of hundreds if not thousands of organic molecules. However, these particles could serve as a proxy of the broader class of soot coated with fatty acids.

Humic-like substances are very efficient surfactants. One of the commonly used model surfactants is cis-pinonic acid ($C_{10}H_{16}O_3$). This compound originates from boreal forests, and blooming algae in the oceans yielded from the photochemical oxidation of the evaporated α-pinene in the lower troposphere (Luo and Yu, 2010).

Dicarboxylic acids are another important group of organic compounds identified in the atmospheric aerosols. Their contribution to the total particulate carbon ranges from about 1-3% in the urban and semi-urban areas to values close to or even above 10% in the remote marine environment (Kerminen et al., 2000). Dicarboxylic acids have several different sources, including primary emissions from fossil fuel combustion and biomass burning (Chebbi and Carlier, 1996). Here, we examine coatings with oxalic acid ($C_2H_2O_4$), an abundant dicarboxylic

acid in the lower troposphere, comprising a significant fraction of the total diacid mass concentration (Kerminen et al., 2000)."

P6L20: Do you want to investigate the effect of coating of the effect or organic carbon content?

Changed to: "In a study of the effect of coating by organic carbon content on IN, BC particles were coated either with stearic acid (SA) (>99 % purity, Aldrich), cis-pinonic acid (98 % purity, Aldrich), or oxalic acid (>99 % purity, Aldrich)."

P6L20: Add comma after heated

Added.

P6L24: Change to "… coatings consisting of super-cooled aqueous solutions can become crystalline or glassy solids (references)."

This is not what we indicated in this sentence. In a generalized overview of the experiments, all three states of coatings are possible i.e. supercooled aqueous solution, crystalline and glassy solid.

"At the temperature and relative humidity conditions of the ice nucleation experiments, coatings may exist as super-cooled aqueous solutions, as crystalline, or as glassy solids (Hearn and Smith, 2005; Knopf, 2018)."

P6L26: Add: Murray (2008); Zobrist et al. (2008)

Added.

P6L34: Delete "and mixtures"

Removed.

P7L1: Change to: "…single particle mass spectrometry instruments"

Changed.

P7L1: Change to : "Therefore, hundreds of…"

Changed.

P7L1: Please clarify whether PALMS was operated on DMA size selected particles in the text. This should also be clarified in the caption of Fig. A1. From Fig. 2 it looks like PALMS was

operated on size selected aerosol. Please comment whether there is a chemical difference between your 100 nm and 800 nm aerosol particles.

We were unable to accurately measure the chemical composition of the 100 nm DMA-selected BC particles with the PALMS instrument due to their small sizes. The lower particle size threshold for PALMS is ∼200 nm diameter and is set by the amount of detectable scattered light. The uppersize threshold is set by transmission in the aerodynamic lens at ∼3 μm diameter (Zawadowicz et al., 2017).

We changed the sentence on P6L28 to read:

"The DMA-selected (800 nm) BC particles were characterized for chemical composition by the Particle Analysis by Laser Mass Spectrometry (PALMS) instrument, …"

We added "size-selected" to the caption of Fig. A1.

P7L3: From your Fig. 2 this looks like lacey carbon copper grids. Please specify and give manufacturer details.

Good catch! We changed the text to: "In addition, BC particles of 800 nm mobility diameter were collected on 300-mesh copper lacy formvar grids (Ted Pella, Inc.)…"

P7L4: Please be quantitative: How many SEM images were taken for each BC type? What fraction out of all images taken showed a "clustered, spheroidal structure"? This should also be indicated in the caption of Fig. A2.

We analyzed between 40-50 particles per SEM grid per dry dispersed 800 nm size-selected BC type for Regal 400R, R2500U, Monarch 900, and several ethylene soot particles.. Unfortunately, due to limited resources, we were unable to obtain SEM images of all of our samples. On average, 50% of all of the particle types were classified as highly compact (i.e., nearly spherical) clusters (20-95%); the remaining particles were classified as compact (i.e., prolate) clusters. For all samples, the clusters were compact, with little observed internal spaces. There a possibility that prolonged duration of collection on the grid has a structural impact on the captured agglomerates.

We have changed the text to read: "While BC aggregates are often considered highly branched (Fig. 1a), the microscopic images of the collected 800 nm size-selected agglomerates generated by dry dispersion show compact clusters. Based on analysis of 40-50 particles per BC particle type, approximately 50% of the particles were classified as highly compact, near spherical; the remaining were classified as compact, prolate clusters. For analysis purposes, we assume these clusters to be highly compact spherical particles. This extrapolation to the whole agglomerate population will induce potential statistical error and will define the upper limit of the effective surface area calculations."

P7L9: For discussion of eq. (1), please see my comments P2L10.

We assume the Reviewer refers to P3L10. We added: "….where Af(T) is the activated fraction at a given temperature (A$_f$(T) < 0.1)…."

P7L6: Please clarify how the SEM images were used to estimate the effective surface area. I suggest putting the equations used for your calculation into a supplementary information, which should also discuss the assumptions made by you. For instance, from your SEM images in Fig. 1 and A2 it looks like a strong overlap of the individual spherules/monomers, which would lead to an underestimation of the total number of spherules making up an aggregate and with that to an underestimation of the effective surface area.

We agree with the Reviewer that more information is required.  We analyzed SEM images of R2500U, ethylene soot, and Monarch 900.  In SEM images of 800 nm mobility diameter selected BC particles appeared to have highly compact sphere-like shapes. We estimated their surface area using the following method:  First, we assumed the 800 nm mobility diameter can be represented by a spherical shape for the per particle agglomerate. Second, we use an average primary spherule size determined from the SEM images for each specific BC particle type. Third, we diced the 800 nm spherical particle up into "shells" defined by the thickness of the average primary spherule.  We added up the number, and thus surface area, of all of the primary spherules in the top or outer "shell".  We assumed that this represents the total surface area of the particle and accounts for overlap and shielding by not including inner "shells".  The difference between this estimate and the estimate of the per particle surface area determined by summing the number (surface area) for all of the primary spherules in a particle is less than a factor of 3 for the spherule diameters we have used..

We have added the requested information to the Supplement Fig. S2 and explanation therein.

P7L7: Replace "frozen fraction" by "activated fraction"

Changed.

P7L10: Is Lf a SPIN specific correction factor?

Yes, it is obtained in the calibration of the laminar flow in the instrument. We have changed the text to read:  "… Lf is a SPIN-specific correction factor obtained in flow calibrations, …"

P7L11: Why do you only consider the "outer shell" of the aggregate? This seems inconsistent with suggesting a PCF mechanism for ice nucleation. Pores are clearly formed also within the clustered structure.

The inclusion of the inner shells does not significantly vary the result. It was calculated for our cases to change the results up to a factor of 3. As explained in the added supplement.

P7L14: It is not clear to me where the factor 3 results from.

We have clarified this point in our added explanation of how we estimated the per particle surface areas in our response to P7L6 (above) and in the added explanation in SI.

P7L16: Please change to: "(see Sect. 3)."

Changed.

P7L20: Delete "i.e."

Deleted.

P7L24-26: This statement is incorrect. I assume you are just describing the cirrus temperature (T<235 K) regime. Homogeneous freezing of solutes takes place at water subsaturated conditions. Please phrase your statement more carefully.

We changed to "…allowing for ice nucleation."

P7L29: Several micrometer in diameter?

Added "… in diameter".

P7L32: This statement is inconsistent with the suggestion of the authors that BC particles nucleate ice via a PCF mechanism for cirrus temperatures. In case PCF takes place the water is taken up into pores at water subsaturated conditions due to capillary condensation. This pore water subsequently freezes homogeneously and not heterogeneously, at T < 235 K.

We respectfully disagree. See our response to P1L1 above.

P7L31-33: This statement should be followed by a reference.

Added (Krämer et al., 2016).

P7L36: Change "centerline" to "lamina"

Changed.

P8L17-19: Why is this "time dependent"? This is purely a matter of how you interpret your results… This statement should be deleted.

As particles travel through the SPIN instrument along the center line, they cool to the SPIN temperature and are subjected to water vapor at a set of temperature conditions that is subsaturated with respect to water, but supersaturated with respect to ice. These particles can activate as ice nucleation particles (INP), or not. If they do activate, then they start to grow in

size due to vapor to ice condensation. As they grow, they remove water vapor from the centerline. The ultimate number and size of the ice particles measured by the OPC at the outlet of the SPIN instrument, is a function of the ice nucleating properties of the particles and the time-dependencies of the SPIN technique. The kinetics of the technique limits the range of freezing rates measurable by the SPIN, independent of the nucleation rate of the INP. Details can be found in Garimella et al. (2017)."

P8L16: Abbreviate "Figure" with "Fig." throughout the manuscript; see ACP guidelines ("Manuscript composition"): https://www.atmospheric-chemistry-and-physics.net/for_authors/manuscript_preparation.html

Changed.

P8L23: Change "IN agent" to "INP".

Changed.

P8L25-27: "Both types demonstrated temperature dependence of ice onset similar to …, associated with the PCF mechanism, that is, increasing ice onset point with increasing temperature in the range 217-235 K".

- Pease clarify how you arrive at this conclusion and justify why the observed increase leads you to the conclusion that PCF must be responsible for ice formation.

  As we have pointed out, our manuscript primarily reports on our laboratory observations. In discussing these observations and attempting to place our observations within the context of previous experiments, we have referenced previous work on ice nucleation on BC particles, including Hoose and Möhler (2012 Sect. 5.2.3) and Ullrich et al. (2017 Sect. 3.b.3). It is from these sources that we note that the increasing ice onset point with increasing temperature in the range 217-235 K has been associated with the PCF mechanism.

  To clarify our wording, we have changed the text to read: "Both types demonstrated a temperature dependence of ice onset, increasing ice onset point with increasing temperature in the range 217-235 K, similar to some of the earlier observations of soot (e.g., Ullrich et al., 2017; Hoose and Möhler, 2012; Bond et al., 2013). These previous reports suggested that PCF mechanisms based on water-condensation prior to ice nucleation, and not classical deposition nucleation mechanisms (i.e., vapor to solid), may account for these observations."

- Assuming that PCF in fact causes the ice nucleation of your BC particles, the freezing is purely determined by the pore size and the soot-water contact angle. Thus for a given/fixed pore size and contact angle, your ice formation is purely deterministic and should take place, as soon as the critical RHw for pore filling is reached during your RHw scan. In other words, the freezing should take place at the same RHw throughout all temperatures T<235 K, i.e. along an RHw-isoline. This is inconsistent with your results

and your argumentation. I do believe that PCF indeed takes place for your samples, but I think that the increase in onset RH towards 235 K, is due to a lower homogeneous freezing rate at this T compared to T << 235 K.

The Reviewer appears to be making a claim that the IN activity we observe between 217-235 K, increased ice onset with increasing temperature, is in fact due to a PCF mechanism that is controlled by a lowering freezing rate of the water condensed in pores at increasing temperatures. We would be more than happy to include this theory in our discussions if we were provided with a relevant citation. No changes made.

- This could be easily proofed by running some more experiments for R2500U and Regal 330R at T<225 K. In case of PCF, you should observe a similar levelling-off in onset RHw as observed for the ethylene combustion soot. I suggest adding some more experiments at these very low temperatures, to support your argument.

  We agree with the Reviewer and these experiments are on our to-do list! That said, given our resources available for this work, we had to prioritize and were unable to make these measurements as part of this work. At the lowest temperatures, it is more difficult to reach stable conditions for long periods of runs with the SPIN instrument. This is one of the reasons for the low number of points at the lowest temperature, moreover there was no goal to reach lowest temperatures which have higher uncertainty. No changes made.

  We will point out that the SPIN instrument appears to be an appropriate measurement technique for generating IN activity (here presented as ice onset defined at 1% of AF ratio) versus temperature for the same type of particles. If the Reviewer is correct that the increase in IN activity with decreasing temperature is due to a changing freezing rate inside pores, rather than a change in the number of size of pores that could activate ice, these curves could be considered to be isolines of constant active site density. Figure 9 makes this case directly.

P8L28: See my general comment. In case of PCF, the freezing of the water is homogeneous.

We respectfully disagree. See our response to P1L1 above.

P8L32: I suggest to tune down "significant"

Removed.

P8L32: Please reference "in the Introduction" as "in Sect. 1".

Changed to "In Sect. 1, …".

P9L2: I suggest to just using the abbreviation "R2500U" within the text, to be consistent with the legend and improve readability.

Abbreviation now introduced in Sect. 2.1 and then it is referred to as R2500U.

P9L8: Can you quantify "high RH"? Also, please use "RHw" for consistency.

Changed to "$RH_w = 83\%$".

P9L10: The whole discussion is qualitative and seems speculative. Can the authors provide information about pore sizes and pore size distribution associated with the individual soot types?

Pore size and pore distributions were not directly measured in this study. Without these capabilities, we rely on the manufacture's information on BET and OAN measurements to provide insights into potential changes in pore size and number concentrations for each BC particle type sampled. Using these parameters, the relationship between BC surface area and ice nucleation from our measurements are discussed within the context of the cited literature. No changes made.

P9L16: Change to: "… a weaker inverse Kelvin effect and therefore can only fill with water at higher RHw."

Changed.

P9L13-19: Is the higher degree of branching of the Monarch 880 soot (higher OAN number) supported by your SEM analysis? Please comment and provide images to support your argument.

We are unable to support the higher Monarch 880 OAN numbers with direct SEM images, due to the limited number of SEM images (we analyzed ~40 particles per BC type) and limited resources. That said, we are very clear that for this study, we chose the carbon black particle types based on the manufacturer's information and we use this information to the furthest extent we can in our analysis and discussions.

P9L17-20: Most of your Monarch 880 data points lie within uncertainty of the homogeneous freezing curve indicated in your figure, which makes a statement about PCF speculative. At the same time, your Monarch 880 experiments are constrained to T > 230 K. This makes it almost impossible to infer a clear trend, which you try to impose with your colored lines, and compare the ice nucleation trends of the individual soot types. I suggest to extend your measurements for Monarch 880, 900, Regal 330R and R2500U to 217 K, similar to the ethylene soot, to allow for a more complete comparison among the different soot types.

We agree with the Reviewer that, given enough resources and time, we would readily extend our measurements to provide a more complete picture. However, given the resources we had, we feel that our current data set is useful and worth publishing so that the community, as a whole, can integrate these results with past and future results to further develop our knowledge of BC particle IN activities.

As we note in the caption of Figure 4, the solid lines are there to guide the eye so that readers can see the apparent trends in the IN activity as a function of temperature. These lines for each BC particle type, while they differ in slope and extent, all show similar trends in Figure 4.

P9L22: Can you quantify the variability observed in the measured effective density? I assume the values reported in Tab. 1 are mean values. Please indicate the number of particles measured and give the standard deviation, so that the reader can access the degree of variability within each BC type, which then allows to intercompare the different types.

We have added the following information to the subnote "d" in Table 1:

"$^d\rho_{eff}$ is the effective density calculated from the ratio of the vacuum aerodynamic diameter ($d_{va}$) measured by the PALMS (Particle-Analysis-by-Laser-Mass-Spectrometry) instrument and the constant mobility diameter (Dm) of 800 nm, multiplied by the standard density of 1 g cm$^{-3}$. The statistical variability in the calculated $\rho_{eff}$ values are within 1%, based on PALMS $d_{va}$ measurements of > 1000 particles per sample."

P9L24: The statement about the aerodynamic diameter and its relation to mass and shape factor are confusing. Do you not derive the effective density from the *vacuum* aerodynamic diameter and the mobility diameter? Please clarify.

The effective density in Table 1 is derived from the ratio of the vacuum aerodynamic diameter and the mobility diameter, as the Reviewer notes. This effective density is the type III density described in Decarlo et al. (2004). We have changed the text to read: "The aerodynamic diameter of the agglomerates is related to their shape factor (DeCarlo et al., 2004; Jayne et al., 2000)."

P9L27: Add "… can likely be explained"

Added.

P9L28-31: This statement is not supported by your data, or I am misinterpreting it. For instance, ethylene soot has the lowest effective density, but around 225 K shows a similar ice nucleation ability to R2500U, which has the second highest effective density. Please explain. In addition, your argumentation is inconsistent with the effective density given for Regal 400R, which you argue is not PCF active at all. This should be clarified. Alternatively, are you trying to argue that within one soot type, aggregates with higher effective density nucleate ice more readily by PCF compared to aggregates of the same soot type, but which have a lower effective density and that this can for instance explain the variability of your R2500U data points around 230 K? If the latter is the case, this is an interpretation of your data and should be clearly stated as such, as you do not show any measurements comparing compact, high effective density soot particles of a given diameter against low effective density soot particles of the same diameter and soot type, which would support your argument.

We apologize for our lack of specificity. Our statement here is about how different BC particle types exhibit different IN activities that appear to be related to their measured effective densities. These observations relate to the temperature range of > 230 K, where the critical supersaturations at IN threshold formation (here 1% of all particles) increases with increasing temperature. There are two ways to see this. First, at 230 K, the measured critical supersaturations are R2500U < Monarch 900 < Monarch 880 ~ Regal 330R < Ethylene soot, whereas the measured effective densities in Table 1 are R2500U > Monarch 900 > Monarch 880 ~ Regal 330R > Ethylene soot. Thus, it is clear that there is an apparent inverse relationship here (see plot below).

[Figure]

The other way to see this correlation is to note the temperature at which the measured critical supersaturations for the different BC particle types intersect with the homogeneous freezing line (see plot below).

[Figure]

As noted by the Reviewer, it would be very interesting to see if these correlations hold up at lower temperatures or if they change as a function of temperature. These questions will have to be answered in future studies.

We have changed the text to read: "The variability in IN activity for temperatures greater than ~230 K (Figure 4), where the measured critical supersaturations at IN threshold formation (1% of all particles) increases with increasing temperature, appears to be inversely correlated with their measured effective densities (Table 1). At the 230 K isotherm, the critical supersaturations are inversely correlated with the effective densities. This same relationship is also observed in the temperature at which the measured critical supersaturations for the different BC particle types intersect with the homogeneous freezing line. These observations suggest that for temperatures at or greater than 230 K, more effective BC particle INPs have higher effective

densities (i.e., are more compact particles), implying that IN active pore sizes may be related to particle effective densities."

P9L31-37: The statement made in this paragraph about deriving soot particle compaction from the OPC signal needs to be elaborated further.

- The authors state that "… the most active INP (e.g. R2500U, Monarch 900) showed lower polarization shift signature, which suggest a more optically spherical shape." How can this be read out from the two values log(S1) and log(P1) listed in your Tab. 1? Please specify and/or give appropriate references.
  Added (Garimella et al., 2016; Nichman et al., 2016).

- Besides, the log(S1) and log(P1) values for Monrach 880 and 900 are identical, but at the same time these soot types show differences in ice nucleation. Please comment.
  This difference could be explained by a more branched and less compact shape which is also reflected in the effective density. The optical sphericity alone, is not a sufficient criteria in determination of the IN activity of the particle.

- Next, on P8L5: You state that your OPC starts seeing particles, which are _optically_ lager than 500 nm in diameter. How does the optical particle size compare to the mobility size of 800 nm? Is there any clear relationship between these two sizes? Can it be that some of your 800 nm mobility diameter soot particles are aligned in such a way in the OPC, that they scatter only little light, thus cannot be detected and bias your results?
  The particle's optical diameter is determined from the forward scattering signal using the standard Mie scattering assumptions, i.e., spherical geometry and isotropic refractive index. Therefore, there is no simple connection between the optical diameter and the actual diameter for BC particles with voids. However, the accurate sizing of aerosol doesn't play any role in our experiments where we detect the ice crystals which are more than an order of magnitude bigger in size and have different polarization values.

  The polarization measurement is separate from forward sizing of the aerosol. If there was a preferable orientation for the particles, we would expect the data cluster to be a narrow band or single value. This was not the case here. Due to agglomerate variability and the randomizing action of the rotational Brownian motion affecting nanometric particles, we do not expect preferable orientation of the aerosol. If particle shapes with high aspect ratio did occur (they did not in this study based on our SEM images), they could have a preferable orientation and therefore bias the polarization measurement.

  The paragraph was changed to:

  "Another method for physical characterization is the detection of the shift in polarization in the light scattered from 800 nm BC particles in the OPC. A shift in polarization of the linearly polarized incident beam will occur if the particle is optically anisotropic (having aspherical shape, branches, roughness, or variations in internal structure). This polarization shift is used in classification of particles by their 'optical shape' (e.g. Garimella et al. 2016; Nichman et al., 2016; Kobayashi et al., 2014; ; Glen and Brooks, 2013). Even spheroidal shaped particles produce a unique shape-specific phase function

distinctly different from those produced by other spheroidal particles (Mischenko et al., 1997). Thus parallel and perpendicular polarization measurements can be used to differentiate between particles of the same diameter. Francis et al. (2011) showed that both, agglomerate diameter and spherule diameter of soot, affect the polarization within specific boundaries. In our study, the optical sphericity could shed light on the BC particle shape (i.e. round and compact versus branched and lacy) and its influence on the IN mechanism in BC agglomerates. High optical sphericity of a particle is determined by a low polarization shift in the light scattered from the particle. OPC data cluster centers of single-particle optical measurements for each BC sample are listed in Table 1"

P10L1-7: "The results showed no visible sign of compaction effects on the IN activity…" Is this supported by your SEM analysis, i.e. do wet and dry aerosolized particles look identical? If SEM images are not available, I encourage to include another table, where you display similar to Tab. 1, the log(S1) and log(P1) data of the wet and the dry generated aerosol, which should be available. From these values it should, according to the authors, directly be visible whether a compaction of the particles took place or not.

No, our SEM images were too limited to provide direct evidence for compaction when atomizing from water compared with dry disperse for 800 nm mobility diameter selected particles. The OPC results are inconclusive.

We have changed the text to read: "The results showed no apparent generation-dependent effects on IN activity of 800 nm BC particles (Fig. 5). The dry-aerosolized round compact shapes observed in Fig. A2 may explain the lack of further compaction during the atomization process of 800 nm BC aerosol. Further qualitative support for the hypothesis of initial compactness of the particles was provided by the low values measured in the OPC (Table 1), which are associated with the sphericity of the particles."

P10L7: "… of initial compactness of the particles was provided by the low values measured in the OPC (Tab. 1), which are associated with the sphericity of the particles."

- In the last paragraph (P9L31-37) the authors try to make a case, that the OPC data can be used do estimate particle sphericity. Moreover, the authors claim that more spherical (compact) soot types are better INPs for PCF compared to less spherical soots and that the differences in the OPC values listed in Tab. 1 can be used to support this hypothesis. The statement made here (P10L7) now says that in general the soot types are all very spherical already, supported by the OPC values in Tab. 1. In my eyes, this is directly contradictory to earlier statements in the manuscript (P9L31-37). Please explain.
- The authors are not trying to make a case that an OPC with polarization measurements can be used to determine the optical shape of the particle. It was already shown before (e.g. Garimella et al. 2016; Nichman et al., 2016; Kobayashi et al., 2014).
- Polarization values that are low, suggest a more optically spherical shape. However, within this category we are still able to qualitatively rank the degree of sphericity for each type.
- The authors do not claim at any point that generally more spherical BC are better INP for PCF, this is an inductive fallacy. We simply report experimental observations: "In our

experiments, the most active INP (e.g. R2500U, Monarch 900) showed lower polarization shift signatures, which suggests a more optically spherical shape". Moreover, we only talk about agglomerates of 800 nm in diameter. As was explained in the previous comment, sphericity alone is not a sufficient parameter to determine IN activity and it is brought here as a qualitative measurement to support the main IN observations.

- Following your argumentation from P9L31-37, the soot types presented in Fig. 5 have different sphericities, based on the OPC data. Even if the R2500U soot was already perfectly spherical and compact for the dry generation method, the Regal 400R is not (see your OPC values in Tab. 1) and thus should show a difference in compactness (and ice nucleation) upon wet generation. Please explain.
- Once again, compactness is not a synonym for ice nucleation, compactness alone can't be used as a measure for INP efficiency. As we explained on P3L24-27.
- These two types are not directly comparable for compactness since there are additional parameters that can affect the compactness i.e. the oxidation degree.
- Finally, if there was a compaction in wet generation of R400, the delta in IN efficiency due to the compaction of R400 (which still has a spheroidal shape despite the higher optical polarization values), could be too small to be detected.

P10L14-15: Change to"… the probability of active sites will be higher for larger particles, increasing their probability to nucleate ice."

- One assumption for the active site concept is that active sites are homogeneously distributed over the particle surface and thus ns is constant with particle size. So what changes when going from 100 nm to 800 nm particles is just the probability of having an active site.

  Changed.

P10L19-20: Please explain why the 100 nm aggregates without suitable pores would nucleate ice homogeneously? This is only possible if there is any soluble material on these soots, which can cause homogeneous freezing. Is this the case?

We refer to ice freezing on 100 nm aggregates at or above the homogeneous freezing line, not actual homogeneous freezing.

We have changed the text to read: "The number of aggregates that form the agglomerate define the number and the dimensions of pores that act as IN, which in the extreme case of single aggregates (i.e. without suitable pores) nucleate at or above the homogeneous freezing threshold (e.g. grey circles in Fig. 6)."

P10L21: Change "ice heterogeneously" to "ice below homogeneous freezing conditions".

  Changed.

P10L21-23: Are your ice nucleation data corrected for multiply charged particles? If not, how does the 1% AF chosen as appropriate ice nucleation onset threshold compare to the amount of multiple charged particles and to the maximum AF reached within one RH scan? What is the size distribution of the soot samples? Can you give a particle size distribution from your SEM images, or from the measured vacuum aerodynamic diameter in PALMS? Furthermore, I suggest putting the complete AF curves of your measurements into a supplementary information. These AF curves contain very useful information and could support your arguments about a PCF mechanism, which should result in very steep activated fraction curves, due to the homogeneous freezing of the water in the pores.

The 800 nm mobility diameter particles are essentially on the larger-diameter edge of the size distribution. Soots main mode is on the order of several hundreds of nanometers. The doubly charged particles, of the 800 nm singly charged particles, would need to be above 1 micron in diameter and literally in negligible concentration.

Moreover, in the OPC we saw particles only in the first 2 bins but not in any other higher bins.

Some examples of ethylene soot size distributions measured for several fuel settings of the miniburner are attached below.

Secondly the Vac-Aerodynamic diameter was measured for mobility diameter selected particles and was constant for the vast majority of particles – this is how the effective density values were derived and presented in the table 800nm/Vac-AeroD. In regard to PALMS measurements of 100 nm particles, please see comment P7L1.

In order to obtain a quantitative PSD from SEM and get a higher confidence in the statistics, a significantly higher number of scans will be required. SEM technique is generally quantitative to determine particle shape and structure and without advanced tools for automated scanning it is considered to be impractical for PSD analysis over the whole area of the substrate.

An example of AF plot was added to SI (Fig. S1).

In regard to100 nm particles, we would expect the 100 nm BC INP to freeze at the homogeneous line (see also Mahrt et al. 2018). One possible explanation that they do not reach homogeneous nucleation, i.e. nucleate at higher SS% but still below Koops line, is the presence of doubly charged particles which are bigger agglomerates and nucleate better than the single aggregates. However, they nucleate not as well as if these were solely big particles. As stated in the text of Sect 3.1, point (3), last few sentences: "Despite the reduction in activity, the 100 nm soot has nucleated ice below homogeneous freezing conditions. It is possible that a bias introduced by doubly charged particles, passed at the same DMA voltage, maintained the high IN activity of 100 nm mobility diameter soot".

[Figure]

Figure. Mobility diameter distributions of ethylene combustion soot particles for different fuel settings.

P10L8-24: Did you measure the other soot types also for both 100 and 800 nm? Why is this data not presented here? The selection of data points seems a bit random: Why do you only show one data point for 800 nm particles of Monarch 800 around 227 K? Is this statistically significant to compare one RH scan here to two RH scans performed on 100 nm particles at similar temperatures?

Due to limited resources, we measured the IN activity for three different types of BC particles (Monarch 880 and 900, and ethylene soot). In all three cases, the smaller particles appeared to freeze at a higher critical supersaturation for a given temperature (Fig. 6), including both Monarch samples that froze near the homogeneous freezing limit. Similar homogeneous freezing for small BC particles were reported by Mahrt et al. 2018. For each RH scan, we sampled thousands of particles passing through the SPIN system; thus, we feel that these measurements are valuable and worth presenting.

P10L30: Change to "…while the non-oxidized sample showed ice nucleation activity, well below homogeneous freezing conditions."

Changed.

P11L1: This statement is incorrect and should be revised. The PCF mechanism is in fact very sensitive to the pore wall hydrophilicity, usually considered in terms of contact angle, see e.g. Eq. (1) in Marcolli (2016). You can use the indicated equation to calculate the RHw for pore

filling at a fixed pore diameter and check its sensitivity to the contact angle. In other words, a porous particle will not be able to nucleate ice via PCF, i.e. at conditions well below water saturation, if the contact angle is too high, even though if pores of suitable diameters are present. For instance, a pore of 6 nm diameter and a contact angle of 0° requires a RHw of 57% to be filled at -50°C, whereas the same pore would require a RHw of 91% in case the contact angle was 80 °.

We apologize for incorrectly suggesting that the Morishige (2018) study investigated "an oxidized BC sample". We have modified the text to read: "The freezing behavior of water confined in pores of hydrophilic silica or hydrophobic carbon was similar, suggesting that pore hydrophobicity may play a limited role in PCF-type freezing (Morishige, 2018)."

We cite Morishige (2018) as an example of where experimental results suggest that the hydrophobicity of pore surfaces may not be critical for the PCF mechanism. In fact, Morishige (2018) provides an explanation from their experimental observations whereby a thin layer of water separates the confined water from direct contact with the pore walls.

Lupi & Molinero 2014: "Hydrophobic and hydrophilic atomically rough surfaces do not induce layering and do not promote heterogeneous nucleation of ice" – i.e. hydrophilicity or hydrophobicity do not play a direct role. "Layering of liquid water at the surface correlates with the ability of carbon particles to promote ice nucleation, Ice-like order in interfacial liquid water appears as fluctuating patches of bilayer hexagons on the carbon surfaces. These domains of bilayer hexagons were not observed on the atomically rough surfaces". In this context, contact angle measurements, which are as the Reviewer correctly suggested, usually represent the hydrophilicity/phobicity of the surface, may be completely irrelevant for this IN application due to the inherent bias of this measurement.

We have no actions concerning the Reviewer's theoretical point on PCF contact angle dependence.

P11L6-8: I disagree with this statement and I believe this conclusion cannot be drawn from your data. The Regal 400R as presented in Figs. 1 and 7 nucleates ice at homogeneous freezing conditions and does not nucleate ice via PCF, as you correctly argue throughout the manuscript. The absence of PCF freezing for Regal 400R can have various reasons, such as absence of suitable pores, too high contact angle or soluble material on the soot filling the pores and thus preventing pore condensation due to the inverse Kelvin effect etc. Your argument that surface oxidation leads to inhibition of pore filling and freezing (PCF) due to surface polarity would only be true if you were to take a soot (e.g. Monarch 900), which clearly shows PCF (Fig. 4) but has a non-oxidized surface (Tab. 1), and that this PCF freezing disappears upon oxidation of this soot. Finally, from your Fig. A1 it looks like the ethylene soot also shows a tendency to have relatively high O-peaks. This, along with the slightly acidic ph listed in Tab. 1 suggests an oxidized surface to me (see also your description P6L1-5). Nevertheless, the ethylene soot clearly reveals a PCF freezing mechanism, which contradicts your statement.

We agree with the Reviewer that the best experimental technique to test surface oxidation effect on BC particles would be take one type of BC particle, test the IN activity, oxidize the surface, and retest the IN activity. Without this specific capability, we chose the next best approach. We conducted IN activity measurements of Cabot Regal 400R and Regal 330R carbon black particle

types, where these two types were specifically chosen as they have very similar physical properties, as supplied by the same manufacturer and shown in Table 1, but different chemical properties on their surfaces. Our objective with these experiments was to explicitly investigate how the different surface chemistries for these two carbon black particles types might affect IN activity. As we observed and show clearly in Figure 7, at a temperature of ~228 K the oxidized surface Regal 400R induced ice formation near Koop's homogeneous limit, whereas the non-oxidized surface Regal 330R induced ice formation below the homogeneous line (i.e. ~1.35 supersaturation with respect to ice). Further, the oxidized surface Regal 400R sample induced ice nucleation near the homogeneous line down to temperatures of ~220 K. In fact, the Regal 400R was the only BC particle type we studied that did not exhibit a more efficient freezing behavior than homogeneous freezing. Our observations of ice formation near the homogeneous ice limit at all temperatures for the surface oxidized Regal 400R sample suggests that this specific surface oxidation suppresses or inhibits pore-induced-freezing mechanisms. Similar observations of inhibited ice nucleation on oxidized surfaces reported in the literature are discussed in previous replies and in the manuscript.

The Reviewer is correct that we do not know why or how the surface oxidation of Regal 400R affects ice formation. We do, however, know that this effect was not expected based on our own interpretation of surface hydrophilicity effects, as well as based on the Reviewer's expectations from PCF theory as embodied by increasing surface wetting on the assumed contact angle. Thus, currently, we have no direct explanation.

We hypothesized, based on early work by Fletcher (1959), that high polarity at the surface of pores in the Regal 400R may inhibit ice nucleation. The Reviewer adds that Regal 400R may have an "absence of suitable pores, too high contact angle or soluble material on the soot filling the pores. On the other hand, oxidation of the carbonaceous surfaces could also involve a loss of surface planarity (Cabrera-Sunfelix & Darling 2007), introducing the known effect of surface curvature on IN (Lupi, 2014). Unfortunately, we cannot directly discount any of these other proposed hypotheses. That said, we did, to the limit of our resources, attempt to address them by choosing carbon black particle types from the same manufacturer, with very similar physical properties, but explicitly different surface chemical properties. Regal 400R suspends in water better than Regal 330R, consistent with a higher surface oxidation and a lower contact angle for water on the surface. We tested Regal 400R IN activity using two different aerosol generation techniques (Figure 5), both of which gave the same result, consistent with no significant soluble material on the surface.

In light of this discussion, we have changed the text to read: "Häusler et al. (2018) suggested that agglomeration may lead to a favorable positioning of the functional sites and therefore to an increase in the IN activity, even though a decrease in the surface area occurs. However, it was found that the increased proportion of oxygen increases the hydrophilicity of graphene, reduces agglomeration and hence increases the surface area and reduces the number of pores (Häusler et al., 2018). Thus, it is possible that by oxidizing the surface of Regal 400R particles, the micro structure of the particles changed (Cabrera-Sunfelix & Darling 2007), reducing the number of PCF active pores. Fletcher (1959) noted that a highly polar surfaces could raise the free energy of formation of ice embryos, reducing the efficiency of heterogeneous ice formation, providing another potential explanation. Finally, if the surface is highly oxidized, the Regal 400R particles may condense monolayers of water more readily than unoxidized Regal 330R, affecting the ice

formation. All of these potential explanations are also consistent with the surprisingly high effective density of Regal 400R sample."

Finally, the Reviewer points to Figure A1 and measured pH and suggests that the ethylene soot has an apparent intermediary level of measured oxygen, between the surface oxidized Regal 400R and the other carbon black materials. We agree with the potential for the ethylene soot to have a slightly oxidized surface. It is understandable that combustion generated soot may have some surface oxidation, though it is apparent that it is not close to the level of Regal 400R. Our IN-activity observations suggest that the ethylene soot particles exhibited efficient ice formation, especially relative to Regal 400R and/or homogeneous ice nucleation. If ethylene soot represents the mid-point between non-oxidized and heavily oxidized surfaces on BC particles, then small amounts of surface oxidation does not appear to significantly affect IN activity.

We have rewritten Section (4) and revised Figure 7 to include ethylene soot in the discussion.

P11L14: This argument about ice lattice seems out of place here and should be removed.

Removed. The text has been changed to read: "The combined contribution of single spherule size, particle size, surface oxidation, and morphology to IN activity affects the spatial arrangement and thus the adjacent angles in the pores that dictate the formation, and perhaps the type, of ice (Bi et al., 2017; Zhu et al., 2018)."

P11L16-20: This is a repetition from P5L6-7 and P3L4-6 and should be removed here.

Reduced to:
"Surface oxidation is not the only process altering the IN activity. Crawford et al. (2011) showed that alteration of the organic carbon content from minimum (5 %) to medium (30 %) results in a clear transition between heterogeneous and homogeneous freezing mechanism, respectively. On the other hand, some organic acids enhance IN activity (Zobrist et al., 2006; Wang and Knopf, 2011)."

P11L23: "The cis-pionic and stearic acid when atomized, nucleate ice homogeneously." I suggest adding these data points to your Fig. 8.

We added a figure with the remaining coating experiments, including the pure compounds to the Supplement.

P11L25: Do you mean "ice supersaturation" of 10%?

Changed to ice supersaturation.

P11L24: If your coating of cis-pionic and stearic acid fills all the pores of your BC agglomerates, the ice nucleation onset should be shifted towards homogeneous freezing conditions, as the pores are blocked and can thus not trigger ice nucleation via PCF. The results in your Fig. 8c seem not very robust, with one blue data point lying at the homogeneous freezing curve and the other one lies within uncertainty of the uncoated (grey) data point.

Overall there is an observable shift upwards to higher SS (not only with cis-pinonic acid). We did not have a good way to filter the homogeneously formed organic particles. We could only control the generation of organic vapors (as explained in P6L21-22) but without knowing the extent of coating or pore filling. The blue data point underlines the sensitivity of this experiment.

Since we had no homogeneously nucleated particles for the coating part of the experiment, the blue data point is essentially R2500U fully coated with stearic acid (one data point is an experiment which involves thousands of particles).

P12L1-2: In Fig. 9 you compare your ice onset nucleation in terms of an activated fraction threshold to a ns parametrization by Ullrich et al. (2017). This does not make any sense. Please revise this figure and convert your AF based onset data points to ns. I feel you can omit this figure and plot the Ullrich et al. parametrization directly in your Fig. 9 for better comparison. Also, your Fig. 9 should include all your soot data.

Figure 9 was changed to present active site density isolines of the most active samples in the study in comparison to the parametrized isolines from Ullrich et al. (2017). Figure 10 was changed to present RHw vs Temperature at low temperature regime as requested by Reviewer 1.

P12L3-6: This is a repetition from the methods described on P7L6-16 and should be deleted here.

Deleted.

P12L9-11: How do the authors arrive at this conclusion? If I look at your non-oxidized data point around -54 °C in Fig. 9, the error bars cross the isolines of both 1011 and 1010 m-2. But again, the comparison of AF based thresholds and ns is misleading, see my comment on P12L1-2.

Figure 9 was changed to present isolines.

P12L13-16: This is a repetition of P2L28-30 and should be removed here.

Changed to: "The widespread in IN activity of BC obscures understanding of the radiative properties of clouds and Earth's climate".

P12L18: Replace "hindering" by "limiting"

Changed.

P12L24: "For comparable…"; see my comment P12L1-2.

Changed to: "For comparable oxidized and non-oxidized particles, lower activity was observed in the oxidized particles".

P12L27: The sentence "Hence, the …" should be moved in front of the last sentence to finish the discussion about oxidation, before starting to talk about coating effects.

Moved.

P12L32: I suggest to tune this down and say "… can partly be explained"

Changed.

P12L21: Change to "… dominates, coverage of high clouds…"

Changed.

P12L23: contribute*s*

Changed.

P12L29: Rephrase to: "… our results of BC ice formation below homogeneous freezing conditions…"

Rephrased.

P13L2: Delete "which is sensitive to coating"

Deleted.

P13L2-4: How do the authors derive at this conclusion? Why would such a droplet splintering inside the pore only lead to an underestimation of BC in ice residuals? Such an effect should be similar for other porous materials, such as volcanic ash particles and also porous mineral dust, to name a few… (Marcolli 2016; Wagner et al. 2014).

Our hypothesis is the following. If PCF does occur in atmospheric particles, ice formation in small pores may lead to ice multiplication mechanisms through the fracturing of the original heterogeneous substrate particle. If BC particles only activate via PCF mechanisms, then any subsequent splintering will create an effective underestimation of BC particles in ice residuals. The PCF-splintering mechanism may also occur in materials other than BC particles; however, other materials such as minerals may be more efficient at activating IN via other mechanisms than PCF. Thus, PCF-splintering may not have as significant an effect on the number of measured minerals in ice residuals, as it might for BC particles.

P13L8: Delete: "the in situ type"

This definition was suggested by Krämer et al., 2016 for these conditions of the ice nucleation regime.

P13L9: Delete "simulated"

Deleted.

P13L10-11: the "strong dependence on temperature" is inconsistent with a PCF mechanism, but likely an instrument limitation, see my comment P8L25-27.

Please see our responses to comment P8L25-27.

P13L16: The discussion of the compaction of 200 nm BC particles should be followed by a reference.

We changed the text to read: "Aerosol generation techniques (i.e. dry versus wet-dried), and compaction did not seem to have a significant effect on the IN activity of atomized 800 nm agglomerates."

P13L23: Change to: "… can take place well below homogeneous freezing conditions."

Changed.

P13L26-27: This has systematically been done in a recent study by Mahrt et al. (2018).

We thank the Reviewer for the reference. We have changed our text to read: "In future studies, IN activity enhancement for various BC particle types with size should be tested in more detail, in the range 100 – 800 nm, similar to our experiments here and experiments conducted by Mahrt et al. (2018) on from 100-400 nm BC particle types."

P13L33: Who is "MF"? Do you mean "MW"?

Changed.

Fig. 3:
- I suggest to have the error bars in the same color as the dots to increase readability.

  changed
- You caption should be changed to: "Ice nucleation onset conditions defined as 1% of the total aerosol to nucleate ice for 800 nm BC particles." If you claim PCF to be responsible, the freezing process is not heteregeneous, but homogeneous, see my comment P7L36. Also, there is no need to repeat the temperature range in th ecaption, as it is indicated by the x-axis.

  Caption changed to:

"Ice nucleation onset conditions defined as 1% of the total aerosol to nucleate ice for 800 nm BC particlesIce onset by heterogeneous nucleation from 800 nm BC particles at supersaturated conditions w.r.t. ice in the temperature range 217 to 235 K by 1 % of the particles. Solid black line is the homogeneous freezing threshold (Koop et al., 2000)."

- I assume each dot represents an individual RH scan in SPIN, correct? What do the errorbars represent?

  Due to slight heterogeneities in wall temperatures, the temperature and supersaturation in the SPIN aerosol lamina vary. The effect of this variability on lamina conditions is quantified using results from CFD simulations performed by Kulkarni and Kok (2012). The uncertainty in the average lamina temperature and supersaturation is calculated using 16 thermocouples positioned on each wall. At higher supersaturations and colder lamina temperatures, the wall temperatures exhibit greater variability as the PID temperature controllers rely less on the thermoelectric heaters to maintain cooler wall temperatures. This increases the heterogeneity in wall temperature, leading to higher uncertainties.

  We added the following in the text: ".... error bars, which represent the variability of the laminar conditions based on CFD simulations by Kulkarni and Kok (2012), are presented….".

Fig. 4:

- The solid colored lines should be removed or the parametrizations for the fits should be given and justified.

  Colored lines removed

Fig. 5:

- Include grid lines and boxes around the graph to be consistent with your Figs. 3, 4.

  Grid lines and boxes were included

Fig. 8:

- Add grid lines and boxes.

  Grid lines and boxes were included

- Unit on x axis should read °C not 0C

  Changed

- Why do only a few data points have error bars and why are these of different size for the data points?

  Error bars removed to avoid confusion. Fig S3 was added with more data points and representative error bars to avoid hindering the actual data. The error bars size was explained in comment to Fig. 3.

Fig. 9:

- Why do you constrain yourself to 3 of your soot types measured?

  The Reviewer probably refers to Fig.10. These are the most active INP for which we estimated the effective surface area. Fig.10 was changed as requested by Reviewer1.

- Also, it is not clear how you decide to just plot 1 data point for the three soot types shown? How do you derive this from your data presented in Fig. 3? You should include all your data in this summary plot.

  Fig.10 was changed to include all data.

- Constrain legend to: "Kanji et al. (2017)"

  Legend changed

Fig. A1:

- Delte "800 nm" in the legend and write into figure caption that this data corresponds to 800 nm particles.

  Changed

- I suggest to plot the mean/center for each cluster/soot type into the plot at the same color, but larger symbol size to guide the eye.

  Centroids were added

Tab.1:

- Please add uncertainties to measurements of ph, O:C ratio and OPC values.

  OPC polarization measurement presented as a qualitative supporting information for intercomparison between the samples, the uncertainty remains similar for all the particles.

  An accurate determination of uncertainty requires additional details, such as information regarding the complex refractive index and the theoretical calculation of non-uniform particles' scattering based upon a particle structure model, which we do not currently have access to. If we take the light scattering measurement uncertainty, which can result in an error of 1 bin in sizing, we can assume the error in scattered light polarization to be similar on the order of 20 %. However this number may vary as explained above.

  Uncertainties in PALMS data originate from ablation and ionization efficiencies of individual particles. Peak identification uncertainty is low due to the low amount of impurities in the commercial samples. The distribution of the O and C values within the samples can be seen in Fig. A1 and can be used to calculate the uncertainty of the median value provided in the table. We estimate the deviation from the mean to be 50 % of the value, on average. Since we present the median value in the table, we added this information in the caption of the table.

PH uncertainty was reproducible up to the precision of the instrument. We added this value in the table caption.

- e: subscribt of "m" after D.

Changed.

**References used in this response:**

Alstadt, Valerie J., et al. (2017), 'Heterogeneous Freezing of Carbon Nanotubes: A Model System for Pore Condensation and Freezing in the Atmosphere', *The Journal of Physical Chemistry A,* 121 (42), 8166-75.

Christenson, H. K. (2013), 'Two-step crystal nucleation via capillary condensation', *Crystengcomm,* 15 (11), 2030-39.

Grawe, S., et al. (2018), 'Coal fly ash: Linking immersion freezing behavior and physico-chemical particle properties', *Atmos. Chem. Phys. Discuss.,* 2018, 1-32.

Higuchi, K. and Fukuta, N. (1966), 'Ice in capillaries of solid particles and its effect on their nucleating ability', *Journal of the Atmospheric Sciences,* 23 (2), 187-90.

Hiranuma, N., et al. (2014), 'A comprehensive laboratory study on the immersion freezing behavior of illite NX particles: a comparison of seventeen ice nucleation measurement techniques', *Atmospheric Chemistry and Physics Discussions,* 14 (15), 22045-116.

Mahrt, Fabian, et al. (2018), 'Ice nucleation abilities of soot particles determined with the Horizontal Ice Nucleation Chamber', *Atmospheric Chemistry and Physics,* 2018, 1-41.

Marcolli, C. (2016), 'Pre-activation of aeosol particles by pore condensation and freezing', *Atmos. Chem. Phys.,* 2016, 1-48.

Möhler, O., et al. (2005), 'Effect of sulfuric acid coating on heterogeneous ice nucleation by soot aerosol particles', *Journal of Geophysical Research: Atmospheres,* 110 (D11), D11210.

Murray, B. J. (2008), 'Inhibition of ice crystallisation in highly viscous aqueous organic acid droplets', *Atmospheric Chemistry and Physics,* 8 (17), 5423-33.

Niemand, Monika, et al. (2012), 'A Particle-Surface-Area-Based Parameterization of Immersion Freezing on Desert Dust Particles', *Journal of the Atmospheric Sciences,* 69 (10), 3077-92.

Umo, N. S., et al. (2015), 'Ice nucleation by combustion ash particles at conditions relevant to mixed-phase clouds', *Atmos. Chem. Phys.,* 15 (9), 5195-210.

Vali, G., et al. (2015), 'Technical Note: A proposal for ice nucleation terminology', *Atmos. Chem. Phys.,* 15 (18), 10263-70.

Wagner, Robert, et al. (2014), 'Enhanced high-temperature ice nucleation ability of crystallized aerosol particles after preactivation at low temperature', *Journal of Geophysical Research: Atmospheres,* 119 (13), 8212-30.

Welti, A., et al. (2012), 'Time dependence of immersion freezing: an experimental study on size selected kaolinite particles', *Atmospheric Chemistry and Physics,* 12 (20), 9893-907.

Zobrist, B., et al. (2008), 'Do atmospheric aerosols form glasses?', *Atmospheric Chemistry and Physics,* 8 (17), 5221-44.

**References used in this response by the authors:**

[revised manuscript text omitted]

Mahrt, Fabian, et al. (2018), 'Ice nucleation abilities of soot particles determined with the Horizontal Ice Nucleation Chamber', Atmospheric Chemistry and Physics, 2018, 1-41.

Marcolli, C.: Deposition nucleation viewed as homogeneous or immersion freezing in pores and cavities, Atmos. Chem. Phys., 14(4), 2071–2104, doi:10.5194/acp-14-2071-2014, 2014.

Marmur, A.: Soft contact: measurement and interpretation of contact angles, Soft Matter, 2(1), 12–17, doi:10.1039/B514811C, 2006.

Marmur, Abraham, Claudio Della Volpe, Stefano Siboni, Alidad Amirfazli, and Jaroslaw W. Drelich. "Contact angles and wettability: towards common and accurate terminology." Surface Innovations 5, no. 1 (2017): 3-8.

Marsh, H. and Rodríguez-Reinoso, F., Activated Carbon, Elsevier Science Ltd, Oxford., https://doi.org/10.1016/B978-0-08-044463-5.X5013-4, 2006.

Morishige, K.: Influence of Pore Wall Hydrophobicity on Freezing and Melting of Confined Water, J. Phys. Chem.C, acs.jpcc.8b00538, doi:10.1021/acs.jpcc.8b00538, 2018.

Nichman, L., Fuchs, C., Järvinen, E., Ignatius, K., Höppel, N. F., Dias, A., Heinritzi, M., Simon, M., Tröstl, J., Wagner, A. C., Wagner, R., Williamson, C., Yan, C., Connolly, P. J., Dorsey, J. R., Duplissy, J., Ehrhart, S., Frege, C., Gordon, H., Hoyle, C. R., Kristensen, T. B., Steiner, G., McPherson Donahue, N., Flagan, R., Gallagher, M. W., Kirkby, J., Möhler, O., Saathoff, H., Schnaiter, M., Stratmann, F., and Tomé, A.: Phase transition observations and discrimination of small cloud particles by light polarization in expansion chamber experiments, Atmos. Chem. Phys., 16, 3651-3664, https://doi.org/10.5194/acp-16-3651-2016, 2016.

Persiantseva, N. M.  Popovicheva O. B. and  Shonija N. K.:Wetting and hydration of insoluble soot particles in the upper troposphere, J. Environ. Monit., 6, 939–945, DOI: 10.1039/B407770A, 2004.

Petzold, A., Ogren, J. A., Fiebig, M., Laj, P., Li, S.-M., Baltensperger, U., … Zhang, X.-Y. (2013). Recommendations for reporting "black carbon" measurements. Atmospheric Chemistry and Physics, 13(16), 8365–8379. https://doi.org/10.5194/acp-13-8365-2013.

Phillips, V. T. J., Demott, P. J., Andronache, C., Pratt, K. A., Prather, K. A., Subramanian, R. and Twohy, C.: Improvements to an Empirical Parameterization of Heterogeneous Ice Nucleation and its Comparison with Observations. J. Atmos. Sci., 70:378–409, 2013.

Salame I.I. and Bandosz, T.J.: Experimental Study of Water Adsorption on Activated Carbons, Langmuir,15,(2), 587-593, DOI: 10.1021/la980492h, 1999.

Sedlacek, A. J., Lewis, E. R., Onasch, T. B., Lambe, A. T., and Davidovits, P.: Investigation of Refractory Black CarbonContaining Particle Morphologies Using the Single-Particle Soot

Photometer (SP2), Aerosol Sci. Tech., 49, 872–885, https://doi.org/10.1080/02786826.2015.1074978, 2015.

Strobel, M. and Lyons, C. S. (2011), An Essay on Contact Angle Measurements. Plasma Processes Polym., 8: 8-13. doi:10.1002/ppap.201000041

Ullrich, R., Hoose, C., Möhler, O., Niemand, M., Wagner, R., Höhler, K., Hiranuma, N., Saathoff, H. and Leisner, T.: A New Ice Nucleation Active Site Parameterization for Desert Dust and Soot, J. Atmos. Sci., 74(3), 699–717, doi:10.1175/JAS-D-16-0074.1, 2017.

Vali, G., DeMott, P. J., Möhler, O. and Whale, T. F.: Technical Note: A proposal for ice nucleation terminology, Atmos. Chem. Phys., 15(18), 10263–10270, doi:10.5194/acp-15-10263-2015, 2015.

Welti, A., Lüönd, F., Kanji, Z. A., Stetzer, O., and Lohmann, U.: Time dependence of immersion freezing: an experimental study on size selected kaolinite particles, Atmos. Chem. Phys., 12, 9893-9907, https://doi.org/10.5194/acp-12-9893-2012, 2012.

Zawadowicz, M. A., Froyd, K. D., Murphy, D. M., & Cziczo, D. J. (2017). Improved identification of primary biological aerosol particles using single-particle mass spectrometry. Atmospheric Chemistry and Physics, 17(11), 7193-7212. doi:http://dx.doi.org/10.5194/acp-17-7193-2017.

Zhu, W., Zhu, Y., Wang, L., Zhu, Q., Zhao, W., Zhu, C., Bai, J., Yang, J., Yuan, L.-F., Wu, H.-A. and Zeng, X. C.: Water Confined in Nanocapillaries: Two-Dimensional Bilayer Square-like Ice and Associated Solid-Liquid-Solid Transition, J. Phys. Chem. C, acs.jpcc.8b00195, doi:10.1021/acs.jpcc.8b00195, 2018.

---

## Referee Report (RR1)

**Review to "Laboratory study of heterogeneous ice nucleation on black carbon containing aerosol" by Nichman et al. ACPD, 2018**

The manuscript by Nichman et al. presents a laboratory investigation of the ice nucleation ability of black carbon (BC) particles of different type and sizes in the cirrus regime. The topic of BC ice nucleation is of high relevance for atmospheric science and climate, and as such for ACP.

**General comment:**

The manuscript is well structured and written. I would like to congratulate the authors, who have significantly improved the quality of the manuscript, compared to the previous discussion paper. The five aspects influencing BC ice formation identified by the authors (see P5) are discussed in fair detail, even though I found the introduction quite long and partly hard to read. However, the discussion of the measurement uncertainties in terms of RH and T is poor. This makes it hard to follow parts of the comparisons between the different soot types and the argumentation for some of the figures.
Overall, I recommend the manuscript for publications, after some minor and specific comments, which I list below, have been addressed:

**Specific comments:**

- P1L23: Change to "…that govern the ice nucleation activity of BC."
- P1L31: Delete "lamina".
- P2L16: Do you mean cloud types?
- P2L18: "While field…". This statement should be followed by references. I suggest deleting this sentence here, as you have a detailed discussion of both aspects further down.
- P2L34: Kanji et al. (2017) and Hoose and Möhler (2012) are review-type articles, sourcing data from other studies. Please delete these here and cite primary studies instead. Also, add: Häusler et al. (2018)
- P3L7: "…(PCF) mechanism." Should be followed by a reference.
- P3L11: delete "heterogeneous", this is clear since you say below homogeneous freezing.
- P3L36: Change to "can be expressed"
- P4L1: None of the given references makes sense to me. Are you just trying to cite studies that have used ns? Consider deletion.
- P4L11: Check formatting: "2014" of Marcolli is not in parenthesis, while other studies are. Add David et al. (2019).
- P4L13: Delete "diameter"
- P4L20: Add reference after "diameter range".
- P4L24: Add Umo et al. (2019)
- P5L3: I suggest tuning this down and saying: "… have not fully been established."
- P5L29: Check formatting/indent.
- P6L14: Define "BET"
- P6L17: "Submicron"
- P6L17: Delete "burner"
- P6L22: Replace "test" by "method"
- P7L17: "organic molecules." Should be followed by a reference.
- P8L35: This is not completely true, ice supersaturation can/is also be achieved when water-saturation conditions are created in SPIN. Please rephrase and formulate more carefully.
- P9L16: Are your ice nucleation data corrected for multiple charged particles? Please specify the sizes of double charged particles for the mobility diameters used by you. Could it be that the OPC detects a large, double-charged BC particle as ice?
- P9L24: Can you set this 1% threshold into context of the maximal activate fraction observed during your RH-scans. Please comment on this in the manuscript.
- P9L34: I could not find the definition for "$SS_i$".
- P10L4: Add Mahrt et al. (2018)
- P10L7: Why do you write "Heterogeneous ice nucleation". On P4L10 you state that water in pores freezes "homogenously". Please clarify in manuscript.

- P10L13: "The displayed…" This statement is not supported by your data shown in Fig. 4, in particular by the data points around T = 228 K. Please clarify. This would become even clearer if you were to include error bars in your data points, similar to your Fig. 3.
- P10L29: Again, it is very hard to justify and/or follow this statement without error bars in Fig. 4.
- P11L33: The reported size dependence seems consistent with that observed in studies by Friedman et al. (2011), Mahrt et al. (2018) and Crawford et al. (2011) that you cite above. You might want to refer back to these studies.
- P11L36: Replace "," by "."
- P12L2: "Hence, the…": I assume that every data point in your Fig. 6 represents a single run. Please comment on how reproducible these are within the manuscript. Also, if I consider the error bars reported in your Fig. 3 I am not sure how different your 100 and 800 nm ethylene soot in Fig. 6 are. Please include error bars in Figs. and in discussion of data.
- P13L7: Delete space between "sample" and "."
- P13L10: Include space before "However"
- P13L27: Delete space before "The"
- P14L10: Please define "channel process"
- P14L17: Please change to "INP"
- P14L32: Consider to replace "down" with "along"
- P16L3: Who is "MF"?
- Figs. 3, 4, 6, 7, 8: The y-axis should be labeled as "Supersaturation **ratio** with respect to ice", or simply "$S_{ice}$", to be consistent with your terminology. "Supersaturation over ice" as used is wrong. I recommend to be consistent through out all figures within the manuscript.

Crawford, I., et al. (2011), 'Studies of propane flame soot acting as heterogeneous ice nuclei in conjunction with single particle soot photometer measurements', *Atmospheric Chemistry and Physics,* 11 (18), 9549-61.

David, R. O., et al. (2019), 'Pore condensation and freezing is responsible for ice formation below water saturation for porous particles', *Proceedings of the National Academy of Sciences of the United States of America,* 116 (17), 8184-89.

Friedman, B., et al. (2011), 'Ice nucleation and droplet formation by bare and coated soot particles', *Journal of Geophysical Research-Atmospheres,* 116.

Häusler, T., et al. (2018), 'Ice Nucleation Activity of Graphene and Graphene Oxides', *Journal of Physical Chemistry C,* 122 (15), 8182-90.

Mahrt, F., et al. (2018), 'Ice nucleation abilities of soot particles determined with the Horizontal Ice Nucleation Chamber', *Atmospheric Chemistry and Physics,* 18 (18), 13363-92.

Umo, N. S., et al. (2019), 'Enhanced ice nucleation activity of coal fly ash aerosol particles initiated by ice-filled pores', *Atmos. Chem. Phys.,* 19 (13), 8783-800.

---

## Author Response (AR2)

Response to 2nd Reviewer minor revisions for Nichman et al. ACPD, 2018.

We thank the editor and reviewers to their efforts. Reviewer's comments are in black. Our responses are in blue.

Review to "Laboratory study of heterogeneous ice nucleation on black carbon containing aerosol" by Nichman et al. ACPD, 2018

The manuscript by Nichman et al. presents a laboratory investigation of the ice nucleation ability of black carbon (BC) particles of different type and sizes in the cirrus regime. The topic of BC ice nucleation is of high relevance for atmospheric science and climate, and as such for ACP.

General comment:
The manuscript is well structured and written. I would like to congratulate the authors, who have significantly improved the quality of the manuscript, compared to the previous discussion paper. The five aspects influencing BC ice formation identified by the authors (see P5) are discussed in fair detail, even though I found the introduction quite long and partly hard to read. However, the discussion of the measurement uncertainties in terms of RH and T is poor. This makes it hard to follow parts of the comparisons between the different soot types and the argumentation for some of the figures.

We thank the Reviewers' for their efforts reviewing our manuscript.
We added error bars in Fig.6. The uncertainties only briefly discussed here but more detailed description of RH,T uncertainties of this exact SPIN instrument can be found in the cited papers (Garimella et al., 2016; Garimella et al., 2017; Wolf et al., 2019).

**Garimella, S., Bjerring Kristensen, T., Ignatius, K., Welti, A., Voigtländer, J., Kulkarni, G. R., Sagan, F., Lee Kok, G., Dorsey, J., Nichman, L., Alexander Rothenberg, D., Roesch, M., Kirchgäßner, A. C. R., Ladkin, R., Wex, H., Wilson, T. W., Antonio Ladino, L., Abbatt, J. P. D., Stetzer, O., Lohmann, U., Stratmann, F. and James Cziczo, D.: The SPectrometer for Ice Nuclei (SPIN): An instrument to investigate ice nucleation, Atmos. Meas. Tech., 9(7), 2781–2795, doi:10.5194/amt-9-2781-2016, 2016.
**Garimella, S., Rothenberg, D. A., Wolf, M. J., David, R. O., Kanji, Z. A., Wang, C., Rösch, M., and Cziczo, D. J.: Uncertainty in counting ice nucleating particles with continuous flow diffusion chambers, Atmos. Chem. Phys., 17, 10855-10864, https://doi.org/10.5194/acp-17-10855-2017, 2017.
**Wolf, M.J., Coe, A., Dove, L.A., Zawadowicz, M.A., Dooley, K., Biller, S.J., Zhang, Y., Chisholm, S.W., Cziczo, D.J.: Investigating the Heterogeneous Ice Nucleation of Sea Spray Aerosols Using Prochlorococcus as a Model Source of Marine Organic Matter, Env. Sci. Tech. 53 (3), 1139-1149, 2019.

Overall, I recommend the manuscript for publications, after some minor and specific comments, which I list below, have been addressed:

Specific comments:

- P1L23: Change to "…that govern the ice nucleation activity of BC."

Changed.

- P1L31: Delete "lamina".

Changed.

- P2L16: Do you mean cloud types?

Changed to "... homogeneous and heterogeneous freezing mechanisms."

- P2L18: "While field…". This statement should be followed by references. I suggest deleting this sentence here, as you have a detailed discussion of both aspects further down.

Sentence has been deleted, as suggested.

- P2L34: Kanji et al. (2017) and Hoose and Möhler (2012) are review-type articles, sourcing data from other studies. Please delete these here and cite primary studies instead. Also, add: Häusler et al. (2018)

Several primary studies are already listed in the brackets, we removed the review-type articles and added Hausler et al. 2018.

- P3L7: "…(PCF) mechanism." Should be followed by a reference.

Added (Macolli, 2014)

- P3L11: delete "heterogeneous", this is clear since you say below homogeneous freezing.

Deleted

- P3L36: Change to "can be expressed"

Changed.

- P4L1: None of the given references makes sense to me. Are you just trying to cite studies that have used ns? Consider deletion.

References deleted as suggested.

- P4L11: Check formatting: "2014" of Marcolli is not in parenthesis, while other studies are. Add David et al. (2019).

Done

- P4L13: Delete "diameter"

Done.

- P4L20: Add reference after "diameter range".

Added (Vali, 2014)

- P4L24: Add Umo et al. (2019)

Done

- P5L3: I suggest tuning this down and saying: "… have not fully been established."

Changed.

- P5L29: Check formatting/indent.

Done.

- P6L14: Define "BET"

Removed "(high BET) value" here, as we use and define BET below.

- P6L17: "Submicron"

Done.

- P6L17: Delete "burner"

Changed "burner" to "flame".

- P6L22: Replace "test" by "method"

Done.

- P7L17: "organic molecules." Should be followed by a reference.

Added: Goldstein and Galbally, 2007.
**Goldstein, A.H. and Galbally, I.E.: Known and Unexplored Organic Constituents in the Earth's Atmosphere, Environ. Sci. Technol. 41 (5), 1514-1521, DOI: 10.1021/es072476p, 2007.

- P8L35: This is not completely true, ice supersaturation can/is also be achieved when water-saturation conditions are created in SPIN. Please rephrase and formulate more carefully.

Changed to "Because of the nonlinear relationship between saturation vapor pressure and temperature, supersaturation with respect to ice, in water-subsaturated conditions up to and including water-saturation, is achieved  along the center of the chamber, allowing for ice nucleation."

- P9L16: Are your ice nucleation data corrected for multiple charged particles? Please specify the sizes of double charged particles for the mobility diameters used by you. Could it be that the OPC detects a large, double-charged BC particle as ice?

This was answered in detail to comment **P10L21-23** of Reviewer2.
The data is not corrected for doubly charged particles; however, the 800 nm mobility diameter particles are essentially on the larger-diameter edge of the size distribution of BC particles. Soot's main mode is on the order of several hundreds of nanometers as was shown in the figure attached in that comment in previous reply..
The doubly charged particles, of the 800 nm singly charged particles, would need to be approximately 1.48 micron in diameter (fractal dimension of 3 assumed due to sphericity) and literally in negligible concentration, and have higher losses in the lines. Moreover, the OPC detected particles only in the first 2 bins (associated with the 800 nm mobility diameter) but not in any other higher size bins.
In any case, if there were doubly charged particles (1.48 micron) present, the OPC would not detect them as ice since the OPC classification is based on several scattering and polarization

parameters rather than size alone. These parameters have thresholds for aerosol, water, and ice (see Garimella et al., 2016).

In regard to 100 nm ethylene soot particles, we expected all the 100 nm BC INP to freeze at the homogeneous line (see Mahrt et al. 2018). One possible explanation that it did not reach homogeneous nucleation, i.e. nucleate at a higher SS% but still below Koops line, is the presence of doubly charged particles (~151 nm) or triply charged (~195 nm), which are bigger agglomerates and nucleate better than the single aggregates, therefore improving the observed IN activity of the selected mobility diameter. However, they are not as active as if these were solely big particles. The broader PSD of flame generated soot is expected to have more multiply charged particles in comparison to commercial well defined batches of pigments.

Therefore, we state in the text of Sect 3.1, point (3), last few sentences: "Despite the reduction in activity, the 100 nm soot has nucleated ice below homogeneous freezing conditions. It is possible that a bias introduced by multiply charged particles, passed at the same DMA voltage, maintained the high IN activity of 100 nm mobility diameter soot".

*Garimella, S., Kristensen, T. B., Ignatius, K., Welti, A., Voigtländer, J., Kulkarni, G. R., Sagan, F., Kok, G. L., Dorsey, J., Nichman, L., Rothenberg, D. A., Rösch, M., Kirchgäßner, A. C. R., Ladkin, R., Wex, H., Wilson, T. W., Ladino, L. A., Abbatt, J. P. D., Stetzer, O., Lohmann, U., Stratmann, F., and Cziczo, D. J.: The SPectrometer for Ice Nuclei (SPIN): an instrument to investigate ice nucleation, Atmos. Meas. Tech., 9, 2781-2795, https://doi.org/10.5194/amt-9-2781-2016, 2016.

- P9L24: Can you set this 1% threshold into context of the maximal activate fraction observed during your RH-scans. Please comment on this in the manuscript.

At highest supersaturation, we reached RH above homogeneous nucleation. In this case, depositional activation percentage is not defined because of the homogeneous freezing conditions. Above homogeneous freezing we reached 100% activation. In other cases, when the scan was stopped  below homogeneous freezing the maximal percentage varied 30 - 80 %. New sentences:
*"This figure shows a plot of water vapor supersaturation ratio with respect to ice versus temperature at which (1 %) ice onset occurs. As the scan continues, activated fraction increases.  Finally, full (100 %) activation is observed if  water saturation has been reached".*

- P9L34: I could not find the definition for "SSi".

Changed to read, "The relationship between the supersaturation with respect to ice at the ice nucleation onset, SSi, and temperature…

- P10L4: Add Mahrt et al. (2018)

"...similar to some of the earlier observations of soot (e.g.,...)"
Here we refer to earlier experimental observations of soot. Mahrt et al. 2018 was conducted in parallel with this study, and it is already mentioned in several places in this manuscript.

- P10L7: Why do you write "Heterogeneous ice nucleation". On P4L10 you state that water in pores freezes "homogenously". Please clarify in manuscript.

P4L10 currently reads, "The PCF mechanism proposes that empty spaces between aggregated primary particles fill with water due to capillary condensation at relative humidities (RHw) below water saturation, which freezes homogeneously (Marcolli, 2014; Christenson (2013); Higuchi and Fukuta (1966))."

We have added the following sentence to the manuscript on P4L11 (following the above sentence):

"The PCF mechanism, which models ice formation inside small pores using homogeneous freezing theory, describes the heterogeneous formation of ice on solid particles, such as soot particles, due to the presence of the porous structure. Therefore, we refer to PCF as a heterogenous ice freezing mechanism (Vali et al., 2015)."

- P10L13: "The displayed…" This statement is not supported by your data shown in Fig. 4, in particular by the data points around T = 228 K. Please clarify. This would become even clearer if you were to include error bars in your data points, similar to your Fig. 3.

This one scan when taken out of context doesn't represent well the observed trend described here. We tuned down the sentence to be more accurate.
Previous:
*"The displayed onset supersaturation difference shows that the IN activity of R2500U is significantly higher than that of Regal 330R"*
New:
*"The displayed onset supersaturation difference shows that the IN activity of R2500U is higher than that of Regal 330R for the majority of runs in the temperature range 228 – 233 K".*

All the error bars and the compared data are shown already in Fig. 3. An addition of error bars in Fig.4 will overcrowd the figure, essentially in the same way as seen in Fig.3.
We added references to both Figs 3,4 in the text for reference to error bars:
*"The most notable feature in Figs. 3, 4 is the gradient between the data sets for BC particles R2500U (red squares) and Regal 330R (yellow squares), including the BC types in between".*

- P10L29: Again, it is very hard to justify and/or follow this statement without error bars in Fig. 4.

This was answered in detail to comment P9L28-31 of Reviewer2,

It is clear to the authors that there are some overlaps in data points of the experimental results; however, least squares fitted curves, suggested in the previous version of the manuscript show a clearer difference between the datasets of each compound.

All the error bars needed and the compared data are shown in Fig. 3. An addition of error bars in Fig.4 will overcrowd the figure essentially in the same way as seen in Fig.3.
We added references to both Figs 3,4 in the text for reference to error bars:
*"Therefore, in accord with the PCF mechanism, one would expect Monarch 880 to display lower IN activity in comparison to Monarch 900 and R2500U, as is observed (Figs. 3, 4)".*

- P11L33: The reported size dependence seems consistent with that observed in studies by Friedman et al. (2011), Mahrt et al. (2018) and Crawford et al. (2011) that you cite above. You might want to refer back to these studies.

"To test the extremes of the aerosol diameters used in previous studies we selected 100 nm and 800 nm BC particles".
We have included references of previous studies. Mahrt et al. 2018 was conducted in parallel with this study.

- P11L36: Replace "," by ".".

Done.

- P12L2: "Hence, the…": I assume that every data point in your Fig. 6 represents a single run. Please comment on how reproducible these are within the manuscript. Also, if I consider the error bars reported in your Fig. 3 I am not sure how different your 100 and 800 nm ethylene soot in Fig. 6 are. Please include error bars in Figs. and in discussion of data.

Yes this is correct, each data point represents a single run (P9L20). The blue squares are single experiments and thus the two runs at the same temperature represent a (crude) measure of reproducibility.  The error bars in Fig.3  represent the variability of the laminar conditions based on CFD simulations by Kulkarni and Kok (2012) (P9L27) and are estimates of accuracy (based on T and RH uncertainties), whereas multiple measurements per temperature gives idea of precision (reproducibility) of the measurements.
Fig.3 doesn't include 100 nm data points with error bars,  therefore we agree that these should be presented in Fig. 6.
Text changed:
*"The error bars of Monarch 880 data points partially overlap in some of the 100 and 800 nm*

*runs. Nonetheless, some runs don't have an overlap in uncertainty at the same temperature and*

*the trend of lower IN activity, in 100 nm particles, repeats itself. Similarly, the IN activity of 100*

*nm ethylene flame soot is reduced in comparison to the 800 nm soot. Despite the reduction in*

*activity, the 100 nm soot has nucleated ice below homogeneous freezing conditions. It is possible that a bias introduced by multiply charged particles of flame soot's broad size distribution, passed at the same DMA voltage, maintained the high IN activity of 100 nm mobility diameter soot.*

- P13L7: Delete space between "sample" and "."

Done.

- P13L10: Include space before "However"

Done.

- P13L27: Delete space before "The"

Done.

- P14L10: Please define "channel process"

We have added: "..., utilizing natural gas impingement on iron channels to produce carbon black..."
More information can be found in the cited papers and references therein.

- P14L17: Please change to "INP"

Done.

- P14L32: Consider to replace "down" with "along"

Done.

- P16L3: Who is "MF"?

Thank you.  Changed to "MW".

- Figs. 3, 4, 6, 7, 8: The y-axis should be labeled as "Supersaturation ratio with respect to ice", or simply "Sice", to be consistent with your terminology. "Supersaturation over ice" as used is wrong. I recommend to be consistent through out all figures within the manuscript.

Y axes changed.

Crawford, I., et al. (2011), 'Studies of propane flame soot acting as heterogeneous ice nuclei in conjunction with single particle soot photometer measurements', Atmospheric Chemistry and Physics, 11 (18), 9549-61.

David, R. O., et al. (2019), 'Pore condensation and freezing is responsible for ice formation below water saturation for porous particles', Proceedings of the National Academy of Sciences of the United States of America, 116 (17), 8184-89.

Friedman, B., et al. (2011), 'Ice nucleation and droplet formation by bare and coated soot particles', Journal of Geophysical Research-Atmospheres, 116.

Häusler, T., et al. (2018), 'Ice Nucleation Activity of Graphene and Graphene Oxides', Journal of Physical Chemistry C, 122 (15), 8182-90.

Mahrt, F., et al. (2018), 'Ice nucleation abilities of soot particles determined with the Horizontal Ice Nucleation Chamber', Atmospheric Chemistry and Physics, 18 (18), 13363-92.

Umo, N. S., et al. (2019), 'Enhanced ice nucleation activity of coal fly ash aerosol particles initiated by ice-filled pores', Atmos. Chem. Phys., 19 (13), 8783-800.